# ERROR ANALYSIS OF DISCRETE FLOW WITH GENERATOR MATCHING

## ABSTRACT

Discrete flow models offer a powerful framework for learning distributions over discrete state spaces and have demonstrated superior performance compared to the discrete diffusion model. However, their convergence properties and error analysis remain largely unexplored. In this work, we develop a unified framework grounded in stochastic calculus theory to systematically investigate the theoretical properties of discrete flow. Specifically, we derive the KL divergence of two path measures regarding two continuous-time Markov chains (CTMCs) with different transition rates by deriving a Girsanov-type theorem, and provide a comprehensive analysis that encompasses the error arising from transition rate estimation and early stopping, where the first type of error has rarely been analyzed by existing works. Unlike discrete diffusion models, discrete flow incurs no truncation error caused by truncating the time horizon in the noising process. Building on generator matching and uniformization, we establish non-asymptotic error bounds for distribution estimation. Our results provide the first error analysis for discrete flow models.

## 1 INTRODUCTION

Discrete diffusion models have achieved significant progress in large language models (Nie et al., 2025; Zhu et al., 2025; Zhao et al., 2025; Yang et al., 2025). By learning the time reversal of the noising process of a continuous-time Markov chain (CTMC), the models transform a simple distribution (e.g., uniform (Hoogeboom et al., 2021; Lou et al., 2024) and masked (Ou et al., 2025; Shi et al., 2024; Sahoo et al., 2024)) that is easy to sample to the data distribution that has discrete structures. Discrete flow models Campbell et al. (2024); Gat et al. (2024); Shaul et al. (2025) provide a flexible framework for learning generating transition rate analogous to continuous flow matching (Albergo & Vanden-Eijnden, 2023; Liu et al., 2023; Lipman et al., 2023), offering a more comprehensive family of probability paths.

Recent theoretical analysis for discrete diffusion models has emerged through numerous studies (Chen & Ying, 2024; Zhang et al., 2025; Ren et al., 2025a;b). To obtain the transition rate in the reversed process, the concrete scores in these analyses are obtained by minimizing the concrete score entropy introduced in Lou et al. (2024); Benton et al. (2024c). In those works, the distribution errors of discrete diffusion models are divided into three parts: (a) truncation error from truncating the time horizon in the noising process; (b) concrete score estimation error; (c) discretization error from sampling algorithms. In our paper, we aim to investigate the theoretical properties of the discrete flow-based models using the generator matching training objective (Holderrieth et al., 2025) and the uniformization sampling algorithm (Chen & Ying, 2024), which offers zero truncation error and discretization error. Our analysis takes transition rate estimation error into account instead of imposing a stringent condition on it in previous works, which is related to the early stopping parameter. We decompose the estimation error into stochastic error and approximation error, and then balance the stochastic error and early stopping error by choosing the early stopping parameter. Our stochastic error bound is aligned with the SOTA result in continuous flow (Gao et al., 2024b); the early stopping error bound matches the most recent result in discrete diffusion (Zhang et al., 2025). Furthermore, we present a comprehensive error analysis for the neural network class with the ReLU activation function, by controlling stochastic error, approximation error and early stopping error simultaneously.

The main contributions in this paper are summarized as follows.

1. We obtain a Girsanov-type theorem for CTMC with rigorous proofs. The KL divergence of two path measures of two CTMCs is derived in terms of the integral of the Bregman divergence of two transition rates, which motivates us to use the generator matching objective to analyze the distribution error. In the sampling stage, we use the uniformization technique to sample in an exact way.

2. We establish the non-asymptotic error bound for distribution error in discrete flow models, by taking estimation error into consideration, which is less explored in existing works. There are three sources of error in our framework: (a) stochastic error from estimation through empirical risk minimization; (b) approximation error of the selected function class; (c) early stopping error. We carefully analyze the stochastic error using empirical process theory, and discuss the choice of early stopping parameter to balance the stochastic error and early stopping error. We also provide a comprehensive analysis of our assumptions and the terminal time singularity of the transition rate. In addition, we present a thorough error analysis for the neural network class with the ReLU activation function, simultaneously analyzing these three sources of error. To the best of our knowledge, this is the first theoretical error analysis for discrete flow models.

All technical proofs are deferred to the Appendix.

NOTATION. Let $[N] = \{1, 2, \ldots, N\}$ for a positive integer $N$. We use $\dot{\kappa}_t$ to denote the time derivative of a function $\kappa_t$ of $t$. For a $\mathcal{D}$-dimensional vector $z$, let $z^d$ and $z^{\backslash d}$ denote the $d$-th element of the vector $z$ and the $(\mathcal{D}-1)$-dimensional vector $(z^1, \ldots, z^{d-1}, z^{d+1}, \ldots, z^{\mathcal{D}})^\top$. We use $\mathbb{1}(\cdot)$ to denote an indicator function. We denote the Hamming distance of two vectors $z, x$ by $\mathrm{d}^H(z, x)$. For two quantities $x, z$, define the Kronecker delta $\delta_x(z)$ satisfying $\delta_x(z) = 1$ if $x = z$ and $\delta_x(z) = 0$ if $x \neq z$. For a random data $\mathbb{D}_n = \{Z_i\}_{i \in [n]}$, we denote $\|f(Z)\|_{L^\infty(\mathbb{P}_n)} = \max_{i \in [n]} |f(Z_i)|$, where $\mathbb{P}_n$ is the empirical measure of $\mathbb{D}_n$. In this paper, some universal constants $C, c > 0$ are allowed to vary from line to line.

## 2 RELATED WORKS

**Discrete Flow.** Discrete flow models (Campbell et al., 2024; Gat et al., 2024; Shaul et al., 2025) provide a more flexible framework to construct the transition rate and probability path than discrete diffusion models (Campbell et al., 2022; Lou et al., 2024), which also achieve superior performance in graph generation (Qin et al., 2025), visual generation and multimodal understanding (Wang et al., 2025). The training objective of discrete flow models can be cross-entropy (Campbell et al., 2024; Gat et al., 2024), negative ELBO (Shaul et al., 2025), or Bregman divergence (Holderrieth et al., 2025). In this work, we use the estimator through minimizing the empirical version of Bregman divergence to control the distribution error.

**Theoretical Analysis of Discrete Diffusion.** Discrete diffusion models (Austin et al., 2021; Campbell et al., 2022; Vignac et al., 2023; Sun et al., 2023; Lou et al., 2024; Benton et al., 2024c) have emerged as a CTMC-based framework for learning distribution on finite state spaces. Some recent works (Chen & Ying, 2024; Zhang et al., 2025; Ren et al., 2025a;b) investigate the theoretical property of discrete diffusion models. Chen & Ying (2024) proposed to use uniformization algorithm for sampling in an exact way, and derived the distribution error bound for the scenario where the state space is a hypercube. Zhang et al. (2025) studied the theoretical results of discrete diffusion based on Girsanov-based method in the general state space $\mathcal{S}^{\mathcal{D}}$. Ren et al. (2025a) derived the error bound for both $\tau$-leaping and uniformization algorithms using stochastic integrals. Ren et al. (2025b) proposed high-order solvers for sampling and derived error bound similar to Ren et al. (2025a), which enjoy more accuracy than $\tau$-leaping algorithm. The training objective in these literature is the concrete score entropy introduced in Lou et al. (2024); Benton et al. (2024c). Unfortunately, in existing works, it is typical to assume strong regularity conditions directly on the estimation error, which is related to the early stopping parameter. Furthermore, in discrete diffusion models, there is a non-zero truncation error arising from the truncation of the time horizon in the noising process.

**Theoretical Analysis of Continuous Flow.** Continuous flow matching (Liu et al., 2023; Albergo & Vanden-Eijnden, 2023; Lipman et al., 2023) is a powerful simulation-free method for learning continuous data distribution based on continuous normalizing flow (Chen et al., 2018). Error bounds on the Wasserstein distance between two flow ODEs have been extensively studied (Albergo & Vanden-Eijnden, 2023; Benton et al., 2024b; Gao et al., 2024a;b). Albergo & Vanden-Eijnden (2023) and

Benton et al. (2024b) derived the Wasserstein bound in terms of the spatial Lipschitz constants of the velocity fields by utilizing Grönwall's inequality. Gao et al. (2024a) established the spatial Lipschitz regularity of the velocity field for a range of target distributions. Gao et al. (2024b) presented a comprehensive error analysis of continuous flow matching by rigorously bounding four types of errors, including early stopping error, discretization error, stochastic error and approximation error.

# 3 THEORETICAL BACKGROUND ON DISCRETE FLOW-BASED MODELS

In this section, we introduce some theoretical background about continuous-time Markov chains (Norris, 1998) and discrete flow-based models (Campbell et al., 2024; Gat et al., 2024; Shaul et al., 2025; Wang et al., 2025).

## 3.1 CONTINUOUS-TIME MARKOV CHAIN AS STOCHASTIC INTEGRAL

In this subsection, we consider the probability space $(\Omega, \mathcal{F}, \mathbb{P})$. We formally define continuous-time Markov chains (CTMCs) as follows.

**Definition 1** (CTMC). *Consider a $\mathcal{D}$-dimensional finite state space $\mathcal{S}^{\mathcal{D}}$, where $\mathcal{S} = \{1, 2, \ldots, |\mathcal{S}|\}$. Let $u_t(z, x)_{z,x \in \mathcal{S}^{\mathcal{D}}}$ be a (time-dependent) rate matrix that satisfying: (a) $u_t(z, x)$ is continuous in $t$; (b) $u_t(z, x) \geq 0$ for any $z \neq x$; (c) $\sum_{z \in \mathcal{S}^{\mathcal{D}}} u_t(z, x) = 0$ for any $x \in \mathcal{S}^{\mathcal{D}}$. We call a process $\{X(t)\}_{t \geq 0}$ a continuous time Markov chain with the transition rate matrix $u_t(z, x)_{z,x \in \mathcal{S}^{\mathcal{D}}}$ and the natural filtration $\mathcal{F}_t = \sigma(\{X(s) : 0 \leq s \leq t\})$ if it satisfies that for any $z, x \in \mathcal{S}^{\mathcal{D}}$*

1. *the transition rate is the generator of CTMC: $\mathbb{P}(X(t + h) = z | X(t) = x) = \delta_x(z) + u_t(z, x)h + o(h)$;*

2. *it has Markov property: $\mathbb{P}(X(t + h) = z | \mathcal{F}_t) = \mathbb{P}(X(t + h) = z | X(t))$.*

Given a rate matrix $u_t$ with uniformly bounded entries, the following uniformization technique allows us to construct a CTMC with $u_t$ through a Poisson process, which offers us a sampling algorithm without discretization error (see, e.g., Ren et al., 2025a; Chen & Ying, 2024).

**Proposition 1** (Uniformization). *Assume that $u_t(z, x)_{z,x \in \mathcal{S}^{\mathcal{D}}}$ is a rate matrix satisfying that (a) $-u_t(x, x) \leq M$ for any $x \in \mathcal{S}^{\mathcal{D}}$; (b) $u_t(z, x)$ is $L$-Lipschitz continuous in $t$ for any pair $(z, x) \in \mathcal{S}^{\mathcal{D}} \times \mathcal{S}^{\mathcal{D}}$. Suppose that $T_1, T_2, \ldots$ are the arrival times of a Poisson process $N(t)$ with rate $M$. Let $X(t)$ be a jump process with initial distribution $p_0$ and natural filtration $\mathcal{F}_t$ such that at $t = T_1, T_2, \ldots$, the process $X(t)$ jumps to position $z \neq X(t-)$ with probability $u_t(z, X(t-))/M$. Then $X(t)$ is a Markov process satisfying for $z \neq x$,*

$$\mathbb{P}(X(t + h) = z | X(t) = x) = u_t(z, x) + R_t,$$

*where $R_t \leq (M^2 + L)h^2 = O(h^2)$. Thus, $X(t)$ is a CTMC with the rate $u_t$.*

Here, the remainder $R_t$ is uniformly bounded for any $t$ and $z \neq x$ under the assumptions in Proposition 1, which is crucial for developing the theory of CTMC.

To rewrite a CTMC $X(t)$ as a stochastic integral, we define a random measure $N((t_1, t_2], A)$ associated with the CTMC $X(t)$.

**Definition 2** (Random Measure Associated with CTMC). *Suppose that $X(t)$ is a CTMC with rate $u_t$ and natural filtration $\mathcal{F}_t$. Define the random measure associated with $X(t)$ and $A \subseteq \mathcal{S}^{\mathcal{D}}$ as*

$$N((t_1, t_2], A) = \#\{t_1 < s \leq t_2; \Delta X(s) \neq 0, X(s) \in A\},$$

*where $\Delta X(t) = X(t) - X(t-)$.*

Suppose that $N((t_1, t_2], A)$ is the random measure associated with a CTMC $X(t)$, where $X(t)$ is constructed by uniformization with the Poisson process $N(t)$. Then $N(t, A) \stackrel{\triangle}{=} N((0, t], A) \leq N(t) < \infty$ a.s. for any $t \geq 0$ and $A \subseteq \mathcal{S}^{\mathcal{D}}$ (see Lemma 2.3.4 in Applebaum, 2009). Thus, similar to the Poisson random measure associated with a Lévy process (see Section 2.3.2 in Applebaum, 2009), there are some equivalent representations:

$$N((t_1, t_2], A) = \sum_{t_1 < s \leq t_2} \mathbb{1}_A((X(s)))(1 - \delta_{X(s-)}(X(s))) = \sum_{n \in \mathbb{N}} \delta_{(t_1, t_2]}(T_n^A),$$

where $\{T_n^A\}_{n\in\mathbb{N}}$ are arrival times of the counting process $N(t,A)$. Consequently, $X(t)$ can be written as the following stochastic integral:

$$
\begin{aligned}
X(t) &= X(0) + \sum_{n\in\mathbb{N}} \Big[ X(T_n) - X(T_{n-1}) \Big] \delta_{[0,t]}(T_n) \\
&= X(0) + \sum_{n\in\mathbb{N}} \sum_{z} \Big[ z - X(T_n-) \Big] \mathbb{1}_z(X(T_n)) \delta_{[0,t]}(T_n) \\
&= X(0) + \int_0^t \int_{\mathcal{S}^{\mathcal{D}}} (z - X(s-)) N(\mathrm{d}s, \mathrm{d}z),
\end{aligned}
\tag{1}
$$

where $\{T_n\}_{n\in\mathbb{N}}$ are arrival times of $N(t,\mathcal{S}^{\mathcal{D}})$. Here, we formally define the integrator $N(t,A)$ in the CTMC representation as Definition 2, which is clearer and more formal than the relevant definition compared to Proposition 3.2 in Ren et al. (2025a).

**Remark 1** (Comparison to Lévy-Itô Decomposition). *Recall that a Lévy process $X(t)$ has the following Lévy-Itô decomposition (see Theorem 2.4.16 in Applebaum, 2009):*

$$
X(t) = X(0) + bt + B(t) + \int_{|x|<1} x\tilde{N}(t,dx) + \int_{|x|\geq 1} xN(t,dx),
$$

*where $B(t)$ is a Brownian motion, $N(t,A)$ is the Poisson random measure associated with the Lévy process $X(t)$, and $\tilde{N}(t,A)$ is the compensator of $N(t,A)$. Here, the second argument in $N(t,A)$ is the jump size of the associated Lévy process at time $t$. However, the second argument of the random measure associated with a CTMC in Definition 2 is the position after jump at time $t$. Both the integrands in equation 1 and that in Lévy-Itô decomposition are jump size. Moreover, since CTMCs do not have independent increments, the random measure in Definition 2 is not independently scattered; that is, $N(t,A_1)$ and $N(t,A_2)$ are not necessarily independent for disjoint $A_1, A_2 \subseteq \mathcal{S}^{\mathcal{D}}$, which is not a Poisson random measure. There is no need to use the independently scattered property in our technical proofs.*

Given a CTMC $X(t)$ with transition rate $u_t$ and marginal densities $\{p_t\}_{t\geq 0}$ w.r.t. the counting measure, the following proposition demonstrates that the marginal densities $p_t$ satisfy the following Kolmogorov forward equation (a.k.a. continuity equation).

**Proposition 2** (Kolmogorov Forward Equation). *The CTMC $X(t)$ satisfies the following equation:*

$$
\dot{p}_t(x) = \sum_{z\in\mathcal{S}^{\mathcal{D}}} u_t(x,z)p_t(z) = \underbrace{\sum_{z\neq x} u_t(x,z)p_t(z)}_{\text{Incoming Flux}} - \underbrace{\sum_{s\neq x} u_t(z,x)p_t(x)}_{\text{Outgoing Flux}}.
$$

In our work, we say $u_t$ can generate the probability path $p_t$, if it satisfies the above Kolmogorov equation.

## 3.2 DISCRETE FLOW-BASED MODELS

We aim to learn a transition rate $u_t$ of a CTMC $X(t)$ that can transport from a source distribution $p_0$ to a target data distribution $p_1$. To obtain such a transition rate for sampling, a natural method is to learn the conditional expectation of the conditional transition rate $u_t(z,x|x_1)$ that generates the conditional probability path $p_{t|1}(\cdot|x_1)$, since $u_t(z,x) = \mathbb{E}[u_{\mathbf{t}}(z,x|X(1))|X(\mathbf{t})=x,\mathbf{t}=t]$ can generate the target probability path $p_t$ (see Proposition 3.1 of Campbell et al., 2024), i.e., it satisfies the Kolmogorov forward equation, where $X(\mathbf{t}) \sim p_{\mathbf{t}|1}(\cdot|X(1))$ given $X(1)$ and $\mathbf{t} \sim \mathcal{U}([0,1])$. Therefore, we are free to define the conditional probability path and the conditional transition rate.

It is worth noting that in the sampling stage, considering a $|\mathcal{S}|^{\mathcal{D}}$-dimensional vector-valued function $(u_t(z,x))_{z\in\mathcal{S}^{\mathcal{D}}}$ of current time $t$ and state $x$ is intractable when $\mathcal{D}$ is relatively large. To handle such a high-dimensional scenario, a common approach is to construct a coordinate-wise conditional probability path and transition rate, that is,

$$
p_{t|1}(x|x_1) = \prod_{d=1}^{\mathcal{D}} p_{t|1}^d(x^d|x_1^d), \text{ and } u_t(z,x|x_1) = \sum_{d=1}^{\mathcal{D}} \delta_{x^{\backslash d}}(z^{\backslash d}) u_t^d(z^d, x^d|x_1^d),
\tag{2}
$$

which means that the elements of the vector $X(t)$ are independent conditional on $X(1)$. Here, $u_t^d(z^d, x^d | x_1^d)$ is the conditional transition rate that generates the conditional probability path $p_{t|1}^d$. A popular choice of probability path and the associated conditional transition rate used in the previous works (Campbell et al., 2024; Gat et al., 2024) is

$$p_{t|1}^d(x^d | x_1^d) = (1 - \kappa_t) p_0^d(x^d) + \kappa_t \delta_{x_1^d}(x^d); \ u_t^d(z^d, x^d | x_1^d) = \frac{\dot{\kappa}_t}{1 - \kappa_t} (\delta_{x_1^d}(z^d) - \delta_{x^d}(z^d)), \quad (3)$$

where $\kappa_t : [0,1] \to [0,1]$ is a non-decreasing function satisfying $\kappa_0 = 0$ and $\kappa_1 = 1$. Note that the conditional transition rate will blow up as $t \to 1^-$. Thus, in the sampling stage, we employ the early stopping technique; that is, we only consider the time interval $[0, 1 - \tau]$ for a sufficiently small positive parameter $\tau$. We will discuss this time singularity in Section B.1.

After defining the conditional path and rate, the unconditional transition rate is given by

$$u_t(z, x) = \sum_{x_1} u_t(z, x | x_1) p_{1|t}(x_1 | x) = \sum_{d=1}^{\mathcal{D}} \delta_{x^{\backslash d}}(z^{\backslash d}) \sum_{x_1^d} u_t^d(z^d, x^d | x_1^d) p_{1|t}^d(x_1^d | x)$$

$$\triangleq \sum_{d=1}^{\mathcal{D}} \delta_{x^{\backslash d}}(z^{\backslash d}) u_t^d(z^d, x), \quad (4)$$

where $p_{1|t}^d(x_1^d | x) = \sum_{x_1^{\backslash d}} p_{1|t}(x_1 | x)$. Therefore, it suffices to consider a sparse rate matrix $u_t$ satisfying $u_t(z, x) = 0$ for any $z, x \in \mathcal{S}^{\mathcal{D}}$ with Hamming distance $d^H(z, x) > 1$.

**Remark 2.** *With a slight abuse of notation, throughout this article, the summation over the state space is restricted to states where the Hamming distance is not larger than 1; specifically, we use $\sum_{z \in A} = \sum_{z \in A : d^H(z, X(t-)) \leq 1}$.*

In our work, similar to Chen & Ying (2024); Zhang et al. (2025), we consider the *uniform* source distribution. There are three reasons: (a) a uniform distribution has full support, which is similar to the Gaussian distribution used in continuous flow-based models; (b) in any state, each coordinate can possibly be changed in the sampling stage, compared to an absorbing source distribution; (c) it has a good performance in practice (e.g., flow-based multimodal large language models Wang et al., 2025).

**Training via Bregman divergence.** Let $\tau \in (0, 1/2)$ be an early stopping parameter. Suppose that we have i.i.d. samples $\{\mathbf{t}_i, X_i(1)\}_{i \in [n]}$, and $X_i(\mathbf{t}_i) \sim p_{\mathbf{t}_i|1}(\cdot | X_i(1))$ for each $i \in [n]$, where $\mathbf{t}_i$ samples from the uniform distribution $\mathcal{U}([0, 1 - \tau])$. Denote $\mathbb{D}_n = \{Z_i\}_{i \in [n]} = \{(\mathbf{t}_i, X_i(\mathbf{t}_i), X_i(1))\}_{i \in [n]}$ and $v(x, Z_i) = u_{\mathbf{t}_i}(x, X_i(\mathbf{t}_i) | X_i(1))$, where $u_t(z, x | x_1)$ is a conditional transition rate that generates the conditional probability path $p_{t|1}(x | x_1)$. Denote the Bregman divergence with a convex function $F(\cdot)$ as $D_F$, which is defined as

$$D_F(a || b) = F(a) - F(b) - \langle \nabla F(b), a - b \rangle.$$

Inspired by generator matching Holderrieth et al. (2025), we consider the following rate estimator through empirical risk minimization (ERM) with a function class $\mathcal{G}_n$:

$$\hat{u} = \arg\min_{u \in \mathcal{G}_n} \frac{1}{n} \sum_{i=1}^n \sum_{z \neq X_i(\mathbf{t}_i)} D_F(v(z, Z_i) || u_{\mathbf{t}_i}(z, X_i(\mathbf{t}_i)))$$

$$= \arg\min_{u \in \mathcal{G}_n} \frac{1}{n} \sum_{i=1}^n \sum_{z \neq X_i(\mathbf{t}_i)} \left\{ -u_{\mathbf{t}_i}(z, X_i(\mathbf{t}_i) | X_i(1)) \log u_{\mathbf{t}_i}(z, X_i(\mathbf{t}_i)) + u_{\mathbf{t}_i}(z, X_i(\mathbf{t}_i)) \right\} \quad (5)$$

$$\triangleq \arg\min_{u \in \mathcal{G}_n} \mathcal{L}_n(u),$$

where we take $F(x) = x \log x$ and the summation over $\{z : z \neq X_i(\mathbf{t}_i)\}$ has only $O(\mathcal{D}|S|)$ complexity due to the sparsity of the rate matrix (Remark 2). Here, the training objective is finite if we choose an appropriate function class satisfying Assumption 2 in Section 5.

**Sampling via uniformization.** For accurate sampling without discretization error, following the uniformization argument introduced in Proposition 1, we consider the uniformization Algorithm 1 in the Appendix (also see, e.g. Algorithm 1 in Chen & Ying, 2024). Here, the discretization used in Algorithm 1 (see Section A in the Appendix) can reduce the average sampling steps compared to using a uniform bound $M \geq \sup_{t \in [0, 1-\tau]} \sum_{z \neq x} \hat{u}_t(z, x)$ for a large time interval $[0, 1 - \tau]$.

## 4 CHANGE OF MEASURE AND BOUND FOR KL DIVERGENCE

In this section, we develop a Girsanov-type theorem for CTMCs. As a result, we establish the bound for the KL divergence of two marginal distributions regarding two CTMCs with different rates.

First, we derive the compensator of the random measure $N(t, A)$ defined in Definition 2, where $A \subseteq \mathcal{S}^{\mathcal{D}}$.

**Proposition 3** (Compensator). *Under the assumptions in Proposition 1, the (martingale-valued) compensated random measure of $N(t, A)$ is*

$$\tilde{N}(t, A) = N(t, A) - \int_0^t \sum_{z \in A} u_s(z, X(s-)) \mathbb{1}(z \neq X(s-)) ds.$$

### 4.1 CHANGE OF MEASURE

Suppose that $X(t)$ is a CTMC with transition rate $u^X$ and natural filtration $\mathcal{F}_t$ under probability space $(\Omega, \mathcal{F}, \mathbb{P}^X)$. Our goal is to find a probability measure $\mathbb{P}^Y \ll \mathbb{P}^X$ such that $X(t)$ is a $\mathbb{P}^Y$-CTMC with a rate $u^Y$. Denote $N(t, A)$ as the random measure associated with $X(t)$. Let $\exp(W(t))$ be an exponential $\mathbb{P}^X$-martingale (see Lemma 3 in Appendix), where $W(t)$ has the following form:

$$W(t) = W(0) + \int_0^t \sum_{x \neq X(s-)} F(s, x) ds + \int_0^t \int_{x \in \mathcal{S}^{\mathcal{D}}} K(s, x) N(ds, dx). \quad (6)$$

Since $\exp(W(t))$ is a $\mathbb{P}^X$-martingale with mean one, we can define $\frac{d\mathbb{P}^Y}{d\mathbb{P}^X}\Big|_t = \frac{d\mathbb{P}^Y_t}{d\mathbb{P}^X_t} = \exp(W(t))$, where $\mathbb{P}_t$ is the restriction of the measure $\mathbb{P}$ on $(\Omega, \mathcal{F}_t)$. If we take some specific predictable processes $F$ and $K$, the following theorem shows that

$$\tilde{N}^Y(t, A) \stackrel{\triangle}{=} N(t, A) - \int_0^t \sum_{z \in A} u_s^Y(z, X(s-)) \mathbb{1}(z \neq X(s-)) ds$$

is a $\mathbb{P}^Y$-martingale, and $X(t)$ is a $\mathbb{P}^Y$-CTMC with rate $u_t^Y$.

**Theorem 1** (Change of Measure). *Assume that both transition rates $u_t^X$ and $u_t^Y$ satisfy the conditions in Proposition 1. Suppose that $u_t^X(x, X(t-)) = 0$ implies $u_t^Y(x, X(t-)) = 0$. Choosing the predictable processes*

$$K(t, x) = \log \frac{u_t^Y(x, X(t-))}{u_t^X(x, X(t-))} \quad and \quad F(t, x) = (u_t^X(x, X(t-)) - u_t^Y(x, X(t-))) \mathbb{1}(x \neq X(t-))$$

*in Equation 6. The R-N derivative can be written as $\frac{d\mathbb{P}^Y}{d\mathbb{P}^X}\Big|_t = \exp(W(t))$, where*

$$W(t) = \int_0^t \sum_{x \neq X(s-)} [u_s^X(x, X(s-)) - u_s^Y(x, X(s-))] ds + \int_0^t \sum_{x \neq X(s-)} \log \frac{u_s^Y(x, X(s-))}{u_s^X(x, X(s-))} N(ds, x).$$

*Then $\tilde{N}^Y(t, A)$ is a $\mathbb{P}^Y$-martingale for any $A \subseteq \mathcal{S}^{\mathcal{D}}$. Moreover, $X(t)$ is a $\mathbb{P}^Y$-CTMC with rate $u_t^Y$.*

Theorem 1 is similar to Theorem F.12 in Pham et al. (2025), and can also be derived by applying Theorem F.12 in Pham et al. (2025) twice, following the argument in their proof of Theorem 2.3. Compared to the Girsanov's theorem used in Zhang et al. (2025), our result can be applied to two CTMCs with arbitrary transition rates. In our result, we provide an explicit expression of the RN derivative in terms of two transition rates. This form is closely related to Theorem 9 in Chen et al. (2023) and Theorem 5.2.12 in Applebaum (2009) for Brownian motion with drift.

### 4.2 BOUND FOR KL DIVERGENCE

Suppose that $X(t)$ has marginal distribution $p_t^X$ on the probability space $(\Omega, \mathcal{F}, \mathbb{P}^X)$ and marginal distribution $p_t^Y$ on the probability space $(\Omega, \mathcal{F}, \mathbb{P}^Y)$. According to Theorem 1, we can derive the bound for KL divergence of $p_t^X$ and $p_t^Y$.

**Theorem 2** (Bound for KL Divergence). *Suppose that $X(t)$ is a CTMC with natural filtration $\mathcal{F}_t$ and rate $u_t^X$ under measure $\mathbb{P}^X$; with rate $u_t^Y$ under measure $\mathbb{P}^Y$. Under the conditions in Theorem 1, the KL divergence between $\mathbb{P}_t^X$ and $\mathbb{P}_t^Y$ is*

$$D_{KL}(\mathbb{P}_t^X||\mathbb{P}_t^Y) = \mathbb{E}^X\Big\{ \int_0^t \sum_{x \neq X(s)} D_F(u_s^X(x, X(s))||u_s^Y(x, X(s)))ds \Big\},$$

*where $D_F$ is the Bregman divergence with function $F(x) = x \log x$. Consequently, it holds that*

$$D_{KL}(p_t^X||p_t^Y) \leq \mathbb{E}^X\Big\{ \int_0^t \sum_{x \neq X(s)} D_F(u_s^X(x, X(s))||u_s^Y(x, X(s)))ds \Big\}.$$

Theorem 2 is *crucial* for developing the estimation error. Due to the sparsity of the transition rate matrix, intuitively, if $D_F(u_t^X(x, z)||u_t^Y(x, z)) \leq \epsilon$ uniformly for $t \in [0, 1 - \tau]$ and $(x, z)$ with $d^H(x, z) = 1$, then $D_{KL}(p_{1-\tau}^X||p_{1-\tau}^Y) \leq \mathcal{D}|\mathcal{S}|\epsilon$, which is linear in $\mathcal{D}$. This matches the results in discrete diffusion models (Chen & Ying, 2024; Zhang et al., 2025; Ren et al., 2025a) and continuous diffusion models (Benton et al., 2024a). In the next section, we will systematically perform an error analysis for our distribution estimation based on discrete flow models.

## 5 MAIN RESULTS

In this section, we establish the error bound for discrete flow-based models. We will first present the additional notation and assumptions. Then we will present the error bounds for transition rate estimation and early stopping, and discuss the implications of the results.

### 5.1 ADDITIONAL NOTATIONS AND ASSUMPTIONS

We first introduce some additional notation. Similar to the training objective (Equation 5), we denote the oracle transition rate as

$$u^0 = \arg\min_u \mathbb{E}\Big\{ \sum_{z \neq X(\mathbf{t})} \Big[ -u_\mathbf{t}(z, X(\mathbf{t})|X(1)) \log u_\mathbf{t}(z, X(\mathbf{t})) + u_\mathbf{t}(z, X(\mathbf{t})) \Big] \Big\} \triangleq \arg\min_u \mathcal{L}(u),$$

where $Z = (\mathbf{t}, X(\mathbf{t}), X(1))$ is a test point independent of the data $\mathbb{D}_n$. Through marginalization trick (Lipman et al., 2024), we have $u_t^0(z, x) = \mathbb{E}[u_\mathbf{t}(z, x|X(1))|X(\mathbf{t}) = x, \mathbf{t} = t]$. We also define the best approximation in the function class $\mathcal{G}_n$ as

$$u^* = \arg\min_{u \in \mathcal{G}_n} \mathbb{E}\Big\{ \sum_{z \neq X(\mathbf{t})} \Big[ -u_\mathbf{t}(z, X(\mathbf{t})|X(1)) \log u_\mathbf{t}(z, X(\mathbf{t})) + u_\mathbf{t}(z, X(\mathbf{t})) \Big] \Big\} \triangleq \arg\min_{u \in \mathcal{G}_n} \mathcal{L}(u).$$

Define

$$g(u, z, Z) \triangleq -u_\mathbf{t}(z, X(\mathbf{t})|X(1)) \log \frac{u_\mathbf{t}(z, X(\mathbf{t}))}{u_\mathbf{t}^0(z, X(\mathbf{t}))} + u_\mathbf{t}(z, X(\mathbf{t})) - u_\mathbf{t}^0(z, X(\mathbf{t})).$$

Then, the approximation error of $u^*$ is

$$\begin{aligned}
\mathcal{L}(u^*) - \mathcal{L}(u^0) &= \mathbb{E}\Big[ \sum_{z \neq X(\mathbf{t})} g(u^*, z, Z) \Big] \\
&= \mathbb{E}\Big\{ \sum_{z \neq X(\mathbf{t})} \Big[ -u_\mathbf{t}(z, X(\mathbf{t})|X(1)) \log \frac{u_\mathbf{t}^*(z, X(\mathbf{t}))}{u_\mathbf{t}^0(z, X(\mathbf{t}))} + u_\mathbf{t}^*(z, X(\mathbf{t})) - u_\mathbf{t}^0(z, X(\mathbf{t})) \Big] \Big\} \\
&= \mathbb{E}\Big[ \sum_{z \neq X(\mathbf{t})} D_F\Big(u_\mathbf{t}^0(z, X(\mathbf{t}))||u_\mathbf{t}^*(z, X(\mathbf{t}))\Big) \Big] \\
&= \inf_{u \in \mathcal{G}_n} \mathbb{E}\Big[ \sum_{z \neq X(\mathbf{t})} D_F\Big(u_\mathbf{t}^0(z, X(\mathbf{t}))||u_\mathbf{t}(z, X(\mathbf{t}))\Big) \Big],
\end{aligned}$$
(7)

where the third equation we use the marginalization trick.

**Assumption 1** (Boundedness). *The conditional transition rate is uniformly bounded from above: $u_t(z, x|x_1) \leq \overline{M}_c$ uniformly for any $t \in [0, 1 - \tau], z, x, x_1 \in \mathcal{S}^\mathcal{D}$, where $d^H(z, x) = 1$. Moreover, the oracle rate is uniformly bounded from below: $u_t^0(z, x) > \underline{M}_c$ uniformly for any $t \in [0, 1 - \tau], z, x \in \mathcal{S}^\mathcal{D}$, where $d^H(z, x) = 1$.*

**Assumption 2** (Function Class). *The functions in $\mathcal{G}_n$ (divided by the oracle transition rate) are uniformly bounded by $\underline{M}$ and $\overline{M}$ from below and above, respectively: $u_t(z,x)/u_t^0(z,x) \in [\underline{M}, \overline{M}]$ for any $t \in [0, 1-\tau]$, $(z,x) \in \{\mathcal{S}^{\mathcal{D}} \times \mathcal{S}^{\mathcal{D}} : d^H(z,x) = 1\}$ and $u \in \mathcal{G}_n$.*

**Remark 3.** *We only impose conditions on the state pairs with Hamming distance equal to one. Assumption 1 can be satisfied by choosing mixture path and uniform source distribution in Equation 3; see Section B.2 for further discussion. Assumption 2 provides the strong convexity of the Bregman divergence and is crucial to satisfying the condition in Theorem 1. The parameters in Assumption 2 can be fixed if the ranges of $u$ and $u^0$ are bounded by two positive constants from above and below; see Section B.3 for a specific example.*

## 5.2 ESTIMATION ERROR DECOMPOSITION

Suppose that $p_t$ (resp. $\hat{p}_t$) is the marginal distribution of a CTMC with rate $u_t^0$ (resp. $\hat{u}_t$) at time $t$, where $p_0 = \hat{p}_0$. For a random sample $\mathbb{D}_n$, according to Theorem 2 in the previous section, we have

$$\mathbb{E}_{\mathbb{D}_n}[D_{KL}(p_{1-\tau}||\hat{p}_{1-\tau})] \leq (1-\tau)\mathbb{E}_{\mathbb{D}_n}\mathbb{E}_Z\Big\{ \sum_{z \neq X(\mathbf{t})} g(\hat{u}, z, Z) \Big\} = \mathbb{E}_{\mathbb{D}_n}[\mathcal{L}(\hat{u}) - \mathcal{L}(u^0)].$$

The following proposition shows that this error bound can be decomposed to the stochastic error and the approximation error.

**Proposition 4** (Estimation Error Decomposition). *For random sample $\mathbb{D}_n$, the excess risk of the estimator $\hat{u}$ through ERM (Equation 5) satisfies*

$$\mathbb{E}_{\mathbb{D}_n}[\mathcal{L}(\hat{u}) - \mathcal{L}(u^0)] \leq \underbrace{\mathbb{E}_{\mathbb{D}_n}[\mathcal{L}(\hat{u}) + \mathcal{L}(u^0) - 2\mathcal{L}_n(\hat{u})]}_{\text{Stochastic Error}} + 2 \underbrace{\inf_{u \in \mathcal{G}_n} \mathbb{E}\Big[ \sum_{z \neq X(\mathbf{t})} D_F(u_{\mathbf{t}}(z, X(\mathbf{t})||u_{\mathbf{t}}^0(z, X(\mathbf{t}))) \Big]}_{\text{Approximation Error}}.$$

## 5.3 STOCHASTIC ERROR BOUND

Now, we establish the upper bound of the stochastic error in Proposition 4 using the empirical process theory. Before presenting our stochastic error bound, we first introduce the definition of uniform covering number.

**Definition 3** (Uniform Covering Number, Jiao et al. (2023)). *Let $S$ be a subset of $\mathbb{R}^n$. Given a positive real number $\epsilon$, a subset $\mathcal{C}$ of $S$ is called an $\epsilon$-covering of $S$ w.r.t. the infinity norm if for any $x \in S$, there is $z \in \mathcal{C}$ such that $\|z - x\|_{L^\infty} < \epsilon$. The minimal cardinality $\mathcal{N}_n(\epsilon, S, L^\infty)$ of all possible $\mathcal{C}$ is called the covering number of $S$. For a given sequence $\mathbf{s} = \{(t_i, z_i, x_i)\}_{i=1}^n$, let $\mathcal{F}|_{\mathbf{s}} = \Big\{ \Big( u_{t_1}(z_1, x_1), u_{t_2}(z_2, x_2), \ldots, u_{t_n}(z_n, x_n) \Big) : u \in \mathcal{F} \Big\}$. We define the covering number of $\mathcal{F}$ constrained on $\mathbf{s}$ as $\mathcal{N}_n(\epsilon, \mathcal{F}|_{\mathbf{s}}, L^\infty)$. The uniform covering number is defined as*

$$\mathcal{N}_n(\epsilon, \mathcal{F}, L^\infty) = \max \{\mathcal{N}_n(\epsilon, \mathcal{F}|_{\mathbf{s}}, L^\infty) : \mathbf{s} \in ([0, 1-\tau] \times \mathcal{S}^{\mathcal{D}} \times \mathcal{S}^{\mathcal{D}})^n\}.$$

**Theorem 3** (Stochastic Error). *Assume that Assumption 1 and Assumption 2 hold. Let $\mathbb{P}_n$ be the empirical measure w.r.t. $\mathbb{D}_n$. If $n \geq CK_1^2|\mathcal{S}|^2\mathcal{D}[\log \mathcal{N}_{|\mathcal{S}|n}(1/(2n), \mathcal{G}_n, L^\infty) + \log \mathcal{D}]$, then the stochastic error in Proposition 4 has the following upper bound:*

$$\mathbb{E}_{\mathbb{D}_n}[\mathcal{L}(\hat{u}) + \mathcal{L}(u^0) - 2\mathcal{L}_n(\hat{u})] \leq \frac{CK_1^2|\mathcal{S}|^2\mathcal{D}[\log \mathcal{N}_{|\mathcal{S}|n}(1/(2n), \mathcal{G}_n, L^\infty) + \log \mathcal{D}]}{n} \triangleq \gamma_n(\mathcal{D}, |\mathcal{S}|, \mathcal{G}_n),$$

*where $\mathcal{N}_{|\mathcal{S}|n}(1/(2n), \mathcal{G}_n, L^\infty)$ is the uniform covering number of $\mathcal{G}_n$, and $K_1 = \frac{2\overline{M}_c^2(\overline{M}+1)}{\min(1,\underline{M})\underline{M}_c}$.*

Theorem 3 shows that the stochastic error has the convergence rate

$$O(n^{-1}K_1^2|\mathcal{S}|^2\mathcal{D}[\log \mathcal{N}(1/(2n), \mathcal{G}_n, L^\infty(\mathbb{P}_n)) + \log \mathcal{D}]).$$

This convergence rate is nearly linear in $\mathcal{D}$ (up to some logarithmic multiplier), which nearly aligns with the stochastic error bound of continuous flow matching (Gao et al., 2024b). We will further discuss this stochastic error bound in Section 5.5. By Pinsker's inequality and Jensen's inequality, we have the following upper bound for the total variation between $p_{1-\tau}$ and $\hat{p}_{1-\tau}$.

$$\mathbb{E}_{\mathbb{D}_n}\Big[\text{TV}(p_{1-\tau}, \hat{p}_{1-\tau})\Big] \leq \sqrt{\frac{1}{2}\mathbb{E}_{\mathbb{D}_n}\Big[D_{\text{KL}}(p_{1-\tau}||\hat{p}_{1-\tau})\Big]}$$

$$\leq \sqrt{\frac{1}{2}\gamma_n(\mathcal{D}, |\mathcal{S}|, \mathcal{G}_n)} + \sqrt{\inf_{u \in \mathcal{G}_n} \mathbb{E}\Big[D_F\Big( \sum_{z \neq X(\mathbf{t})} u_{\mathbf{t}}^0(z, X(\mathbf{t}))|| \sum_{z \neq X(\mathbf{t})} u_{\mathbf{t}}(z, X(\mathbf{t})) \Big)\Big]}.$$

### 5.4 EARLY STOPPING ERROR BOUND

We present the early stopping error bound for the total variation between $p_1$ and $p_{1-\tau}$ in the following theorem.

**Theorem 4** (Early Stopping Error). *Consider the conditional probability path in Equation 3. Suppose that the source distribution is uniform. Then the total variation between $p_1$ and $p_{1-\tau}$ has the following error bound*

$$TV(p_1, p_{1-\tau}) \leq 1 - \exp\left\{ \mathcal{D} \log\left( -(1 - \kappa_{1-\tau})\frac{|\mathcal{S}| - 1}{|\mathcal{S}|} + 1 \right) \right\} \triangleq \varrho(\mathcal{D}, |\mathcal{S}|, \tau).$$

If the time schedule we used is the linear schedule $\kappa_t = t$, then the early stopping error of the discrete flow in Theorem 4 has the same convergence rate as that of the discrete diffusion derived in Theorem 1 of Zhang et al. (2025) as $\tau \to 0^+$.

### 5.5 OVERALL DISTRIBUTION ERROR

Using triangle inequality, the total variation between $p_1$ and $\hat{p}_{1-\tau}$ can be bounded by the early stopping error and the estimation error. Combining the results of Proposition 4, Theorem 3 and Theorem 4, the overall bound for our distribution estimation is

$$
\begin{aligned}
&\mathbb{E}_{\mathbb{D}_n}\left[ \mathrm{TV}(p_1, \hat{p}_{1-\tau}) \right] \\
&\leq \mathbb{E}_{\mathbb{D}_n}\left[ \mathrm{TV}(p_{1-\tau}, \hat{p}_{1-\tau}) \right] + \mathrm{TV}(p_1, p_{1-\tau}) \\
&\leq \underbrace{\sqrt{\frac{1}{2}\gamma_n(\mathcal{D}, |\mathcal{S}|, \mathcal{G}_n)}}_{\text{Stochastic Error}} + \sqrt{\underbrace{\inf_{u \in \mathcal{G}_n} \mathbb{E}\left[ D_F\left( \sum_{z \neq X(\mathbf{t})} u_{\mathbf{t}}^0(z, X(\mathbf{t})) \| \sum_{z \neq X(\mathbf{t})} u_{\mathbf{t}}(z, X(\mathbf{t})) \right) \right]}_{\text{Approximation Error}}} + \underbrace{\varrho(\mathcal{D}, |\mathcal{S}|, \tau)}_{\text{Early Stopping Error}},
\end{aligned}
\tag{8}
$$

where $\gamma_n(\mathcal{D}, |\mathcal{S}|, \mathcal{G}_n)$ and $\varrho(\mathcal{D}, |\mathcal{S}|, \tau)$ are defined in Theorem 3 and Theorem 4, respectively.

Given the time schedule $\kappa_t$, the sequence length $\mathcal{D}$ and the vocabulary size $|\mathcal{S}|$, there are two quantities related to the above error bound. As $\tau \to 0^+$, the stochastic error increases and the early stopping error decreases; as $\mathcal{N}_{|\mathcal{S}|n}(1/(2n), \mathcal{G}_n, L^\infty)$ increases, the stochastic error increases and the approximation error decreases. We can choose a suitable early stopping parameter $\tau$ to balance the stochastic error and the early stopping error. We defer the error analysis for the specific neural network class with the ReLU activation function to Section B.3.

**Balancing the Stochastic Error and Early Stopping Error.** Given $\overline{M}, \underline{M}$ in Assumption 2 and the vocabulary size $|\mathcal{S}|$, we analyze the convergence rate in Equation 8. Consider the linear schedule $\kappa_t = t$. Suppose that $\mathcal{D} \leq \mathcal{N}_{|\mathcal{S}|n}(1/(2n), \mathcal{G}_n, L^\infty)$ and the approximation error goes to zero when the size of $\mathcal{G}_n$ goes to infinity. We mainly focus on the choice of the early stopping parameter $\tau$ to balance the stochastic error and the early stopping error. Note that if $\tau\mathcal{D} \to 0$, the early stopping error has the convergence rate $O(\tau\mathcal{D})$. By taking $\overline{M}_c, \underline{M}_c$ of Assumption 1 into consideration ($K_1^2$ has the convergence rate $\tau^{-4}$ by Section B.2), the stochastic error has the convergence rate $O(n^{-1}\tau^{-4}\mathcal{D} \log \mathcal{N}_{|\mathcal{S}|n}(1/(2n), \mathcal{G}_n, L^\infty))$. Choosing $\tau = [n^{-1}\mathcal{D}^{-1} \log \mathcal{N}_{|\mathcal{S}|n}(1/(2n), \mathcal{G}_n, L^\infty)]^{1/6}$, if $n \geq c\mathcal{D}^5 \log \mathcal{N}_{|\mathcal{S}|n}(1/(2n), \mathcal{G}_n, L^\infty)$, then the summation of the early stopping error and the squared root of the stochastic error has the convergence rate $O([n^{-1}\mathcal{D}^5 \log \mathcal{N}_{|\mathcal{S}|n}(1/(2n), \mathcal{G}_n, L^\infty)]^{1/6})$.

## 6 SIMULATION

To study the empirical performance of discrete flow-based models with different dimensions, sample sizes, and hyperparameters, we conduct several simulation experiments, demonstrating the consistency of our theoretical analysis and empirical results. The data distribution and implementation details are described in Section F.1. Fig. 1, Fig. 2 and Fig. 3 present the simulation results for the empirical version of estimation error (up to an additive constant) with different sample size $n$, dimension $\mathcal{D}$ and early stopping parameter $\tau$. We can see that the estimation error is nearly linear in $\mathcal{D}$, and the estimation error decreases as $n$ and $\tau$ increase. In addition, we also calculate the total variation (of the empirical joint distribution on the first 3 dimensions, $8^3 = 512$ states in total) with different dimension $\mathcal{D}$ and $\tau$ (see Section F.2 for details). As presented in Fig. 4, the total variation decreases first and then increases as $\tau$ increases, with the minimum achieved around $\tau = 0.03$ or $0.05$.

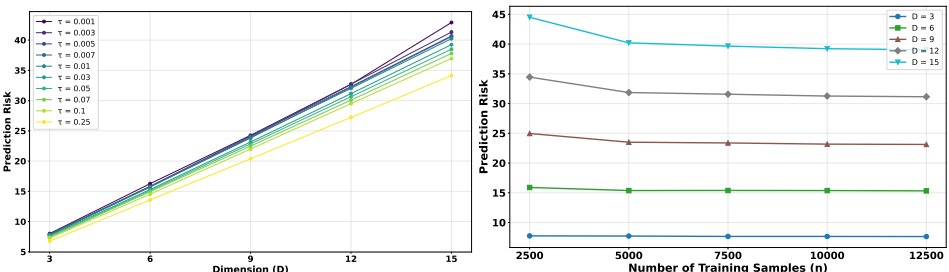

Figure 1: Prediction risk v.s. dimension.    Figure 2: Prediction risk v.s. sample size.

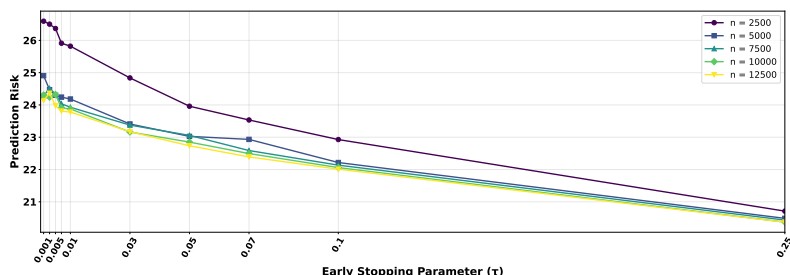

Figure 3: Prediction risk v.s. early stopping parameter.

# 7    CONCLUSION AND FUTURE WORKS

In this article, we have established the first non-asymptotic error bounds for the total variation between the data distribution and the distribution generated by the learned transition rate in discrete flow models. To derive the KL divergence between two CTMCs, we develop a Girsanov-type theorem based on stochastic calculus theory. Building on this result, the estimation error can be decomposed into stochastic error and approximation error. We control the stochastic error by using empirical process theory and balance the stochastic error and the early stopping error by carefully analyzing the boundedness condition. Our theoretical framework serves as an essential theoretical guarantee for studies grounded in CTMCs.

There are two directions deserving future research. Firstly, the error bound we derived has zero discretization error via uniformization technique; while, in practice, the uniformization algorithm does not enjoy sampling efficiency. It would be interesting to investigate the theoretical results of accuracy-efficiency trade-offs for discrete flow models. Secondly, the current work analyzes the approximation error for ReLU networks only; deriving the approximation error for networks with self-attention layers is a challenging problem in approximation theory, requiring greater effort and more careful analysis.

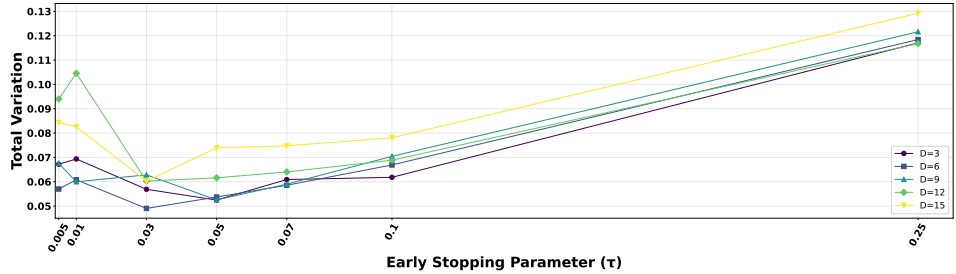

Figure 4: Total variation (of the empirical joint distribution on the first 3 dimensions) v.s. early stopping parameter with uniformization algorithm.

## ETHICS STATEMENT

This work adheres to the ICLR Code of Ethics. Our research mainly focuses on the theoretical understanding of the error of the discrete flow model and does not involve human subjects or sensitive personal data.

## REPRODUCIBILITY STATEMENT

The main theoretical contributions and their assumptions are summarized in Section 4, Section 5 and Section B. The complete proof can be found in Section C, Section D, and Section E.

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

APPENDIX

The appendix is organized as follows. Section A gives the uniformization algorithm. Section B offers some discussions on time singularity, assumptions and approximation error analysis. Section C presents some useful lemmas. Section D gives the proof of CTMC theory. Section 5 provides the proof of main results. Section F presents additional simulation experiments and implementation details.

## A  UNIFORMIZATION ALGORITHM

---

**Algorithm 1** Sampling via Uniformization

---

**Require:** A learned transition rate $\hat{u}$, an early stopping parameter $\tau > 0$, time partition $0 = t_0 < t_1 < \cdots < t_N = 1 - \tau$, parameters $\lambda_1, \lambda_2, \ldots, \lambda_N$ satisfying $\sup_{t \in [t_k, t_{k+1}]} \sum_{z \neq x} \hat{u}_t(z, x) \leq \lambda_{k+1}$ for any $k \in \{0, 1, \ldots, N-1\}, x \in \mathcal{S}^{\mathcal{D}}$.

1: Draw $Y_0 \sim \mathcal{U}(\mathcal{S}^{\mathcal{D}})$.
2: **for** $k = 0$ to $N - 1$ **do**
3:     Draw $M \sim \text{Poisson}(\lambda_{k+1}(t_{k+1} - t_k))$.
4:     Sample $M$ points i.i.d. from $\mathcal{U}([t_k, t_{k+1}])$ and sort them as $\tau_1 < \tau_2 < \cdots < \tau_M$.
5:     Set $Z_0 = Y_k$.
6:     **for** $j = 0$ to $M - 1$ **do**
7:         Set $Z_{j+1} = \begin{cases} z, & \text{with probability } \hat{u}_{\tau_j}(z, Z_j)/\lambda_{k+1} \\ Z_j, & \text{with probability } \sum_{z \neq Z_j} \hat{u}_{\tau_j}(z, Z_j)/\lambda_{k+1} \end{cases}$, where $z \neq Z_j$.
8:     **end for**
9:     Set $Y_{k+1} = Z_M$.
10: **end for**
11: **return** $Y_N \sim \hat{p}_{1-\tau}$

---

## B  DISCUSSIONS

### B.1  DISCUSSION ON TIME SINGULARITY

We consider one token case and linear time schedule $\kappa_t = t$ for simplicity. Note that the conditional transition rate is $u_t(z, x|x_1) = \frac{\dot{\kappa}_t}{1-\kappa_t}(\delta_{x_1}(z) - \delta_x(z))$, which can generate the conditional probability path $p_{t|1}(x|x_1) = \kappa_t \delta_{x_1}(x) + (1 - \kappa_t)p_0(x)$, where $\kappa_t \to 1$ as $t \to 1$. Then, when $p_1$ has full support, the marginal transition rate has a bounded limit as $t \to 1$:

$$
\begin{aligned}
u_t(z, x) &= \frac{\dot{\kappa}_t}{1 - \kappa_t}(p_{1|t}(z|x) - \delta_x(z)) \\
&= \frac{\dot{\kappa}_t}{1 - \kappa_t}\left( \frac{p_1(z)p_{t|1}(x|z)}{\sum_z p_1(z)p_{t|1}(x|z)} - \delta_x(z) \right) \\
&= \frac{\dot{\kappa}_t p_0(x)(p_1(z) - \delta_x(z))}{\kappa_t p_1(x) + (1 - \kappa_t)p_0(x)} \\
&\to \frac{\dot{\kappa}_1 p_0(x)(p_1(z) - \delta_x(z))}{p_1(x)}.
\end{aligned}
$$

When $p_1$ is not fully supported, for $x \in \{x \in \mathcal{S}^{\mathcal{D}} : p_1(x) = 0\}$, the above limit goes to infinity if $p_1(z) \in (0, 1)$ since $u_t(z, x) = \frac{\dot{\kappa}_t(p_1(z) - \delta_x(z))}{1 - \kappa_t}$ in this case. This result is analogous to the continuous flow matching counterparts (see Proposition C.4 in Wan et al., 2025).

For the transition rate estimator, however, since the conditional transition rate explodes as $t \to 1$, it is hard to control the estimation error of $\hat{u}$ (see the discussion in Section B.2). Therefore, it is necessary to use the early stopping technique to balance the errors arising from estimation and early stopping.

## B.2 DISCUSSION ON ASSUMPTION 1

In this subsection, we discuss Assumption 1 in some specific scenario, which also provides a reference for time schedule selection.

First, we discuss the upper bound $\overline{M}_c$ in Assumption 1.

1. Consider the polynomial schedule $\kappa_t = t^s$ in Equation 3, where $s \geq 1$. Then, by mean value theorem, for any $z^d \neq x^d$, if $t \in (0, 1)$, we have

$$u_t^d(z^d, x^d | x_1^d) \leq \frac{\dot{\kappa}_t}{1 - \kappa_t} = \frac{st^{s-1}}{s(t + \theta(1-t))^{s-1}(1-t)} \leq \frac{1}{1-t},$$

which means that the conditional transition rate has the uniform upper bound $\overline{M}_c = \frac{1}{\tau}$.

2. Consider the cosine schedule $\kappa_t = \cos^2(\frac{\pi}{2}(1-t))$ in Equation 3, which is kinetic optimal as mentioned in Shaul et al. (2025). Then, by $\tan(a) > a$ ($a \in (0, \pi/2)$), for any $z^d \neq x^d$, if $t \in (0, 1)$, we have

$$u_t^d(z^d, x^d | x_1^d) \leq \frac{\dot{\kappa}_t}{1 - \kappa_t} = \frac{\pi \cos(\frac{\pi}{2}(1-t)) \sin(\frac{\pi}{2}(1-t))}{\sin^2(\frac{\pi}{2}(1-t))} = \frac{\pi}{\tan(\frac{\pi}{2}(1-t))} \leq \frac{2}{1-t},$$

which means that the conditional transition rate has the uniform upper bound $\overline{M}_c = \frac{2}{\tau}$.

Given other quantities, if $\tau \to 0$, then the estimation error will go to infinity as presented in Theorem 3, since $K_1$ will go to infinity as $\overline{M}_c \to +\infty$. By the proof of Theorem 3, under the above two scenarios, the factor $K_1^2$ in the error bound of Theorem 3 grows at the order of $O(\tau^{-4})$.

Next, we discuss the lower bound $\underline{M}_c$ for the oracle transition rate $u_t^0(z, x)$, where $d^H(z, x) = 1$. We define the mixing coefficient $\alpha \in (0, 1)$ such that $\alpha \leq \frac{p_1(x_1^{\backslash d} | x_1^d)}{p_1(x_1^{\backslash d})} \leq \alpha^{-1}$ for any $x_1 \in \mathcal{S}^{\mathcal{D}}$ and $d \in [\mathcal{D}]$ (for example, if $p_1(x_1) = \prod_{d=1}^{\mathcal{D}} p_1(x_1^d)$, then $\alpha = 1$). We denote $\beta = \sup_{d, z^d, x^d}(p_1^d(z^d)/p_1^d(x^d))$, where $p_1^d$ is the marginal distribution of $p_1$ (for example, if $p_1^d$ is of uniform distribution, then $\beta = 1$). Suppose that the source distribution is uniform; that is, $p_{t|1}^d(x^d | x_1^d) = \frac{1 - \kappa_t}{|\mathcal{S}|} + \kappa_t \delta_{x_1^d}(x^d)$. Then, for any $d \in [\mathcal{D}]$ and $(z, x) \in \mathcal{S}^{\mathcal{D}} \times \mathcal{S}^{\mathcal{D}}$ such that $z^d \neq x^d, z^{\backslash d} = x^{\backslash d}$, we have

$$u_t^0(z, x) = \sum_{x_1} u_t^d(z^d, x^d | x_1^d) p_{1|t}(x_1 | x)$$

$$= \frac{\sum_{x_1} u_t^d(z^d, x^d | x_1^d) p_{t|1}^d(x^d | x_1^d) \prod_{i \neq d} p_{t|1}^i(x^i | x_1^i) p_1(x_1)}{\sum_{x_1} p_{t|1}^d(x^d | x_1^d) \prod_{i \neq d} p_{t|1}^i(x^i | x_1^i) p_1(x_1)}$$

$$= \frac{\frac{\dot{\kappa}_t}{|\mathcal{S}|} p_1^d(z^d) \sum_{x_1^{\backslash d}} \prod_{i \neq d} p_{t|1}^i(x^i | x_1^i) p_1(x_1^{\backslash d} | X(1)^d = z^d)}{\frac{1 - \kappa_t}{|\mathcal{S}|} \sum_{x_1^{\backslash d}} \prod_{i \neq d} p_{t|1}^i(x^i | x_1^i) p_1(x_1^{\backslash d}) + \kappa_t p_1^d(x^d) \sum_{x_1^{\backslash d}} \prod_{i \neq d} p_{t|1}^i(x^i | x_1^i) p_1(x_1^{\backslash d} | X(1)^d = x^d)}$$

$$\geq \frac{|\mathcal{S}|^{-1} \dot{\kappa}_t \alpha}{(1 - \kappa_t)\beta + \kappa_t \beta/\alpha}$$

$$\geq |\mathcal{S}|^{-1} \dot{\kappa}_t \alpha^2/\beta.$$

To give a specific example, we consider an AR($k$) autoregressive structure for the sequence; that is, $p_1(x_1^d | x_1^{<d}) = p_1(x_1^d | x_1^{((l-k)\vee 1):(d-1)})$ for $d > 2$. Then we have

$$\frac{p_1(x_1^{\backslash d} | x_1^d)}{p_1(x_1^{\backslash d})}$$

$$= \frac{p_1(x_1^1, \ldots, x_1^{d-1}) \prod_{l=d}^{\mathcal{D}} p_1(x_1^l | x_1^{((l-k)\vee 1):(l-1)})}{p_1(x^1, \ldots, x^{d-1}) p_1(x_1^d) \prod_{j=1}^{k} p(x_1^{d+j} | x_1^{(((d-k)\vee 1):(d+j-1))\backslash d}) \prod_{l=d+k+1}^{\mathcal{D}} p_1(x_1^l | x_1^{((l-k)\vee 1):(l-1)})}$$

$$= \frac{p_1(x_1^d | x_1^{((d-k)\vee 1):(d-1)}) \prod_{j=1}^{k} p_1(x_1^{d+j} | x_1^{((d+j-k)\vee 1):(d+j-1)})}{p_1(x_1^d) \prod_{j=1}^{k} p(x_1^{d+j} | x_1^{(((d-k)\vee 1):(d+j-1))\backslash d})}.$$

If $\log p_1(x_1^d)$ and $\log p_1(x_1^d | x_1^{((d-2k)\vee 1):(d-1)}) \in [-c, 0]$ for each $d \in [\mathcal{D}], x_1 \in \mathcal{S}^{\mathcal{D}}$, then the right-hand side of the above equation satisfying $\alpha < \frac{p_1(x_1^{\backslash d} | x_1^d)}{p_1(x_1^{\backslash d})} < \alpha^{-1}$ with $\alpha = \exp(-(k+1)c)$ and $\beta = \exp(-c)$, which are independent of $\mathcal{D}$.

Therefore, if $\alpha$ and $\beta$ are positive constants and the vocabulary size $|\mathcal{S}|$ is fixed, then $\underline{M}_c = (|\mathcal{S}|^{-1}\alpha^2/\beta) \inf_{t\in[0,1-\tau]} \dot{\kappa}_t$ is the uniform lower bound for the oracle transition rate $u_t^0(z,x)$, where $\mathrm{d}^H(z,x) = 1$. Empirically, one may adopt the linear schedule $\kappa_t = t$, which is commonly used in practice. More generally, a composite schedule can be employed: $\kappa_t = \frac{t+\kappa_t^0}{2}$, where $\kappa_t^0$ is a schedule that is a non-decreasing function of $t$.

## B.3 Error Analysis with ReLU Networks

In this subsection, we write $u_t(z,x)$ to $u(t,z,x)$ for any transition rate $u$. We consider using neural network (NN) functions with ReLU activation function to approximate the oracle transition rate $u^0$. To impose some regularity condition on the oracle rate, we first introduce the Hölder class. For a finite constant $B_0 > 0$ and input dimension $d \in \mathbb{N}^+$, the Hölder class of functions $\mathcal{H}^\beta([0,1]^d, B_0)$ is defined as

$$\mathcal{H}^\beta([0,1]^d, B_0) = \left\{ f : [0,1]^d \to \mathbb{R}, \max_{\|\alpha\|_1 \le s} \|\partial^\alpha f\|_\infty \le B_0, \max_{\|\alpha\|_1 = s} \sup_{x \ne y} \frac{|\partial^\alpha f(x) - \partial^\alpha f(y)|}{\|x-y\|_2^r} \le B_0 \right\},$$

where $\beta = r + s, s = \lfloor \beta \rfloor \in \mathbb{N}, \partial^\alpha = \partial^{\alpha_1} \cdots \partial^{\alpha_d}$ with $\alpha = (\alpha_1, \dots, \alpha_d)^\top \in \mathbb{N}^d$ and $\|\alpha\|_1 = \sum_{i=1}^d \alpha_i$.

**Assumption 3** (Hölder smoothness). *There exists a function $\bar{u}^0$ belongs to the Hölder class $\mathcal{H}^\beta([0,1] \times [0,|\mathcal{S}|]^{2\mathcal{D}}, B_0)$ for a given $\beta > 0$ and a finite constant $B_0 > \underline{M}_c$ such that $u^0 = \bar{u}^0$ for any $t \in [0,1]$ and $x \in \mathcal{S}^{\mathcal{D}} \times \mathcal{S}^{\mathcal{D}}$ with $\mathrm{d}^H(x_{1:\mathcal{D}}, x_{(\mathcal{D}+1):(2\mathcal{D})}) = 1$.*

Without loss of generality, we rewrite $\bar{u}^0$ as $u^0$ through this analysis. This assumption requires that the oracle transition rate $u^0$ does not have terminal time singularity at time $t = 1$; e.g., $p_1$ has full support as mentioned in Section B.1.

### B.3.1 Bounding Approximation Error

Following the proof of Lemma B.6 in Xu et al. (2025), we prove that there exists a sequence of neural networks with the ReLU activation function that can control the approximation error. We focus on the region $(t, x) \in [0,1] \times [0, |\mathcal{S}|]^{2\mathcal{D}}$. Let $\tilde{u}^0(t, x') = u^0(t, |\mathcal{S}|x')$ for $x' \in [0,1]^{2\mathcal{D}}$. Then $\tilde{u}^0 \in \mathcal{H}^\beta([0,1]^{2\mathcal{D}+1}, |\mathcal{S}|^\beta B_0)$. Lemma 5 implies that for any $S_1, S_2 \in \mathbb{N}^+$, there exists a function $f^*$ implemented by a ReLU network with depth $\mathrm{D} = 21(\lfloor \beta \rfloor + 1)^2 S_1 \lceil \log_2(8S_1) \rceil$ and width $\mathrm{W} = 38(\lfloor \beta \rfloor + 1)^2 (2\mathcal{D} + 1)^{\lfloor \beta \rfloor + 1} S_2 \lceil \log_2(8S_2) \rceil$ such that

$$|f^*(t, x') - \tilde{u}^0(t, x')| \le 18|\mathcal{S}|^\beta B_0(\lfloor \beta \rfloor + 1)^2 (2\mathcal{D} + 1)^{\lfloor \beta \rfloor + (\beta \vee 1)/2} (S_1 S_2)^{-2\beta/(2\mathcal{D}+1)},$$

for any $(t, x') \in [0,1]^{2\mathcal{D}+1} \backslash \Omega([0,1]^{2\mathcal{D}+1}, K, \delta)$, where $K = \lceil (S_1 S_2)^{2/(2\mathcal{D}+1)} \rceil$ and $\delta$ is an arbitrary number in $(0, 1/(3K))$. Let $f^{**}(t, x) = f^*(t, x/|\mathcal{S}|)$ for $x \in [0, |\mathcal{S}|]^{2\mathcal{D}}$ be a network with depth $\mathrm{D} + 1$. Note that a clip function can be expressed as a two-layer ReLU network, then we define the NN $f^{***}(t, x) = \underline{M}_c + \mathrm{ReLU}(B_0 - \mathrm{ReLU}(B_0 - f^{**}(t, x)) - \underline{M}_c)$ with depth $\mathrm{D} + 3$, whose range is $[\underline{M}_c, B_0]$.

Consequently, we have the following bound:

$$|f^{***}(t, x) - u^0(t, x)| \le 18|\mathcal{S}|^\beta B_0(\lfloor \beta \rfloor + 1)^2 (2\mathcal{D} + 1)^{\lfloor \beta \rfloor + (\beta \vee 1)/2} (S_1 S_2)^{-2\beta/(2\mathcal{D}+1)},$$

where $(t, x) \in [0,1] \times [|\mathcal{S}|]^{2\mathcal{D}}$ such that $(t, x/|\mathcal{S}|) \in [0,1] \times [|\mathcal{S}|]^{2\mathcal{D}} \backslash \Omega([0,1]^{2\mathcal{D}+1}, K, \delta)$ and

$$\Omega([0,1]^{2\mathcal{D}+1}, K, \delta) = \cup_{i=1}^{2\mathcal{D}+1} \{x = [x_1, x_2, \dots, x_{2\mathcal{D}+1}]^\top : x_i \in \cup_{k=1}^{K-1}(k/K - \delta, k/K)\}.$$

Note that $\mathbf{t} \sim \mathcal{U}([0, 1 - \tau])$, whose probability measure is absolutely continuous w.r.t. the Lebesgue measure. Moreover, when $\delta$ is sufficiently small, the set $\{1/|\mathcal{S}|, 2/|\mathcal{S}|, \ldots, 1\}$ and the set $\cup_{k=1}^{K-1}(k/K - \delta, k/K)$ are disjoint. Then, the event

$$\{(\mathbf{t}, z/|\mathcal{S}|, x/|\mathcal{S}|) \in [0, 1]^{2\mathcal{D}+1}\backslash\Omega([0, 1]^{2\mathcal{D}+1}, K, \delta) \text{ for any } (z, x) \in \mathcal{S}^{\mathcal{D}} \times \mathcal{S}^{\mathcal{D}}\}$$

holds with probability greater than $1 - 2K\delta$. Thus, by the smoothness of the Bregman divergence, following the proof of Theorem 4.2 in Jiao et al. (2023), since $\delta$ is an arbitrary number in $(0, 1/(3K))$, the approximation error has the following bound

$$\inf_{u \in \mathcal{G}_n} \mathbb{E}\Big[D_F\Big(\sum_{z \neq X(\mathbf{t})} u^0(\mathbf{t}, z, X(\mathbf{t})) \| \sum_{z \neq X(\mathbf{t})} u(\mathbf{t}, z, X(\mathbf{t}))\Big)\Big]$$

$$\leq \frac{1}{2\underline{M}_c}\mathbb{E}\Big|\sum_{z \neq X(\mathbf{t})} f^{***}(\mathbf{t}, z, X(\mathbf{t})) - \sum_{z \neq X(\mathbf{t})} u^0(\mathbf{t}, z, X(\mathbf{t}))\Big|^2 \tag{9}$$

$$\leq C\underline{M}_c^{-1}|\mathcal{S}|^{2\lfloor\beta\rfloor}B_0^2(\lfloor\beta\rfloor + 1)^4(2\mathcal{D} + 1)^{2\lfloor\beta\rfloor+\beta\vee 1}(S_1 S_2)^{-4\beta/(2\mathcal{D}+1)},$$

where $\mathcal{G}_n$ is the class of ReLU networks with range $[\underline{M}_c, B_0]$, depth $\mathrm{D}^* = 21(\lfloor\beta\rfloor + 1)^2 S_1 \lceil\log_2(8S_1)\rceil + 3$, width $\mathrm{W}^* = 38(\lfloor\beta\rfloor + 1)^2(2\mathcal{D} + 1)^{\lfloor\beta\rfloor+1}S_2\lceil\log_2(8S_2)\rceil$, $S_1, S_2 \in \mathbb{N}^+$. If $\underline{M}_c$ is fixed, then the parameters in Assumption 2 are fixed constants.

### B.3.2 CHOOSING HYPERPARAMETERS

Before the end of this section, the convergence rate might omit a logarithmic multiplier of $n\tau^4$. Similar to Section 5.5, given the vocabulary size $|\mathcal{S}|$, by Lemma 6 and Lemma 7, if $n \geq C\tau^{-4}|\mathcal{S}|^2\mathcal{D}\mathrm{S}^*\mathrm{D}^* \log \mathrm{S}^* \log n$, the stochastic error is

$$\gamma_n(\mathcal{D}, |\mathcal{S}|, \mathcal{G}_n) = O\Big(\frac{|\mathcal{S}|^2\mathcal{D}\mathrm{S}^*\mathrm{D}^* \log \mathrm{S}^* \log n}{n\tau^4}\Big),$$

where $\mathrm{S}^*$ is the number of parameters of the ReLU networks in $\mathcal{G}_n$. Note that for a ReLU network with depth $\mathrm{D}^*$ and width $\mathrm{W}^*$ and input dimension $2\mathcal{D} + 1$, we have (assume that $\mathcal{D} \lesssim \mathrm{W}^*\mathrm{D}^*$)

$$\mathrm{S} \leq \underbrace{\mathrm{W}^*(2\mathcal{D} + 1) + \mathrm{W}^*}_{\text{input layer}} + \underbrace{((\mathrm{W}^*)^2 + \mathrm{W}^*)(\mathrm{D}^* - 1)}_{\text{hidden layer}} + \underbrace{\mathrm{W}^* + 1}_{\text{output layer}} = O((\mathrm{W}^*)^2\mathrm{D}^*).$$

Therefore, by choosing $S_1 = O((n\tau^4)^{\frac{2\mathcal{D}+1}{(4\mathcal{D}+4\beta+2)}})$ and $S_2 = O(1)$, we have

$$\mathrm{W}^* = O\Big((2\mathcal{D} + 1)^{\lfloor\beta\rfloor+1}\Big); \ \mathrm{D}^* = O\Big((n\tau^4)^{\frac{2\mathcal{D}+1}{(4\mathcal{D}+4\beta+2)}} \log(n\tau^4)\Big);$$

$$\mathrm{S}^* = O\Big(\mathcal{D}^{2\lfloor\beta\rfloor+2}n^{\frac{2\mathcal{D}+1}{(4\mathcal{D}+4\beta+2)}} \log(n\tau^4)\Big),$$

yielding that

$$\gamma_n(\mathcal{D}, |\mathcal{S}|, \mathcal{G}_n) = O\Big(|\mathcal{S}|^2\mathcal{D}^{2\lfloor\beta\rfloor+2}(\log(n\tau^4))^4(n\tau^4)^{-\frac{2\beta}{(2\mathcal{D}+2\beta+1)}}\Big).$$

By Equation 9, the approximation error is

$$\inf_{u \in \mathcal{G}_n} \mathbb{E}\Big[D_F\Big(\sum_{z \neq X(\mathbf{t})} u^0(\mathbf{t}, z, X(\mathbf{t})) \| \sum_{z \neq X(\mathbf{t})} u(\mathbf{t}, z, X(\mathbf{t}))\Big)\Big]$$

$$= O\Big(|\mathcal{S}|^{2\beta}(2\mathcal{D} + 1)^{2\lfloor\beta\rfloor+\beta\vee 1}(n\tau^4)^{-\frac{2\beta}{(2\mathcal{D}+2\beta+1)}}\Big).$$

If the vocabulary size $|\mathcal{S}|$ is fixed, then the summation of the approximation error and the stochastic error has the convergence rate (up to some logarithmic multiplier)

$$\mathbb{E}_{\mathbb{D}_n}[D_{KL}(p_{1-\tau}\|\hat{p}_{1-\tau})] = O\Big(\mathcal{D}^{2\lfloor\beta\rfloor+2\vee\beta}(n\tau^4)^{-\frac{2\beta}{(2\mathcal{D}+2\beta+1)}}\Big). \tag{10}$$

### B.3.3 DISTRIBUTION CONSISTENCY

Combining Equation 8, Theorem 4 and Equation 10, as $\tau\mathcal{D} \to 0$, we have

$$\mathbb{E}_{\mathbb{D}_n}\Big[\mathrm{TV}(p_1, \hat{p}_{1-\tau})\Big] = O\Big(\mathcal{D}^{\lfloor\beta\rfloor+1\vee\beta}(n\tau^4)^{-\frac{\beta}{(2\mathcal{D}+2\beta+1)}} + \tau\mathcal{D}\Big).$$

When $\mathcal{D}^{\lfloor\beta\rfloor+1\vee\beta}(n\tau^4)^{-\frac{\beta}{(2\mathcal{D}+2\beta+1)}} + \tau\mathcal{D}$ goes to zero (e.g., the sequence dimension $\mathcal{D}$ is fixed and the early stopping parameter $\tau = o(n^{-1/4})$), we can find that the distribution estimation is consistent in the sense of $\mathbb{E}_{\mathbb{D}_n}[\mathrm{TV}(p_1, \hat{p}_{1-\tau})] \to 0$.

## C  SOME USEFUL LEMMAS

**Lemma 1** (Exit Time). *Under the assumptions in Proposition 1, considering the stopping time* $T_t = \min\{h > 0 : \Delta X(t+h) \neq 0\}$, *we have*

$$\mathbb{P}(T_t > h | \mathcal{F}_t) = \exp\Big(\int_t^{t+h} u_s(X(t), X(t))ds\Big).$$

*Proof.* Consider the event $E_n = \{X(t) = X(t + \frac{h}{2^n}) = \cdots = X(t+h)\}$. Since $E_{n+1} \subseteq E_n$, by dominated convergence theorem and Markov property, we have

$$
\begin{aligned}
&\mathbb{P}(T_t > h | \mathcal{F}_t) \\
&= \mathbb{P}(\cap_n E_n | \mathcal{F}_t) = \lim_{n\to\infty} \mathbb{P}(E_n | \mathcal{F}_t) \\
&= \lim_{n\to\infty} \prod_{i=1}^{2^n} \Big(\mathbb{P}\Big(X(t + \frac{i}{2^n}h) = X(t)\Big| X(t + \frac{i-1}{2^n}h) = X(t), \mathcal{F}_t\Big)\Big) \\
&= \lim_{n\to\infty} \exp\Big(\sum_{i=1}^{2^n} \frac{h}{2^n} \frac{\log\mathbb{P}\Big(X(t + \frac{i}{2^n}h) = X(t)\Big| X(t + \frac{i-1}{2^n}h) = X(t), \mathcal{F}_t\Big) - \log 1}{h/(2^n)}\Big) \\
&= \lim_{n\to\infty} \exp\Big(\sum_{i=1}^{2^n} \frac{h}{2^n} \frac{u_{t+\frac{i-1}{2^n}h}(X(t), X(t))\frac{h}{2^n} + R_i}{h/(2^n)}\Big) \\
&= \exp\Big(\int_t^{t+h} u_s(X(t), X(t))\mathrm{d}s\Big),
\end{aligned}
$$

where the remainder $R_i \leq C(h/2^n)^2$ for some constant $C$ not depending on $i$ by Proposition 1. $\square$

**Lemma 2** (Itô's formula). *Consider the random measure $N(t, A)$ associated with a CTMC $X(t)$, which is defined in Definition 2. Define a stochastic integral $W(t) = W(0) + \int_0^t \int_A K(s, x)N(ds, dx)$, where $K$ is a predictable process and $A \subseteq \mathcal{S}^\mathcal{D}$. For each $f \in C(\mathbb{R}), t \geq 0$, we have*

$$f(W(t)) - f(W(0)) = \int_0^t \int_A [f(W(s-) + K(s, x)) - f(W(s-))]N(ds, dx).$$

*Proof.* This proof is similar to Lemma 4.4.5 in Applebaum (2009). Let $T_0^A = 0$ and $T_n^A = \inf\{t > T_{n-1}^A : \Delta N(t, A) \neq 0\}$. Note that

$$
\begin{aligned}
f(W(t)) - f(W(0)) &= \sum_{n=1}^{\infty} f(W(t \wedge T_n^A)) - f(W(t \wedge T_n^A-)) \\
&= \sum_{n=1}^{\infty} [f(W(t \wedge T_n^A-) + K(t \wedge T_n^A, X(t \wedge T_n^A))) - f(W(t \wedge T_n^A-))] \\
&= \int_0^t \int_A [f(W(s-) + K(s, x)) - f(W(s-))]N(ds, dx),
\end{aligned}
$$

which completes the proof. $\square$

**Lemma 3** (Exponential martingale). *Suppose that $N(t, A)$ is the random measure associated with the CTMC $X(t)$. Consider the following stochastic integral*

$$W(t) = W(0) + \int_0^t \sum_{x \neq X(s-)} F(s, x) ds + \int_0^t \int_{x \in \mathcal{S}^{\mathcal{D}}} K(s, x) N(ds, dx).$$

*Then $\exp(W(t))$ is an exponential martingale if for each $x \in \mathcal{S}^{\mathcal{D}}$,*

$$F(t, x) = -(e^{K(t,x)} - 1) u_t(x, X(t-)) \mathbb{1}(x \neq X(t-)).$$

*Proof.* By Itô's product formula (e.g., Theorem 4.4.13 in Applebaum, 2009) and Itô's formula (Lemma 2), we have

$$d[\exp(W(t))] = \exp(W(t)) \sum_{x \neq X(t-)} F(t, x) dt + \exp(W(t-)) \int_{x \in \mathcal{S}^{\mathcal{D}}} (e^{K(t,x)} - 1) N(dt, dx)$$

$$= \exp(W(t-)) \sum_{x \neq X(t-)} \left\{ F(t, x) dt + (e^{K(t,x)} - 1) N(dt, x) \right\}$$

$$= \exp(W(t-)) \sum_{x \neq X(t-)} \left\{ (e^{K(t,x)} - 1) \tilde{N}(dt, x) \right\},$$

which completes the proof by Proposition 3. $\qquad\square$

**Lemma 4.** *For any random variable $Z \in \mathcal{F}_{t+h}$, we have*

$$\mathbb{E}^Y(Z|\mathcal{F}_t) = \frac{\mathbb{E}^X\left[ \frac{d\mathbb{P}^Y}{d\mathbb{P}^X}\Big|_{t+h} Z|\mathcal{F}_t \right]}{\frac{d\mathbb{P}^Y}{d\mathbb{P}^X}\Big|_t}.$$

*Proof.* For any set $A \in \mathcal{F}_t$, we have

$$\mathbb{E}^Y(Z\mathbb{1}_A) = \mathbb{E}^X\left[ \mathbb{1}_A \mathbb{E}^X\left( \frac{d\mathbb{P}^Y}{d\mathbb{P}^X}\Big|_{t+h} Z|\mathcal{F}_t \right) \right]$$

$$= \mathbb{E}^Y\left[ \frac{\mathbb{1}_A \mathbb{E}^X\left( \frac{d\mathbb{P}^Y}{d\mathbb{P}^X}\Big|_{t+h} Z|\mathcal{F}_t \right)}{\frac{d\mathbb{P}^Y}{d\mathbb{P}^X}\Big|_t} \right],$$

which completes the proof by the definition of the R-N derivative. $\qquad\square$

**Lemma 5** (Theorem 3.3 in Jiao et al. (2023)). *Assume that $f \in \mathcal{H}^\beta([0,1]^d, B_0)$ with $\beta = s + r, s \in \mathbb{N}$ and $r \in (0, 1]$. For any $S_1, S_2 \in \mathbb{N}^+$, there exists a function $\phi_0$ implemented by a ReLU network with width $W = 38(\lfloor \beta \rfloor + 1)^2 d^{\lfloor \beta \rfloor + 1} S_2 \lceil \log_2(8S_2) \rceil$ and depth $D = 21(\lfloor \beta \rfloor + 1)^2 S_1 \lceil \log_2(8S_1) \rceil$ such that*

$$|f(x) - \phi_0(x)| \leq 18 B_0 (\lfloor \beta \rfloor + 1)^2 d^{\lfloor \beta \rfloor + (\beta \vee 1)/2} (S_1 S_2)^{-2\beta/d},$$

*for all $x \in [0,1]^d \backslash \Omega([0,1]^d, K, \delta)$, where*

$$\Omega([0,1]^d, K, \delta) = \cup_{i=1}^d \{x = [x_1, x_2, \ldots, x_d]^\top : x_i \in \cup_{k=1}^{K-1}(k/K - \delta, k/K)\},$$

*with $K = \lfloor (S_1 S_2)^{2/d} \rfloor$ and $\delta$ an arbitrary number in $(0, 1/(3K)]$*

**Lemma 6** (Theorem 12.2 in Anthony & Bartlett (2009)). *Let $\mathcal{G}$ be a set of real functions that map a domain $\mathcal{X}$ to a bounded interval $[0, B]$. The pseudo-dimension $Pdim(\mathcal{G})$ of $\mathcal{G}$ is defined as the largest integer $m$ for which there exists $(x_1, \ldots, x_m, y_1, \ldots, y_m) \in \mathcal{X} \times \mathbb{R}^m$ such that for any $(b_1, \ldots, b_m) \in \{0, 1\}^m$ there exists $f \in \mathcal{G}$ such that $f(x_i) > y_i$ if and only if $b_i = 1$ for any $i \in [m]$. Then, for $n \geq Pdim(\mathcal{G})$ and $B \geq \epsilon$, we have*

$$\mathcal{N}_n(\epsilon, \mathcal{G}, L^\infty) \leq \left( \frac{eBn}{\epsilon Pdim(\mathcal{G})} \right)^{Pdim(\mathcal{G})}.$$

**Lemma 7** (Theorem 7 in Bartlett et al. (2019)). *Let $\mathcal{G}$ be a ReLU neural network function class with depth $L$ and number of parameters $S$. Then, there exists a universal constant $C$ such that*

$$Pdim(\mathcal{G}) \leq CSL \log S.$$

## D PROOF OF RESULTS FOR CTMC

### D.1 PROOF OF PROPOSITION 1

*Proof.* By construction, $X(t)$ has Markov property since the Poisson process $N(t)$ has independent increments. By conditioning argument (e.g., Theorem 3.7.9 in Durrett, 2019) and the definition of Poisson processes, for $z \neq x$, we have

$$\mathbb{P}(X(t+h) = z | X(t) = x) = Mh \exp(-Mh) \mathbb{P}(X(t+h) = z | X(t) = x, N(t+h) - N(t) = 1) + O(h^2)$$

$$= Mh \exp(-Mh) \times \Big( \int_0^h \underbrace{\frac{1}{h}}_{\text{jump location: uniform distribution}} \frac{u_{t+\theta}(z, x)}{M} d\theta \Big) + O(h^2)$$

$$= u_t(z, x)h + R_t,$$

where the remainder $R_t$ satisfying

$$R_t \leq Lh^2 \exp(-Mh) + \mathbb{P}(N(t+h) - N(t) > 1, X(t+h) = z | X(t) = x)$$
$$\leq \exp(-Mh)[Lh^2 + \exp(Mh) - 1 - Mh] \leq (M^2 + L)h^2 = O(h^2).$$

Consequently, since the state space is finite, we have

$$\mathbb{P}(X(t+h) = x | X(t) = x) = 1 - \sum_{z \neq x} \mathbb{P}(X(t+h) = z | X(t) = x) = 1 + u_t(x, x)h + O(h^2),$$

which completes the proof. ∎

### D.2 PROOF OF PROPOSITION 2

*Proof.* By Definition 1, we have

$$p_{t+h}(x) - p_t(x) = \sum_{z \in \mathcal{S}^{\mathcal{D}}} (p_{t+h|t}(x|z) - \delta_z(x)) p_t(z) = \sum_{z \in \mathcal{S}^{\mathcal{D}}} (u_t(x, z)h + o(h)) p_t(z).$$

Then,

$$\dot{p}_t(x) = \lim_{h \to 0^+} \frac{p_{t+h}(x) - p_t(x)}{h} = \sum_{z \in \mathcal{S}^{\mathcal{D}}} u_t(x, z) p_t(z),$$

which completes the proof. ∎

### D.3 PROOF OF PROPOSITION 3

*Proof.* In this proof, we consider the event $\{X(t) = x\}$ if we are given $\mathcal{F}_t$. Following the proof of Proposition 1, conditioning on $\mathcal{F}_t$, since the Poisson process $N(t)$ has independent increments, we have

$$\mathbb{P}(N(t+h, A) - N(t, A) = 1 | \mathcal{F}_t)$$
$$\leq \mathbb{P}(N(t+h) - N(t) = 1) \sum_{z \in A} \mathbb{P}(X(t+h) = z, X(t) \neq z | N(t+h) - N(t) = 1, \mathcal{F}_t)$$
$$+ \mathbb{P}(N(t+h) - N(t) > 1)$$
$$= \sum_{z \in A} u_t(z, X(t)) \mathbb{1}(z \neq X(t)) h + R_t,$$

where $R_t$ is bounded by $|A|(M^2 + L)h^2$. Since the rate of $X(t)$ is bounded above by $M$, by uniformization, we have

$$\mathbb{E}\Big[ \big\{ N(t+h, A) - N(t, A) \big\} \mathbb{1}(N(t+h, A) - N(t, A) \geq 2) \Big| \mathcal{F}_t \Big]$$
$$\leq \mathbb{E}\Big[ \big\{ N(t+h) - N(t) \big\} \mathbb{1}(N(t+h) - N(t) \geq 2) \Big| \mathcal{F}_t \Big]$$
$$= Mh - (Mh \exp(-Mh)) \leq M^2 h^2.$$

Then,

$$\mathbb{E}\Big[N(t+h,A) - N(t,A)|\mathcal{F}_t\Big] = \sum_{z \in A} u_t(z, X(t))\mathbb{1}(z \neq X(t))h + R.$$

Here, the remainder $R$ is bounded by $M^2h^2 + |A|(M^2 + L)h^2$. Then, since $X(t)$ has only finite jumps in $[s, t]$ almost surely, by dominated convergence theorem, we have

$$\mathbb{E}\Big[N(t,A) - N(s,A)\Big|\mathcal{F}_s\Big]$$

$$= \sum_{i=1}^{n} \mathbb{E}\Big[\mathbb{E}[N(s + \frac{i}{n}(t-s), A) - N(s + \frac{i-1}{n}(t-s), A)|\mathcal{F}_{s + \frac{i-1}{n}(t-s)}]\Big|\mathcal{F}_s\Big]$$

$$= \sum_{i=1}^{n} \mathbb{E}\Big[\sum_{z \in A} u_{s + \frac{i-1}{n}(t-s)}(z, X(s + \frac{i-1}{n}(t-s)))\mathbb{1}(z \neq X(s + \frac{i-1}{n}(t-s)))\frac{1}{n}(t-s)\Big|\mathcal{F}_s\Big]$$

$$+ O(\frac{1}{n})$$

$$\to \mathbb{E}[\int_s^t \sum_{z \in A} u_r(z, X(r-))\mathbb{1}(z \neq X(r-))\mathrm{d}r|\mathcal{F}_s],$$

as $n \to \infty$. Thus, the process

$$\tilde{N}(t, A) = N(t, A) - \int_0^t \underbrace{\sum_{z \in A} u_s(z, X(s-))\mathbb{1}(z \neq X(s-))}_{\text{predictable process}} \mathrm{d}s$$

is a (martingale-valued) compensated random measure of the random measure $N(t, A)$. We are done.

$\square$

### D.4 Proof of Theorem 1

*Proof.* By computing directly, we can obtain that

$$\frac{\mathrm{d}\mathbb{P}^Y}{\mathrm{d}\mathbb{P}^X}\Big|_t = \exp\Big\{\int_0^t \sum_{x \neq X(s-)} [u_s^X(x, X(s-)) - u_s^Y(x, X(s-))]\mathrm{d}s + \int_0^t \sum_{x \neq X(s-)} \log\frac{u_s^Y(x, X(s-))}{u_s^X(x, X(s-))} N(\mathrm{d}s, x)\Big\}.$$

Next, we divide the proof into two parts: first we prove that $\tilde{N}^Y(t, A)$ is a $\mathbb{P}^Y$-martingale, and then we conclude that $X(t)$ is a $\mathbb{P}^Y$-CTMC with rate $u_t^Y$.

$\tilde{N}^Y(t, A)$ **is a $\mathbb{P}^Y$-martingale.** By Itô's product formula (see Theorem 4.4.13 in Applebaum, 2009) and Lemma 3, we have

$$\mathrm{d}[\tilde{N}^Y(t, A)\exp(W(t))] = \tilde{N}^Y(t-, A)\exp(W(t-)) \sum_{x \neq X(t-)} \Big\{(e^{K(t,x)} - 1)\tilde{N}^X(\mathrm{d}t, x)\Big\}$$

$$+ \exp(W(t-))\tilde{N}^Y(\mathrm{d}t, A)$$

$$+ \exp(W(t-)) \underbrace{\sum_{x \neq X(t-):x \in A} \Big\{(e^{K(t,x)} - 1)N(\mathrm{d}t, x)\Big\}}_{\text{quadratic variation}}$$

$$= \tilde{N}^Y(t-, A)\exp(W(t-)) \sum_{x \neq X(t-)} \Big\{(e^{K(t,x)} - 1)\tilde{N}^X(\mathrm{d}t, x)\Big\}$$

$$+ \exp(W(t-))\tilde{N}^X(\mathrm{d}t, A)$$

$$+ \exp(W(t-)) \sum_{x \neq X(t-):x \in A} \Big\{(e^{K(t,x)} - 1)N(\mathrm{d}t, x)\Big\}$$

$$- \exp(W(t-)) \sum_{x \neq X(t-):x \in A} \left[ \frac{u_t^Y(x, X(t-))}{u_t^X(x, X(t-))} - 1 \right] u_t^X(x, X(t-)) \mathrm{d}t$$

$$= \tilde{N}^Y(t-, A) \exp(W(t-)) \sum_{x \neq X(t-)} \left\{ \frac{u_t^Y(x, X(t-))}{u_t^X(x, X(t-))} - 1) \tilde{N}^X(\mathrm{d}t, x) \right\}$$

$$+ \exp(W(t-)) \sum_{x \neq X(t-):x \in A} \frac{u_t^Y(x, X(t-))}{u_t^X(x, X(t-))} \tilde{N}^X(\mathrm{d}t, x),$$

which implies that $\tilde{N}^Y(t, A) \exp(W(t))$ is a $\mathbb{P}^X$-martingale. By Lemma 5.2.11 in Applebaum (2009), $\tilde{N}^Y(t, A)$ is a $\mathbb{P}^Y$-martingale.

$X(t)$ **is a** $\mathbb{P}^Y$**-CTMC with rate** $u_t^Y$. Considering $Z = \mathbb{1}(X(t+h) = z)$ in Lemma 4, we can obtain that

$$\mathbb{P}^Y(X(t+h) = z|\mathcal{F}_t) = \mathbb{E}^X[\exp(W(t+h) - W(t))\mathbb{1}(X(t+h) = z)|\mathcal{F}_t].$$

Since the right-hand side in above equation only depends on $X(t)$, thus $X(t)$ has Markov property under $\mathbb{P}^Y$. It suffices to show that $\mathbb{P}^Y(X(t+h) = z|\mathcal{F}_t) = u_t(z, X(t))h + o(h)$ for any $z \neq X(t)$.

Since $\exp(W(t+h) - W(t)) \leq \exp(Ch)$ for some constant $C$, by Lemma 4, we have

$$\mathbb{P}^Y(N(t+h, A) - N(t, A) > 1|\mathcal{F}_t) = \frac{\mathbb{E}^X\left[\frac{\mathrm{d}\mathbb{P}^Y}{\mathrm{d}\mathbb{P}^X}\Big|_{t+h} \mathbb{1}(N(t+h, A) - N(t, A) > 1)|\mathcal{F}_t\right]}{\frac{\mathrm{d}\mathbb{P}^Y}{\mathrm{d}\mathbb{P}^X}\Big|_t}$$

$$\leq \exp(Ch)\mathbb{P}^X(N(t+h, A) - N(t, A) > 1|\mathcal{F}_t)$$

$$= O(h^2).$$

Note that, for $\mathbb{P}^Y$-submartingale $(\tilde{N}^Y)^2$, by Itô product formula, we have Doob-Meyer decomposition:

$$\mathrm{d}(\tilde{N}^Y)^2(t, A) = 2\tilde{N}^Y(t-)\mathrm{d}\tilde{N}^Y(t, A) + \mathrm{d}[\tilde{N}^Y, \tilde{N}^Y](t, A)$$

$$= 2\tilde{N}^Y(t-, A)\mathrm{d}\tilde{N}^Y(t, A) + \mathrm{d}N(t, A).$$

Then, by Lemma 4 again, $\mathbb{E}^Y[(\tilde{N}^Y(t+h, A) - \tilde{N}^Y(t, A))^2|\mathcal{F}_t] = \mathbb{E}^Y[N(t+h, A) - N(t, A)|\mathcal{F}_t] = O(h)$. By Cauchy-Schwarz inequality, we can obtain

$$\mathbb{E}^Y[(N(t+h, A) - N(t, A))\mathbb{1}(N(t+h, A) - N(t, A) \geq 2)|\mathcal{F}_t]$$

$$\leq \sqrt{\mathbb{E}^Y[(N(t+h, A) - N(t, A))^2|\mathcal{F}_t]\mathbb{P}^Y(N(t+h, A) - N(t, A) > 1|\mathcal{F}_t)}$$

$$\leq \sqrt{\left(2\mathbb{E}^Y[(\tilde{N}^Y(t+h, A) - \tilde{N}^Y(t, A))^2|\mathcal{F}_t] + 2\mathbb{E}^Y\left[\left(\int_t^{t+h} \sum_{z \in A} u_s^Y(z, X(s-))\mathbb{1}(z \neq X(s-))\mathrm{d}s\right)^2 \Big| \mathcal{F}_t\right]\right)}$$

$$\times \sqrt{\mathbb{P}^Y(N(t+h, A) - N(t, A) > 1|\mathcal{F}_t)}$$

$$= O(h^{3/2})$$

Then, conditioning on $\mathcal{F}_t$, for $z \neq X(t)$, we have

$$\mathbb{P}^Y(X(t+h) = z|\mathcal{F}_t) \leq \mathbb{P}^Y(N(t+h, z) - N(t, z) = 1|\mathcal{F}_t) + O(h^2)$$

$$= \mathbb{E}^Y(N(t+h, z) - N(t, z)|\mathcal{F}_t) + O(h^{3/2})$$

$$= \int_t^{t+h} u_t^Y(z, X(t))\mathrm{d}s + \int_t^{t+h} u_s^Y(z, X(t)) - u_t^Y(z, X(t))\mathrm{d}s$$

$$+ \mathbb{E}^Y(\int_t^{t+h} u_s^Y(z, X(s-)) - u_s^Y(z, X(t))\mathrm{d}s|\mathcal{F}_t) + O(h^{3/2})$$

$$= u_t(z, X(t))h + O(h^{3/2}),$$

since

$$\mathbb{E}^Y\big(\int_t^{t+h} u_s^Y(z, X(s-)) - u_s^Y(z, X(t))\mathrm{d}s|\mathcal{F}_t\big) \le 2Mh\mathbb{P}^Y(N(t+h, \mathcal{S}^\mathcal{D}) - N(t, \mathcal{S}^\mathcal{D}) \ge 1|\mathcal{F}_t)$$

$$\le 2Mh\mathbb{E}^Y(N(t+h, \mathcal{S}^\mathcal{D}) - N(t, \mathcal{S}^\mathcal{D})|\mathcal{F}_t)$$

$$\le 2M^2h^2 = O(h^2).$$

This completes the proof.

$\square$

### D.5 PROOF OF THEOREM 2

*Proof.* We compute directly:

$$D_{\mathrm{KL}}(\mathbb{P}_t^X||\mathbb{P}_t^Y)$$

$$= \mathbb{E}^X\Big[ -\log(\frac{\mathrm{d}\mathbb{P}^Y}{\mathrm{d}\mathbb{P}^X}\Big|_t)\Big]$$

$$= -\mathbb{E}^X\Big\{ \int_0^t \sum_{x \ne X(s-)} [u_s^X(x, X(s-)) - u_s^Y(x, X(s-))] + u_s^X(x, X(s-)) \log \frac{u_s^Y(x, X(s-))}{u_s^X(x, X(s-))}\mathrm{d}s \Big\}$$

$$- \mathbb{E}^X\Big\{ \int_0^t \sum_{x \ne X(s-)} \log \frac{u_s^Y(x, X(s-))}{u_s^X(x, X(s-))} \tilde{N}^X(\mathrm{d}s, x) \Big\}$$

$$= \mathbb{E}^X\Big\{ \int_0^t \sum_{x \ne X(s)} D_F(u_s^X(x, X(s))||u_s^Y(x, X(s)))\mathrm{d}s \Big\},$$

where the last equation holds since the integrator is the Lesbesgue measure.

By Jensen's inequality, we have

$$D_{\mathrm{KL}}(p_t^X||p_t^Y) = \mathbb{E}^X\Big[ -\log(\frac{\mathrm{d}\mathbb{P}^Y}{\mathrm{d}\mathbb{P}^X}\Big|_{\sigma(X_t)})\Big]$$

$$\le \mathbb{E}^X\Big[ -\log(\frac{\mathrm{d}\mathbb{P}^Y}{\mathrm{d}\mathbb{P}^X}\Big|_{\mathcal{F}_t})\Big]$$

$$= \mathbb{E}^X\Big\{ \int_0^t \sum_{x \ne X(s)} D_F(u_s^X(x, X(s))||u_s^Y(x, X(s)))\mathrm{d}s \Big\}.$$

This completes the proof.

$\square$

## E PROOF OF MAIN RESULTS

### E.1 PROOF OF PROPOSITION 4

*Proof.* This proof is similar to the proof of Lemma 3.1 in Jiao et al. (2023). Note that

$$\mathbb{E}_{\mathbb{D}_n}[\mathcal{L}_n(\hat{u}) - \mathcal{L}_n(u^0)] \le \mathbb{E}_{\mathbb{D}_n}[\mathcal{L}_n(u^*) - \mathcal{L}_n(u^0)],$$

which yields that

$$-\mathcal{L}(u^0) \le 2\mathcal{L}(u^*) - \mathcal{L}(u^0) - \mathbb{E}_{\mathbb{D}_n}[2\mathcal{L}_n(\hat{u})].$$

Then we have

$$\mathbb{E}_{\mathbb{D}_n}[\mathcal{L}(\hat{u}) - \mathcal{L}(u^0)] \le \mathbb{E}_{\mathbb{D}_n}[\mathcal{L}(\hat{u}) + \mathcal{L}(u^0) - 2\mathcal{L}_n(\hat{u})] + 2[\mathcal{L}(u^*) - \mathcal{L}(u^0)],$$

which completes the proof by using Equation 7.

$\square$

### E.2 PROOF OF THEOREM 3

*Sketch of proof.* We first decompose the objective into $\mathcal{D}$ terms

$$\mathbb{E}_{\mathbb{D}_n}[\mathcal{L}(\hat{u}) + \mathcal{L}(u^0) - 2\mathcal{L}_n(\hat{u})] = \mathbb{E}_{\mathbb{D}_n}\Big[\mathbb{E}_Z[\sum_{d=1}^{\mathcal{D}} \sum_{d:z\neq X(\mathbf{t})} g(\hat{u}, z, Z)] - \frac{2}{n}\sum_{i=1}^{n}\sum_{d=1}^{\mathcal{D}} \sum_{d:z\neq X_i(\mathbf{t}_i)} g(\hat{u}, z, Z_i)\Big],$$

where we use $\sum_{d:z\neq X(\mathbf{t})} = \sum_{z^d \neq X(\mathbf{t})^d; z^{\backslash d}=X(\mathbf{t})^{\backslash d}}$ for notational simplicity. To bound the expectation, we focus on the tail probability

$$\mathbb{P}\Big(\exists (u,d) \in \mathcal{G}_n \times [\mathcal{D}] : \mathbb{E}\sum_{d:z\neq X(\mathbf{t})} g(u,z,Z) - \frac{2}{n}\sum_{i=1}^{n}\sum_{d:z\neq X_i(\mathbf{t}_i)} g(u,z,Z_i) > \frac{t}{\mathcal{D}}\Big).$$

Next, we replace the expectation by an empirical mean of a "ghost" sample $\mathbb{D}'_n$ independent of $\mathbb{D}_n$. Additionally, to use concentration inequality, we have to consider the nonnegative empirical mean $\frac{1}{n}\sum_{i=1}^{n}\Big\{\sum_{d:z\neq X_i(\mathbf{t}_i)} g(u,z,Z_i)\Big\}^2$ instead of $\frac{1}{n}\sum_{i=1}^{n}\sum_{d:z\neq X_i(\mathbf{t}_i)} g(u,z,Z_i)$. By symmetrization argument and introducing Rademacher variables $\{\epsilon_i\}_{i=1}^{n}$, the above probability can be bounded by

$$\mathbb{P}\Big\{\exists (u,d) \in \mathcal{G}_n \times [\mathcal{D}], \frac{1}{n}\sum_{i=1}^{n}\epsilon_i\Big\{\sum_{d:z\neq X_i(\mathbf{t}_i)} g(u,z,Z_i)\Big\}^2 > \Big[\frac{ct}{\mathcal{D}} + \frac{c}{n}\sum_{i=1}^{n}\Big\{\sum_{d:z\neq X_i(\mathbf{t}_i)} g(u,z,Z_i)\Big\}^2\Big]\Big\}$$

$$+ \mathbb{P}\Big(\exists (u,d) \in \mathcal{G}_n \times [\mathcal{D}] : \frac{1}{n}\sum_{i=1}^{n}\epsilon_i \sum_{d:z\neq X_i(\mathbf{t}_i)} g(u,z,Z_i) > \frac{ct}{\mathcal{D}} + \frac{c}{n}\sum_{i=1}^{n}\Big\{\sum_{d:z\neq X_i(\mathbf{t}_i)} g(u,z,Z_i)\Big\}^2\Big).$$

Finally, conditioning on $\mathbb{D}_n$ and introducing a covering, we can apply some concentration inequalities to bound these two terms.

*Proof.* This proof is similar to the proof of Theorem 11.4 in Györfi et al. (2002), where they only consider the squared loss. Note that

$$\mathbb{E}_{\mathbb{D}_n}[\mathcal{L}(\hat{u}) + \mathcal{L}(u^0) - 2\mathcal{L}_n(\hat{u})]$$

$$= \mathbb{E}_{\mathbb{D}_n}[\mathcal{L}(\hat{u}) - \mathcal{L}(u^0) - 2\mathcal{L}_n(\hat{u}) + 2\mathcal{L}(u^0)]$$

$$= \mathbb{E}_{\mathbb{D}_n}\Big[\mathbb{E}_Z[\sum_{z\neq X(\mathbf{t})} g(\hat{u},z,Z)] - \frac{2}{n}\sum_{i=1}^{n}\sum_{z\neq X_i(\mathbf{t}_i)} g(\hat{u},z,Z_i)\Big]$$

$$= \mathbb{E}_{\mathbb{D}_n}\Big[\mathbb{E}_Z[\sum_{d=1}^{\mathcal{D}}\sum_{d:z\neq X(\mathbf{t})} g(\hat{u},z,Z)] - \frac{2}{n}\sum_{i=1}^{n}\sum_{d=1}^{\mathcal{D}}\sum_{d:z\neq X_i(\mathbf{t}_i)} g(\hat{u},z,Z_i)\Big],$$

Notice that

$$\mathbb{E}_{\mathbb{D}}\Big[\mathbb{E}_Z[\sum_{d=1}^{\mathcal{D}}\sum_{d:z\neq X(\mathbf{t})} g(\hat{u},z,Z)] - \frac{2}{n}\sum_{i=1}^{n}\sum_{d=1}^{\mathcal{D}}\sum_{d:z\neq X_i(\mathbf{t}_i)} g(\hat{u},z,Z_i)\Big]$$

$$\leq a_n + \int_{a_n}^{\infty}\mathbb{P}\Big(\mathbb{E}_Z[\sum_{d=1}^{\mathcal{D}}\sum_{d:z\neq X(\mathbf{t})} g(\hat{u},z,Z)] - \frac{2}{n}\sum_{i=1}^{n}\sum_{d=1}^{\mathcal{D}}\sum_{d:z\neq X_i(\mathbf{t}_i)} g(\hat{u},z,Z_i) > t\Big)\mathrm{d}t,$$

where $a_n$ is a quantity depending on $n$, which we will choose later. Thus, we can focus on the following probability first.

$$\mathbb{P}\Big(\mathbb{E}_Z[\sum_{d=1}^{\mathcal{D}}\sum_{d:z\neq X(\mathbf{t})} g(\hat{u},z,Z)] - \frac{2}{n}\sum_{i=1}^{n}\sum_{d=1}^{\mathcal{D}}\sum_{d:z\neq X_i(\mathbf{t}_i)} g(\hat{u},z,Z_i) > t\Big)$$

Following the proof of symmetrization in probability (e.g., Lemma 2.3.7 in van der Vaart & Wellner (2023)), consider the following event

$$\mathcal{B}_1 = \{\hat{\mathcal{A}}(t) \text{ is a non-empty set}\},$$

where $t \le 1$ and

$$\hat{\mathcal{A}}(t) = \Big\{(u,d) \in \mathcal{G}_n \times [\mathcal{D}] : \mathbb{E}[\sum_{d:z \neq X(\mathbf{t})} g(u,z,Z)] - \frac{2}{n}\sum_{i=1}^{n}\sum_{d:z \neq X_i(\mathbf{t}_i)} g(u,z,Z_i) > \frac{t}{\mathcal{D}}\Big\}$$

$$= \Big\{(u,d) \in \mathcal{G}_n \times [\mathcal{D}] : \mathbb{E}[\sum_{d:z \neq X(\mathbf{t})} g(u,z,Z)] - \frac{1}{n}\sum_{i=1}^{n}\sum_{d:z \neq X_i(\mathbf{t}_i)} g(u,z,Z_i) \tag{11}$$

$$> \frac{1}{3}\Big(\frac{2t}{\mathcal{D}} + \mathbb{E}\sum_{d:z \neq X(\mathbf{t})} g(u,z,Z) + \frac{1}{n}\sum_{i=1}^{n}\sum_{d:z \neq X_i(\mathbf{t}_i)} g(u,z,Z_i)\Big)\Big\}.$$

Let $(\bar{u},\bar{d})$ be a (random) function such that $(\bar{u},\bar{d}) \in \hat{\mathcal{A}}(t)$ if $\mathcal{B}_1$ holds, and let $(\bar{u},\bar{d}) = (1,1)$ if $\mathcal{B}_1$ does not hold, where $(\bar{u},\bar{d})$ depends on $\mathbb{D}_n$. By assumptions, note that $|g(u,z,Z)| \le \frac{2\overline{M}_c^2(\overline{M}+1)}{\min(1,\underline{M})\underline{M}_c} \triangleq K_1$ for any $u \in \mathcal{G}_n$ and $z \in \mathcal{S}^{\mathcal{D}}$ such that $\mathrm{d}^H(z, X(\mathbf{t})) = 1$. Let $\mathbb{D}'_n = \{Z'_i\}_{i=1}^{n}$ be an independent copy of $\mathbb{D}_n$. By Markov's inequality, we have

$$\mathbb{P}\Big\{\mathbb{E}[\sum_{\bar{d}:z \neq X(\mathbf{t})} g(\bar{u},z,Z)|\mathbb{D}_n] - \frac{1}{n}\sum_{i=1}^{n}\sum_{\bar{d}:z \neq X'_i(\mathbf{t}'_i)} g(\bar{u},z,Z'_i) > \frac{1}{4}\Big[\frac{t}{\mathcal{D}} + \mathbb{E}[\sum_{\bar{d}:z \neq X(\mathbf{t})} g(\bar{u},z,Z)|\mathbb{D}_n]\Big]\Big|\mathbb{D}_n\Big\}$$

$$\le \frac{16\mathbb{E}[(\sum_{\bar{d}:z \neq X(\mathbf{t})} g(\bar{u},z,Z))^2|\mathbb{D}_n]}{n(\frac{t}{\mathcal{D}} + \mathbb{E}[\sum_{\bar{d}:z \neq X(\mathbf{t})} g(\bar{u},z,Z)|\mathbb{D}_n])^2} \le \frac{8K_1\mathcal{D}|\mathcal{S}|}{nt} \triangleq \eta_1.$$

$$\tag{12}$$

Since $\mathcal{B}_1 \in \sigma(\mathbb{D}_n)$, we have

$$(1-\eta_1)\mathbb{P}\Big(\exists u \in \mathcal{G}_n : \mathbb{E}\sum_{d=1}^{\mathcal{D}}\sum_{d:z \neq X(\mathbf{t})} g(u,z,Z) - \frac{2}{n}\sum_{i=1}^{n}\sum_{d=1}^{\mathcal{D}}\sum_{d:z \neq X_i(\mathbf{t}_i)} g(u,z,Z_i) > t\Big)$$

$$\le (1-\eta_1)\mathbb{P}\Big(\exists (u,d) \in \mathcal{G}_n \times [\mathcal{D}] : \mathbb{E}\sum_{d:z \neq X(\mathbf{t})} g(u,z,Z) - \frac{2}{n}\sum_{i=1}^{n}\sum_{d:z \neq X_i(\mathbf{t}_i)} g(u,z,Z_i) > \frac{t}{\mathcal{D}}\Big)$$

$$\le \mathbb{E}\Big(\mathbb{1}(\mathcal{B}_1)\mathbb{P}\Big\{\mathbb{E}[\sum_{\bar{d}:z \neq X(\mathbf{t})} g(\bar{u},z,Z)|\mathbb{D}_n] - \frac{1}{n}\sum_{i=1}^{n}\sum_{\bar{d}:z \neq X'_i(\mathbf{t}'_i)} g(\bar{u},z,Z'_i) \tag{13}$$

$$\le \frac{1}{4}\Big[\frac{t}{\mathcal{D}} + \mathbb{E}[\sum_{\bar{d}:z \neq X(\mathbf{t})} g(\bar{u},z,Z)|\mathbb{D}_n]\Big]\Big|\mathbb{D}_n\Big\}\Big)$$

$$\le \mathbb{P}\Big(\exists (u,d) \in \mathcal{G}_n \times [\mathcal{D}] : \frac{1}{n}\sum_{i=1}^{n}\sum_{d:z \neq X'_i(\mathbf{t}'_i)} g(u,z,Z'_i) - \frac{1}{n}\sum_{i=1}^{n}\sum_{d:z \neq X_i(\mathbf{t}_i)} g(u,z,Z_i)$$

$$> \frac{1}{4}[\frac{t}{\mathcal{D}} + \mathbb{E}\sum_{d:z \neq X(\mathbf{t})} g(u,z,Z)]\Big),$$

where we use equation 12 and the definition of $\hat{A}(t)$ in the second inequality.

Since $\sum_{d:z \neq X'_i(\mathbf{t}'_i)} g(u,z,Z'_i)$ might be negative, we want to focus on the nonnegative quantities $\{\sum_{d:z \neq X'_i(\mathbf{t}'_i)} g(u,z,Z'_i)\}^2$. By a union bound, we can obtain that

$$\mathbb{P}\Big(\exists (u,d) \in \mathcal{G}_n \times [\mathcal{D}] : \frac{1}{n}\sum_{i=1}^{n}\sum_{d:z \neq X'_i(\mathbf{t}'_i)} g(u,z,Z'_i) - \frac{1}{n}\sum_{i=1}^{n}\sum_{d:z \neq X_i(\mathbf{t}_i)} g(u,z,Z_i)$$

$$> \frac{1}{4}[\frac{t}{\mathcal{D}} + \mathbb{E}\sum_{d:z \neq X(\mathbf{t})} g(u,z,Z)]\Big)$$

$$\le 2\mathbb{P}\Big(\exists (u,d) \in \mathcal{G}_n \times [\mathcal{D}] : \frac{1}{n}\sum_{i=1}^{n}(\sum_{d:z \neq X_i(\mathbf{t}_i)} g(u,z,Z_i))^2 - \mathbb{E}[\sum_{d:z \neq X(\mathbf{t})} g(u,z,Z)]^2$$

$$> \frac{1}{2}\Big[\frac{t}{\mathcal{D}} + \frac{1}{n}\sum_{i=1}^{n}(\sum_{d:z \neq X_i(\mathbf{t}_i)} g(u,z,Z_i))^2 + \mathbb{E}[\sum_{d:z \neq X(\mathbf{t})} g(u,z,Z)]^2\Big]\Big)$$

$$+ \mathbb{P}\Big(\exists (u,d) \in \mathcal{G}_n \times [\mathcal{D}] : \frac{1}{n}\sum_{i=1}^{n}\sum_{d:z\neq X_i'(\mathbf{t}_i')} g(u,z,Z_i') - \frac{1}{n}\sum_{i=1}^{n}\sum_{d:z\neq X_i(\mathbf{t}_i)} g(u,z,Z_i)$$

$$> \frac{1}{4}\Big[\frac{t}{\mathcal{D}} + \mathbb{E}\sum_{d:z\neq X(\mathbf{t})} g(u,z,Z)\Big],$$

$$\frac{1}{n}\sum_{i=1}^{n}\Big(\sum_{d:z\neq X_i'(\mathbf{t}_i')} g(u,z,Z_i')\Big)^2 - \mathbb{E}\Big[\sum_{d:z\neq X(\mathbf{t})} g(u,z,Z)\Big]^2$$

$$\leq \frac{1}{2}\Big[\frac{t}{\mathcal{D}} + \frac{1}{n}\sum_{i=1}^{n}\Big(\sum_{d:z\neq X_i'(\mathbf{t}_i')} g(u,z,Z_i')\Big)^2 + \mathbb{E}\Big[\sum_{d:z\neq X(\mathbf{t})} g(u,z,Z)\Big]^2\Big],$$

$$\frac{1}{n}\sum_{i=1}^{n}\Big(\sum_{d:z\neq X_i(\mathbf{t}_i)} g(u,z,Z_i)\Big)^2 - \mathbb{E}\Big[\sum_{d:z\neq X(\mathbf{t})} g(u,z,Z)\Big]^2$$

$$\leq \frac{1}{2}\Big[\frac{t}{\mathcal{D}} + \frac{1}{n}\sum_{i=1}^{n}\Big(\sum_{d:z\neq X_i(\mathbf{t}_i)} g(u,z,Z_i)\Big)^2 + \mathbb{E}\Big[\sum_{d:z\neq X(\mathbf{t})} g(u,z,Z)\Big]^2\Big]\Big)$$

$$\stackrel{\triangle}{=} 2\mathbb{P}(\mathcal{B}_2) + \mathbb{P}(\mathcal{B}_3)$$

**Bounding** $\mathbb{P}(\mathcal{B}_2)$**.** Let $(\bar{u},\bar{d}) \in \mathcal{G}_n \times [\mathcal{D}]$ be a function such that

$$\frac{1}{n}\sum_{i=1}^{n}\Big(\sum_{\bar{d}:z\neq X_i(\mathbf{t}_i)} g(\bar{u},z,Z_i)\Big)^2 - \mathbb{E}\Big[\sum_{\bar{d}:z\neq X(\mathbf{t})} g(\bar{u},z,Z)\Big]^2$$

$$> \frac{1}{2}\Big[\frac{t}{\mathcal{D}} + \frac{1}{n}\sum_{i=1}^{n}\Big(\sum_{\bar{d}:z\neq X_i(\mathbf{t}_i)} g(\bar{u},z,Z_i)\Big)^2 + \mathbb{E}\Big[\sum_{\bar{d}:z\neq X(\mathbf{t})} g(\bar{u},z,Z)\Big]^2\Big],$$

if $\mathcal{B}_2$ holds; and $(\bar{u},\bar{d}) = (1,1)$ otherwise, which depends on $\mathbb{D}_n$ Conditioning on $\mathcal{B}_2$ and $\mathbb{D}_n$, by Markov's inequality, we have

$$\mathbb{P}\Big\{\frac{1}{n}\sum_{i=1}^{n}\Big(\sum_{\bar{d}:z\neq X_i'(\mathbf{t}_i')} g(\bar{u},z,Z_i')\Big)^2 - \mathbb{E}\Big[\Big(\sum_{\bar{d}:z\neq X(\mathbf{t})} g(\bar{u},z,Z)\Big)^2\big|\mathbb{D}_n\Big]$$

$$> \frac{1}{8}\Big[\frac{t}{\mathcal{D}} + \frac{1}{n}\sum_{i=1}^{n}\Big(\sum_{\bar{d}:z\neq X_i'(\mathbf{t}_i')} g(\bar{u},z,Z_i')\Big)^2 + \mathbb{E}\Big[\sum_{\bar{d}:z\neq X(\mathbf{t})} g^2(\bar{u},z,Z)\big|\mathbb{D}\Big]\Big]\Big|\mathbb{D}_n\Big\}$$

$$= \mathbb{P}\Big\{\frac{7}{n}\sum_{i=1}^{n}\Big(\sum_{\bar{d}:z\neq X_i'(\mathbf{t}_i')} g(\bar{u},z,Z_i')\Big)^2 - 7\mathbb{E}\Big[\Big(\sum_{\bar{d}:z\neq X(\mathbf{t})} g(\bar{u},z,Z)\Big)^2\big|\mathbb{D}_n\Big] \tag{14}$$

$$> \frac{t}{\mathcal{D}} + 2\mathbb{E}\Big[\Big(\sum_{z\neq X(\mathbf{t})} g(\bar{u},z,Z)\Big)^2\big|\mathbb{D}_n\Big]\Big|\mathbb{D}_n\Big\}$$

$$\leq \frac{49\mathbb{E}\big[\big(\sum_{\bar{d}:z\neq X(\mathbf{t})} g(\bar{u},z,Z)\big)^4\big|\mathbb{D}_n\big]}{n\big(\frac{t}{\mathcal{D}} + 2\mathbb{E}\big[\big(\sum_{\bar{d}:z\neq X(\mathbf{t})} g(\bar{u},z,Z)\big)^2\big|\mathbb{D}_n\big]\big)^2} \leq \frac{49K_1^2|\mathcal{S}|^2\mathcal{D}}{4nt} \stackrel{\triangle}{=} \eta_2.$$

Then, by using equation 14 and a symmetrization argument, we have

$$(1-\eta_2)\mathbb{P}\Big(\mathcal{B}_2\Big)$$

$$\leq \mathbb{E}\Big(\mathbb{1}(\mathcal{B}_2)\mathbb{P}\Big\{\frac{1}{n}\sum_{i=1}^{n}\Big\{\sum_{d:z\neq X_i'(\mathbf{t}_i')} g(u,z,Z_i')\Big\}^2 - \mathbb{E}\Big[\Big(\sum_{d:z\neq X(\mathbf{t})} g(\bar{u},z,Z)\Big)^2\big|\mathbb{D}_n\Big]$$

$$\leq \frac{1}{8}\Big[\frac{t}{\mathcal{D}} + \frac{1}{n}\sum_{i=1}^{n}\Big\{\sum_{\bar{d}:z\neq X_i'(\mathbf{t}_i')} g(u,z,Z_i')\Big\}^2 + \mathbb{E}\Big[\Big(\sum_{\bar{d}:z\neq X(\mathbf{t})} g(\bar{u},z,Z)\Big)^2\big|\mathbb{D}_n\Big]\Big]\Big|\mathbb{D}_n\Big\}\Big)$$

$$\leq \mathbb{P}\Big(\exists (u,d) \in \mathcal{G}_n \times [\mathcal{D}], \frac{4}{n}\sum_{i=1}^{n}\Big\{\sum_{d:z \neq X_i(\mathbf{t}_i)} g(u,z,Z_i)\Big\}^2 - \frac{7}{n}\sum_{i=1}^{n}\Big\{\sum_{d:z \neq X_i'(\mathbf{t}_i')} g(u,z,Z_i')\Big\}^2$$

$$> 3\Big[\frac{t}{\mathcal{D}} + \mathbb{E}[\sum_{d:z \neq X(\mathbf{t})} g(u,z,Z)]^2\Big]\Big)$$

$$= \mathbb{P}\Big(\exists (u,d) \in \mathcal{G}_n \times [\mathcal{D}], \frac{1}{n}\sum_{i=1}^{n}\Big\{\sum_{d:z \neq X_i'(\mathbf{t}_i')} g(u,z,Z_i')\Big\}^2 - \frac{1}{n}\sum_{i=1}^{n}\Big\{\sum_{d:z \neq X_i(\mathbf{t}_i)} g(u,z,Z_i)\Big\}^2$$

$$> \frac{3}{11}\Big[\frac{2t}{\mathcal{D}} + 2\mathbb{E}[\sum_{d:z \neq X(\mathbf{t})} g(u,z,Z)]^2 + \frac{1}{n}\sum_{i=1}^{n}\Big\{\sum_{d:z \neq X_i'(\mathbf{t}_i')} g(u,z,Z_i')\Big\}^2$$

$$+ \frac{1}{n}\sum_{i=1}^{n}\Big\{\sum_{d:z \neq X_i(\mathbf{t}_i)} g(u,z,Z_i)\Big\}^2\Big]\Big)$$

$$\leq \mathbb{P}\Big(\exists (u,d) \in \mathcal{G}_n \times [\mathcal{D}], \frac{1}{n}\sum_{i=1}^{n}\Big\{\sum_{d:z \neq X_i'(\mathbf{t}_i')} g(u,z,Z_i')\Big\}^2 - \frac{1}{n}\sum_{i=1}^{n}\Big\{\sum_{d:z \neq X_i(\mathbf{t}_i)} g(u,z,Z_i)\Big\}^2$$

$$> \frac{3}{11}\Big[\frac{2t}{\mathcal{D}} + \frac{1}{n}\sum_{i=1}^{n}\Big\{\sum_{d:z \neq X_i'(\mathbf{t}_i')} g(u,z,Z_i')\Big\}^2 + \frac{1}{n}\sum_{i=1}^{n}\Big\{\sum_{d:z \neq X_i(\mathbf{t}_i)} g(u,z,Z_i)\Big\}^2\Big]\Big)$$

$$= \mathbb{P}\Big(\exists (u,d) \in \mathcal{G}_n \times [\mathcal{D}], \frac{1}{n}\sum_{i=1}^{n}\epsilon_i\Big\{\sum_{d:z \neq X_i'(\mathbf{t}_i')} g(u,z,Z_i')\Big\}^2 - \frac{1}{n}\sum_{i=1}^{n}\epsilon_i\Big\{\sum_{d:z \neq X_i(\mathbf{t}_i)} g(u,z,Z_i)\Big\}^2$$

$$> \frac{3}{11}\Big[\frac{2t}{\mathcal{D}} + \frac{1}{n}\sum_{i=1}^{n}\Big\{\sum_{d:z \neq X_i'(\mathbf{t}_i')} g(u,z,Z_i')\Big\}^2 + \frac{1}{n}\sum_{i=1}^{n}\Big\{\sum_{d:z \neq X_i(\mathbf{t}_i)} g(u,z,Z_i)\Big\}^2\Big]\Big)$$

$$\leq 2\mathbb{P}\Big\{\exists (u,d) \in \mathcal{G}_n \times [\mathcal{D}], \frac{1}{n}\sum_{i=1}^{n}\epsilon_i\Big\{\sum_{d:z \neq X_i(\mathbf{t}_i)} g(u,z,Z_i)\Big\}^2$$

$$> \frac{3}{11}\Big[\frac{t}{\mathcal{D}} + \frac{1}{n}\sum_{i=1}^{n}\Big\{\sum_{d:z \neq X_i(\mathbf{t}_i)} g(u,z,Z_i)\Big\}^2\Big]\Big\},$$

where $\{\epsilon_i\}_{i=1}^{n}$ is a sequence of i.i.d. Rademacher random variables, independent of $\mathbb{D}_n$.

Given $\mathbb{D}_n$ and $d \in [D]$, we consider the following sequence with length $n|\mathcal{S}|$:

$$\mathbf{s}_{|\mathcal{S}|n} = \Big\{(\mathbf{t}_i, X_i(\mathbf{t}_i)|_{X_i(\mathbf{t}_i)^d = s}, X_i(\mathbf{t}_i))\Big\}_{i \in [n], s \in \mathcal{S}},$$

where we denote $X_i(\mathbf{t}_i)|_{X_i(\mathbf{t}_i)^d = s} = (X_i(\mathbf{t}_i)^1, \ldots, X_i(\mathbf{t}_i)^{d-1}, s, X_i(\mathbf{t}_i)^{d+1}, \ldots, X_i(\mathbf{t}_i)^{\mathcal{P}})$. Let $\mathcal{H}_\delta^d(\mathbb{D}_n)$ be a $L^\infty$ $\delta$-covering of $\mathcal{G}_n|_{\mathbf{s}_{|\mathcal{S}|n}}$ with minimal size. That is to say, given $\mathbb{D}_n$ and $d \in [D]$, for any $u \in \mathcal{G}_n$, there exists $\tilde{u} \in \mathcal{H}_\delta^d(\mathbb{D}_n)$ such that for any $i \in [n]$ and $z \in \mathcal{S}^{\mathcal{D}}$ satisfying $z^d \neq X_i(\mathbf{t}_i)^d$ and $z^{\backslash d} = X_i(\mathbf{t}_i)^{\backslash d}$

$$|u_{\mathbf{t}_i}(z, X_i(\mathbf{t}_i)) - \tilde{u}_{\mathbf{t}_i}(z, X_i(\mathbf{t}_i))| \leq \delta.$$

Note that, for $\mathrm{d}^H(z, X(\mathbf{t})) = 1$,

$$|g(u_1, z, Z) - g(u_2, z, Z)| \leq (\frac{\overline{M_c}}{\underline{M}\underline{M_c}} + 1)|u_1(z, Z) - u_2(z, Z)| \overset{\triangle}{=} K_2|u_1(z, Z) - u_2(z, Z)|. \tag{15}$$

Then, given $\mathbb{D}_n$ and $d \in [\mathcal{D}]$, for any $u \in \mathcal{G}_n$, there exists $\tilde{u} \in \mathcal{H}_\delta^d(\mathbb{D}_n)$ such that for any $i \in [n]$,

$$\Big|(\sum_{d:z \neq X_i(\mathbf{t}_i)} g(u,z,Z_i))^2 - (\sum_{d:z \neq X_i(\mathbf{t}_i)} g(\tilde{u},z,Z_i))^2\Big| \leq 2K_1 K_2 |\mathcal{S}|^2 \delta \overset{\triangle}{=} \eta_3 \delta.$$

Thus, taking $\delta_1 = \frac{t}{7\eta_3 \mathcal{D}}$, by union bound, we have

$$
\mathbb{P}\Big(\mathcal{B}_2\Big) \leq \frac{2}{1-\eta_2}\mathbb{P}\Big(\exists (u,d) \in \mathcal{G}_n \times [\mathcal{D}], \frac{1}{n}\sum_{i=1}^{n}\epsilon_i\Big\{\sum_{d:z\neq X_i(\mathbf{t}_i)}g(u,z,Z_i)\Big\}^2
$$

$$
> \frac{3}{11}\Big[\frac{t}{\mathcal{D}} + \frac{1}{n}\sum_{i=1}^{n}\Big\{\sum_{d:z\neq X_i(\mathbf{t}_i)}g(u,z,Z_i)\Big\}^2\Big]\Big)
$$

$$
\leq \frac{2}{1-\eta_2}\sum_{d=1}^{\mathcal{D}}\mathbb{P}\Big(\exists u \in \mathcal{H}_{\delta_1}^d(\mathbb{D}_n), \eta_3\delta_1 + \frac{1}{n}\sum_{i=1}^{n}\epsilon_i\Big\{\sum_{d:z\neq X_i(\mathbf{t}_i)}g(u,z,Z_i)\Big\}^2
$$

$$
> \frac{3}{11}\Big[\frac{t}{\mathcal{D}} + \frac{1}{n}\sum_{i=1}^{n}\Big\{\sum_{d:z\neq X_i(\mathbf{t}_i)}g(u,z,Z_i)\Big\}^2 - \eta_3\delta_1\Big]\Big)
$$

$$
= \frac{2}{1-\eta_2}\sum_{d=1}^{\mathcal{D}}\mathbb{P}\Big(\exists u \in \mathcal{H}_{\delta_1}^d(\mathbb{D}_n), \frac{1}{n}\sum_{i=1}^{n}\epsilon_i\Big\{\sum_{d:z\neq X_i(\mathbf{t}_i)}g(u,z,Z_i)\Big\}^2
$$

$$
> \frac{3t}{11\mathcal{D}} + \frac{3}{11n}\sum_{i=1}^{n}\Big\{\sum_{d:z\neq X_i(\mathbf{t}_i)}g(u,z,Z_i)\Big\}^2 - \frac{14}{11}\eta_3\delta_1\Big)
$$

$$
= \frac{2}{1-\eta_2}\sum_{d=1}^{\mathcal{D}}\mathbb{P}\Big(\exists u \in \mathcal{H}_{\delta_1}^d(\mathbb{D}_n), \frac{1}{n}\sum_{i=1}^{n}\epsilon_i\Big\{\sum_{d:z\neq X_i(\mathbf{t}_i)}g(u,z,Z_i)\Big\}^2
$$

$$
> \frac{t}{11\mathcal{D}} + \frac{3}{11n}\sum_{i=1}^{n}\Big\{\sum_{d:z\neq X_i(\mathbf{t}_i)}g(u,z,Z_i)\Big\}^2\Big).
$$

Conditioning on $\mathbb{D}_n$, by Hoeffding's inequality, we have

$$
\mathbb{E}\Big\{\sum_{d=1}^{\mathcal{D}}\mathbb{P}\Big(\exists u \in \mathcal{H}_{\delta_1}^d(\mathbb{D}_n), \frac{1}{n}\sum_{i=1}^{n}\epsilon_i\Big\{\sum_{d:z\neq X_i(\mathbf{t}_i)}g(u,z,Z_i)\Big\}^2
$$

$$
> \frac{t}{11\mathcal{D}} + \frac{3}{11n}\sum_{i=1}^{n}\Big\{\sum_{d:z\neq X_i(\mathbf{t}_i)}g(u,z,Z_i)\Big\}^2\Big|\mathbb{D}_n\Big)\Big\}
$$

$$
\leq \mathbb{E}\Big\{2\sum_{d=1}^{\mathcal{D}}\mathcal{N}_{|\mathcal{S}|n}(\delta_1, \mathcal{H}_{\delta_1}^d(\mathbb{D}_n), L^\infty)]
$$

$$
\times \max_{u \in \mathcal{H}_{\delta_1}^d(\mathbb{D}_n)}\exp\Big(-\frac{cn\Big(\frac{t}{11\mathcal{D}} + \frac{3}{11n}\sum_{i=1}^{n}\Big\{\sum_{d:z\neq X_i(\mathbf{t}_i)}g(u,z,Z_i)\Big\}^2\Big)^2}{\frac{1}{n}\sum_{i=1}^{n}\Big\{\sum_{d:z\neq X_i(\mathbf{t}_i)}g(u,z,Z_i)\Big\}^4}\Big)\Big\}
$$

$$
\leq 2\mathcal{D}\mathcal{N}_{|\mathcal{S}|n}(\frac{t}{7\eta_3\mathcal{D}}, \mathcal{G}_n, L^\infty)\exp(-\frac{cnt}{K_1^2|\mathcal{S}|^2\mathcal{D}}).
$$

Consequently, we have

$$
\mathbb{P}(\mathcal{B}_2) \leq \Big(\frac{4\mathcal{D}}{1-\eta_2}\Big)\mathcal{N}_{|\mathcal{S}|n}(\frac{t}{7\eta_3\mathcal{D}}, \mathcal{G}_n, L^\infty)\exp(-\frac{cnt}{K_1^2|\mathcal{S}|^2\mathcal{D}}). \tag{16}
$$

**Bounding $\mathbb{P}(\mathcal{B}_3)$.** For $F(x) = x\log x$, if $a, b \in [c, C]$, then we have the strong convexity $D_F(a||b) \geq \frac{1}{2C}(a-b)^2$. Thus, for any $(u,d) \in \mathcal{G}_n \times [D]$, by equation 15 and QM-AM inequality,

we have

$$\mathbb{E}\Big[\sum_{d:z\neq X(\mathbf{t})} g(u,z,Z)\Big] = \mathbb{E}\Big[\sum_{d:z\neq X(\mathbf{t})} D_F(u_{\mathbf{t}}^0(z,X(\mathbf{t}))\|u_{\mathbf{t}}(z,X(\mathbf{t})))\Big]$$

$$\geq \frac{C}{\overline{MM}_c}\mathbb{E}\Big[\sum_{d:z\neq X(\mathbf{t})} (u_{\mathbf{t}}^0(z,X(\mathbf{t})) - u_{\mathbf{t}}(z,X(\mathbf{t})))^2\Big]$$

$$\geq \frac{C}{\overline{MM}_cK_2^2}\mathbb{E}\Big[\sum_{d:z\neq X(\mathbf{t})} g^2(u,z,Z)\Big] \qquad (17)$$

$$\geq \frac{C}{\overline{MM}_cK_2^2|\mathcal{S}|}\mathbb{E}\Big[\sum_{d:z\neq X(\mathbf{t})} g(u,z,Z)\Big]^2$$

$$\geq C\alpha\mathbb{E}\Big[\sum_{d:z\neq X(\mathbf{t})} g(u,z,Z)\Big]^2,$$

where $\alpha = (\overline{MM}_cK_2^2|\mathcal{S}|)^{-1} \wedge (3C^{-1}) \wedge (1/2)$. Then, by the definition of $\mathcal{B}_3$, equation 17 and introducing random signs, we have

$$\mathbb{P}(\mathcal{B}_3)$$

$$\leq \mathbb{P}\Big(\exists (u,d) \in \mathcal{G}_n \times [\mathcal{D}] : \frac{1}{n}\sum_{i=1}^n \sum_{d:z\neq X_i'(\mathbf{t}_i')} g(u,z,Z_i') - \frac{1}{n}\sum_{i=1}^n \sum_{d:z\neq X_i(\mathbf{t}_i)} g(u,z,Z_i)$$

$$> \Big(\frac{1}{2} - \frac{C\alpha}{12}\Big)\frac{t}{\mathcal{D}} + \frac{C\alpha}{24}\Big(\frac{1}{n}\sum_{i=1}^n\Big\{\sum_{d:z\neq X_i'(\mathbf{t}_i')} g(u,z,Z_i')\Big\}^2 + \frac{1}{n}\sum_{i=1}^n\Big\{\sum_{d:z\neq X_i(\mathbf{t}_i)} g(u,z,Z_i)\Big\}^2\Big)\Big)$$

$$\leq 2\mathbb{P}\Big(\exists (u,d) \in \mathcal{G}_n \times [\mathcal{D}] : \frac{1}{n}\sum_{i=1}^n \epsilon_i \sum_{d:z\neq X_i(\mathbf{t}_i)} g(u,z,Z_i)$$

$$> \Big(\frac{1}{4} - \frac{C\alpha}{24}\Big)\frac{t}{\mathcal{D}} + \frac{C\alpha}{24n}\sum_{i=1}^n\Big\{\sum_{d:z\neq X_i(\mathbf{t}_i)} g(u,z,Z_i)\Big\}^2\Big),$$

where $\{\epsilon_i\}_{i=1}^n$ is a sequence of i.i.d. Rademacher random variables, independent of $\mathbb{D}_n$. Note that, given $\mathbb{D}_n$ and $d \in [\mathcal{D}]$, for any $u \in \mathcal{G}_n$, there exists $\tilde{u} \in \mathcal{H}_\delta^d(\mathbb{D}_n)$ such that for any $i \in [n]$,

$$\Big|\sum_{d:z\neq X_i(\mathbf{t}_i)} g(u,z,Z_i) - \sum_{d:z\neq X_i(\mathbf{t}_i)} g(\tilde{u},z,Z_i)\Big| \leq K_2|\mathcal{S}|\delta \stackrel{\triangle}{=} \eta_4\delta.$$

Thus, if $\delta_2 = \frac{t}{(16\eta_4+2\eta_3)\mathcal{D}}$, conditioning on $\mathbb{D}_n$, by union bound and Bernstein's inequality (e.g., Lemma A.2 of Györfi et al., 2002), we have

$$2\mathbb{P}\Big(\exists (u,d) \in \mathcal{G}_n \times [\mathcal{D}] : \frac{1}{n}\sum_{i=1}^n \epsilon_i \sum_{d:z\neq X_i(\mathbf{t}_i)} g(u,z,Z_i)$$

$$> \Big(\frac{1}{4} - \frac{C\alpha}{24}\Big)\frac{t}{\mathcal{D}} + \frac{C\alpha}{24n}\sum_{i=1}^n\Big\{\sum_{d:z\neq X_i(\mathbf{t}_i)} g(u,z,Z_i)\Big\}^2\Big)$$

$$= 2\mathbb{E}\Big\{\sum_{d=1}^{\mathcal{D}}\mathbb{P}\Big(\exists u \in \mathcal{G}_n : \frac{1}{n}\sum_{i=1}^n \epsilon_i \sum_{d:z\neq X_i(\mathbf{t}_i)} g(u,z,Z_i)$$

$$> \Big(\frac{1}{4} - \frac{C\alpha}{24}\Big)\frac{t}{\mathcal{D}} + \frac{C\alpha}{24n}\sum_{i=1}^n\Big\{\sum_{d:z\neq X_i(\mathbf{t}_i)} g(u,z,Z_i)\Big\}^2\Big|\mathbb{D}_n\Big)\Big\}$$

$$\leq 2\mathbb{E}\Big\{\sum_{d=1}^{\mathcal{D}}\mathbb{P}\Big(\exists u \in \mathcal{H}_\delta^d(\mathbb{D}_n) : \eta_4\delta_2 + \frac{1}{n}\sum_{i=1}^n \epsilon_i \sum_{d:z\neq X_i(\mathbf{t}_i)} g(u,z,Z_i)$$

$$> \Big(\frac{1}{4} - \frac{C\alpha}{24}\Big)\frac{t}{\mathcal{D}} + \frac{C\alpha}{24n}\sum_{i=1}^n\Big\{\sum_{d:z\neq X_i(\mathbf{t}_i)} g(u,z,Z_i)\Big\}^2 - \frac{C\alpha\eta_3\delta_2}{24}\Big|\mathbb{D}_n\Big)\Big\}$$

$$\leq 2\mathbb{E}\Big\{ \sum_{d=1}^{\mathcal{D}} \mathcal{N}_{|\mathcal{S}|n}(\delta_2, \mathcal{H}_\delta^d(\mathbb{D}_n), L^\infty) \max_{u\in\mathcal{H}_\delta^d(\mathbb{D}_n)} \mathbb{P}\Big(\frac{1}{n}\sum_{i=1}^n \epsilon_i \sum_{d:z\neq X_i(\mathbf{t}_i)} g(u,z,Z_i)$$

$$> \Big(\frac{3}{16} - \frac{C\alpha}{24}\Big)\frac{t}{\mathcal{D}} + \frac{C\alpha}{24n}\sum_{i=1}^n \Big\{ \sum_{d:z\neq X_i(\mathbf{t}_i)} g(u,z,Z_i)\Big\}^2 \Big| \mathbb{D}_n\Big)\Big\}$$

$$\leq 4\mathbb{E}\Big\{ \sum_{d=1}^{\mathcal{D}} \mathcal{N}_{|\mathcal{S}|n}(\delta_2, \mathcal{H}_\delta^d(\mathbb{D}_n), L^\infty)$$

$$\times \max_{u\in\mathcal{H}_\delta^d(\mathbb{D}_n)} \exp\Big( - \frac{cn\Big[\Big(\frac{3}{16}-\frac{C\alpha}{24}\Big)\frac{t}{\mathcal{D}} + \frac{C\alpha}{24n}\sum_{i=1}^n\Big\{\sum_{d:z\neq X_i(\mathbf{t}_i)} g(u,z,Z_i)\Big\}^2\Big]^2}{\frac{2}{n}\sum_{i=1}^n\Big\{\sum_{d:z\neq X_i(\mathbf{t}_i)} g(u,z,Z_i)\Big\}^2 + \frac{2t}{\mathcal{D}}}\Big)\Big\}$$

$$\leq 4\mathcal{D}\mathcal{N}_{|\mathcal{S}|n}\big(\frac{t}{(16\eta_4+2\eta_3)\mathcal{D}}, \mathcal{G}_n, L^\infty\big) \exp\Big( - \frac{c\alpha nt}{\mathcal{D}}\Big),$$

where the last inequality we use the inequality
$$\frac{(a+u)^2}{a+bu} \geq \frac{4a}{b^2}[(b-1)\vee 0] \text{ for any } a,b,u>0.$$
Consequently, we have
$$\mathbb{P}(\mathcal{B}_3) \leq 4\mathcal{D}\mathcal{N}_{|\mathcal{S}|n}\big(\frac{t}{(16\eta_4+2\eta_3)\mathcal{D}}, \mathcal{G}_n, L^\infty\big) \exp\Big( - \frac{c\alpha nt}{\mathcal{D}}\Big). \tag{18}$$

**Deriving the final bound:** Combining Equation 16 and Equation 18 derived in previous parts, if $t\geq CK_1^2|\mathcal{S}|^2\mathcal{D}/n$ (note that $\alpha^{-1}\leq CK_1^2|\mathcal{S}|$ and $K_2\leq CK_1$), we have

$$\mathbb{P}\Big(\exists u\in\mathcal{G}_n : \mathbb{E}\sum_{d=1}^{\mathcal{D}}\sum_{d:z\neq X(\mathbf{t})} g(u,z,Z) - \frac{2}{n}\sum_{i=1}^n\sum_{d=1}^{\mathcal{D}}\sum_{d:z\neq X_i(\mathbf{t}_i)} g(u,z,Z_i) > t\Big)$$

$$\leq \frac{1}{1-\eta_1}(2\mathbb{P}(\mathcal{B}_2) + \mathbb{P}(\mathcal{B}_3))$$

$$\leq C\mathcal{D}\mathcal{N}_{|\mathcal{S}|n}\big(\frac{t}{CK_1^2|\mathcal{S}|^2\mathcal{D}}, \mathcal{G}_n, L^\infty\big) \exp\big(-\frac{cnt}{K_1^2|\mathcal{S}|^2\mathcal{D}}\big).$$

Thus, by taking $a_n = \frac{CK_1^2|\mathcal{S}|^2\mathcal{D}[\log\mathcal{N}_{|\mathcal{S}|n}(1/(2n),\mathcal{G}_n,L^\infty)+\log\mathcal{D}]}{n}$, we can obtain that

$$\mathbb{E}_\mathbb{D}\Big[\mathbb{E}_Z[\sum_{z\neq X(\mathbf{t})} g(\hat{u},z,Z)] - \frac{2}{n}\sum_{i=1}^n\sum_{z\neq X_i(\mathbf{t}_i)} g(\hat{u},z,Z_i)\Big]$$

$$\leq a_n + \int_{a_n}^\infty \mathbb{P}\Big(\mathbb{E}_Z[\sum_{z\neq X(\mathbf{t})} g(\hat{u},z,Z)] - \frac{2}{n}\sum_{i=1}^n\sum_{z\neq X_i(\mathbf{t}_i)} g(\hat{u},z,Z_i) > t\Big)\mathrm{d}t$$

$$\leq a_n + \int_{a_n}^\infty \exp\Big( - \frac{cnt}{K_1^2|\mathcal{S}|^2\mathcal{D}} + \log\mathcal{N}_{|\mathcal{S}|n}(1/(2n),\mathcal{G}_n,L^\infty) + \log\mathcal{D}\Big)\mathrm{d}t$$

$$\leq a_n + \frac{CK_1^2|\mathcal{S}|^2\mathcal{D}}{n}$$

$$\leq \frac{CK_1^2|\mathcal{S}|^2\mathcal{D}[\log\mathcal{N}_{|\mathcal{S}|n}(1/(2n),\mathcal{G}_n,L^\infty)+\log\mathcal{D}]}{n},$$

which completes the proof.

$\square$

### E.3 PROOF OF THEOREM 4

*Proof.* We follow the proof of Theorem 6 (2) in Chen & Ying (2024) and Theorem 1 in Zhang et al. (2025). We have
$$\mathrm{TV}(p_1, p_{1-\tau}) \leq \mathbb{P}(X(1)\neq X(1-\tau))$$

$$= 1 - \sum_{x_1} p(x_1) \prod_{d=1}^{\mathcal{D}} p_{t|1}^d(x_1^d|x_1^d)$$

$$= 1 - \left(\kappa_t + \frac{1 - \kappa_t}{|\mathcal{S}|}\right)^{\mathcal{D}}$$

$$= 1 - \exp\left\{\mathcal{D} \log\left(-(1 - \kappa_{1-\tau})\frac{|\mathcal{S}| - 1}{|\mathcal{S}|} + 1\right)\right\},$$

which completes the proof. $\qquad\square$

# F IMPLEMENTATION DETAILS AND ADDITIONAL EXPERIMENTS

## F.1 IMPLEMENTATION DETAILS

**Data.** We consider the data distribution with a blockwise AR(1) structure; that is, we first sample dimension $d = 1$ from $X(1)^1 \sim \mathcal{U}(\mathcal{S})$ and then we sample dimension $d = 2, 3$ from

$$X(1)^d|X(1)^{d-1} \sim \begin{cases} 0.9\,\mathcal{U}(X(1)^{d-1} + \{-2, -1, \ldots, 2\}) + 0.1\,\mathcal{U}(\mathcal{S}), & \text{if } X(1)^{d-1} \in [3, |\mathcal{S}| - 2] \\ \mathcal{U}(\mathcal{S}), & \text{otherwise} \end{cases},$$

and finally we sample $(X(1)^{3j-2}, X(1)^{3j-1}, X(1)^{3j})$ from distribution same as $(X(1)^1, X(1)^2, X(1)^3)$ for $j = 2, \ldots, \mathcal{D}/3$ (if $\mathcal{D} > 3$).

**Experimental Setup**

- sample size: $n \in \{2500, 5000, 7500, 10000, 12500\}$;
- dimension: $\mathcal{D} \in \{3, 6, 9, 12, 15\}$;
- early stopping parameter: $\tau \in \{0.001, 0.003, 0.005, 0.007, 0.01, 0.03, 0.05, 0.07, 0.1\}$.

**Training and Evaluation.** In our experiments, we consider the linear time scheduler $\kappa_t = t$. Note that $u_t^d(z^d, x) = \frac{1}{1-t}(p_{1|t}^d(z^d|x) - \delta_{x^d}(z^d))$ by equation 3 and equation 4. Thus, we can parameterize the posterior $p_{1|t}^{d,\theta}(z^d|x)$. By equation 4, the estimated transition rate is $\hat{u}_t(z, x) = \sum_{d=1}^{\mathcal{D}} \frac{1}{1-t}\delta_{x^{\backslash d}}(z^{\backslash d})(p_{1|t}^{d,\theta}(z^d|x) - \delta_{x^d}(z^d))$, where the parameters can be obtained by minimizing the following empirical risk (equivalent to the empirical version of ELBO derived in Equation 37 of Shaul et al. (2025)) based on data $\mathbb{D}_n$, which is equivalent to equation 5 (up to a constant not depending on $\theta$):

$$-\frac{1}{n}\sum_{i=1}^{n}\sum_{d=1}^{\mathcal{D}} \frac{1}{1 - \mathbf{t}_i}\left\{\left(1 - \delta_{X_i(1)^d}(X_i(\mathbf{t}_i)^d)\right)\log p_{1|t}^{d,\theta}(X_i(1)^d|X_i(\mathbf{t}_i)) - \delta_{X_i(1)^d}(X_i(\mathbf{t}_i)^d) + p_{1|t}^{d,\theta}(X_i(\mathbf{t}_i)^d|X_i(\mathbf{t}_i))\right\}.$$

Let $\{(\mathbf{t}_i', X_i'(\mathbf{t}_i'), X_i'(1))\}_{i=1}^{n_{\text{test}}}$ be a test dataset independent of $\mathbb{D}_n$. To evaluate the estimation error of the estimated transition rate $\hat{u}$, we use the following empirical prediction risk

$$-\frac{1}{n_{\text{test}}}\sum_{i=1}^{n_{\text{test}}}\sum_{d=1}^{\mathcal{D}} \frac{1}{1 - \mathbf{t}_i'}\left\{\left(1 - \delta_{X_i'(1)^d}(X_i'(\mathbf{t}_i')^d)\right)\log p_{1|t}^{d,\theta}(X_i'(1)^d|X_i(\mathbf{t}_i')) - \delta_{X_i(1)^d}(X_i'(\mathbf{t}_i')^d) + p_{1|t}^{d,\theta}(X_i'(\mathbf{t}_i')^d|X_i'(\mathbf{t}_i'))\right\}.$$

We set $n_{\text{test}} = 100,000$ in our simulation.

**Models** All our logit models use ReLU networks with 4 hidden layers with 256 dimensions. The optimizer is Adam with learning rate 1e-3. We train on the $\mathcal{D}$-dimensional dataset for $2000\mathcal{D}/3$ epochs with batch size 512.

---

**Algorithm 2** Sampling via Tau-leaping (Algorithm 1 in Campbell et al., 2022)

---

**Require:** A learned transition rate $\hat{u}$, an early stopping parameter $\tau > 0$, time partition $0 = t_0 < t_1 < \cdots < t_N = 1 - \tau$.

1: Draw $Y_0 \sim \mathcal{U}(\mathcal{S}^{\mathcal{D}})$.
2: **for** $k = 0$ to $N - 1$ **do**
3:     **for** $d = 1$ to $\mathcal{D}$ **do**
4:         **for** $s \in \mathcal{S} \backslash Y_k^d$ **do**
5:             Draw $P_{ds} \sim \text{Poisson}((t_{k+1} - t_k)\hat{u}_{t_k}^d(s, Y_k))$.
6:         **end for**
7:     **end for**
8:     **for** $d = 1$ to $\mathcal{D}$ **do**
9:         **if** $\sum_{s \in \mathcal{S} \backslash Y_k^d} P_{ds} > 1$ **then**
10:             $Y_{k+1}^d = Y_k^d$
11:         **else**
12:             $Y_{k+1}^d = Y_k^d + \sum_{s \in \mathcal{S} \backslash Y_k^d} P_{ds} \times (s - Y_k^d)$
13:         **end if**
14:     **end for**
15: **end for**
16: **return** $Y_N \sim \hat{p}_{1-\tau}$

---

### F.2   OVERALL PERFORMANCE EVALUATION

**Tau-leaping algorithm.** We present the tau-leaping algorithm (Algorithm 2) described in Campbell et al. (2022).

**Implementation Details.** We train our models with sample size 100,000. To assess the performance of uniformization and tau-leaping sampling algorithms in practice, we calculate the total variation of the empirical joint distribution of the first 3 dimensions ($8^3 = 512$ states in total) between the samples from the true data distribution and the generated samples. We choose $N = 100$ and the time partition $t_i = (1 - \tau) \times i/N$ for both algorithms, and $\lambda_{k+1} = \mathcal{D}/(1 - t_{k+1})$ for uniformization algorithm. For evaluation, we generate 500,000 samples from the true data distribution and 100,000 samples from discrete flow-based models using different sampling algorithms. We also record the runtime of each algorithm for sampling 100,000 samples.

**Overall Performance and Comparison between Uniformization and Tau-leaping.** The simulation results are presented in Table 1. From the simulation results, we can obtain the following conclusions.

- As the early stopping parameter $\tau$ increases, the total variation decreases first and then increases. The minimum is achieved between $\tau = 0.01$ and $\tau = 0.07$.
- As $\tau$ decreases, the runtime of uniformization increases and that of tau-leaping is almost fixed. This is because in each time interval $[t_k, t_{k+1}]$, the number of function calls depends on $t_{k+1}$ for uniformization algorithm and is fixed for tau-leaping algorithm.
- The uniformization sampling algorithm performs well for moderately small $\tau$, and is sometimes worse than tau-leaping algorithm especially for extremely small $\tau$. One possible explanation is that the tau-leaping algorithm might reduce the large estimation error caused by extremely small $\tau$.

## LLM USAGE

We only use LLMs for writing refinement. No ideas or scientific contributions are generated by LLMs.

Table 1: Total variation (on the joint distribution of the first 3 dimensions) and runtime with uniformization and tau-leaping algorithms.

| $\mathcal{D}$ | $\tau$ | Total Variation | | Runtime (s) | |
|---|---|---|---|---|---|
| | | Uniformization | Tau-leaping | Uniformization | Tau-leaping |
| | 0.01 | **0.0670** | 0.0679 | 31.3777 | 10.6464 |
| | 0.03 | **0.0551** | 0.0561 | 30.3401 | 10.6581 |
| 3 | 0.05 | 0.0547 | **0.0534** | 29.1027 | 10.9488 |
| | 0.07 | **0.0590** | 0.0608 | 28.6259 | 10.6501 |
| | 0.1 | **0.0613** | 0.0684 | 27.7543 | 10.6748 |
| | 0.01 | **0.0588** | 0.0612 | 44.1496 | 13.3098 |
| | 0.03 | **0.0488** | 0.0516 | 40.6444 | 13.6291 |
| 6 | 0.05 | **0.0509** | 0.0522 | 40.3871 | 13.8540 |
| | 0.07 | **0.0594** | 0.0639 | 37.7639 | 13.4209 |
| | 0.1 | **0.0647** | 0.0683 | 37.7448 | 13.3129 |
| | 0.01 | 0.0643 | **0.0628** | 56.9279 | 16.0244 |
| | 0.03 | **0.0653** | 0.0666 | 53.7150 | 16.4715 |
| 9 | 0.05 | **0.0539** | 0.0581 | 56.2539 | 15.9569 |
| | 0.07 | **0.0581** | 0.0612 | 50.1394 | 17.3870 |
| | 0.1 | **0.0689** | 0.0719 | 50.7919 | 16.6256 |
| | 0.01 | 0.1061 | **0.0980** | 73.9635 | 18.8918 |
| | 0.03 | **0.0584** | 0.0607 | 64.0545 | 19.3838 |
| 12 | 0.05 | **0.0610** | 0.0640 | 64.8737 | 18.3714 |
| | 0.07 | **0.0647** | 0.0672 | 62.9912 | 18.3160 |
| | 0.1 | **0.0680** | 0.0723 | 57.3696 | 19.2553 |
| | 0.01 | **0.0816** | **0.0816** | 82.3939 | 21.9227 |
| | 0.03 | 0.0627 | **0.0621** | 73.7929 | 21.7484 |
| 15 | 0.05 | **0.0718** | 0.0729 | 70.4029 | 20.3761 |
| | 0.07 | **0.0757** | 0.0775 | 68.3231 | 20.5265 |
| | 0.1 | **0.0784** | 0.0803 | 65.7229 | 21.0171 |

