# OpenReview forum: "Error Analysis of Discrete Flow with Generator Matching"
_ICLR.cc/2026/Conference — Submitted to ICLR 2026_

### Official Review · Reviewer_X6xy · 2025-10-21

**Soundness:** 3
**Presentation:** 3
**Contribution:** 3
**Rating:** 8
**Confidence:** 3

**Summary:**

This paper theoretically investigates the Discrete Flow Model (DFM). Specifically, the authors establish a change-of-measure framework between two Continuous-Time Markov Chains (CTMCs) with different transition rates and subsequently provide a formula on the Kullback-Leibler (KL) divergence of their path measures. Based on this framework, they provide a non-asymptotic convergence analysis for discrete flow models, which further considers the estimation error through a function class (e.g., neural networks). They partition the total distribution error into three distinct terms: stochastic error caused by the randomness of the data samples, approximation error that stems from the expressive limitations of the function class, and, finally, early-stopping error.

**Strengths:**

1. This paper presents the first theoretical analysis for the recently proposed discrete flow model.
2. The error decomposition presented in Section 5 is clearly written and highly intuitive, and the stochastic error bound (Theorem 3) for the estimation is a significant contribution that has been less explored in existing theoretical works.
3. The authors provide results for both general function classes and specific ReLU networks, which successfully demonstrates the generality of their theory.

**Weaknesses:**

1. The results on the general KL divergence bound (Theorem 1 and Theorem 2) are very similar to existing work for discrete diffusion models. I believe the authors should further elaborate on the novelty of this part; otherwise, it may appear to be a simple extension of previous works.
2. Although this is primarily a theoretical work, it would be beneficial to include empirical validations, such as numerical experiments, to demonstrate how the theory translates to practice.

**Questions:**

1. What is the main technical difficulty or contribution in Theorem 1 and Theorem 2 when compared to existing results for discrete diffusion models?
2. Although the uniformization algorithm provides exact simulation for CTMCs, it has a random running time. Is it possible, within this work's framework, to further analyze the error bound for algorithms with a deterministic running time (e.g., through truncation in the uniformization or by using other sampling algorithms)?

**Other Suggestions:**

1. Line 033: "Discrete flow models...**provide**...";
2. Line 077: "**Hamming** distance";
3. Line 277: "...two marginal distributions **regarding** two CTMCs...";
4. Page 23 in Appendix contains significant blank space; please consider reformatting.

---

> ### Author Response · Authors · 2025-11-21
>
> We thank the reviewer for the thoughtful comments and suggestions. We appreciate the time you spent on the paper and your recognition of our contributions and strengths. Below we address the concerns and comments that you have provided.
>
> **Q**: *The results on the general KL divergence bound (Theorem 1 and Theorem 2) are very similar to existing work for discrete diffusion models. I believe the authors should further elaborate on the novelty of this part; otherwise, it may appear to be a simple extension of previous works. What is the main technical difficulty or contribution in Theorem 1 and Theorem 2 when compared to existing results for discrete diffusion models?*
>
> **A**:  Thank you very much for your question. In contrast to the Girsanov's Theorem in existing work of discrete diffusion models [1,2], which requires the transition rate to be formulated as the product of a score function and a forward process rate, Theorem 1 of our paper presents the explicit form of the Radon–Nikodym derivative of two arbitrary CTMCs in terms of two transition rates associated with these CTMCs. This removes the structural restriction imposed in [1, 2] and therefore applies to a broader class of transition dynamics. Our formulation is fully consistent with the continuous-diffusion counterparts, e.g., Theorem 9 in [3] and Theorem 5.2.12 in [4], and can be viewed as their discrete-state analogue for general CTMCs. We have added a detailed comparison with these works following Theorem 1 in the revised version.
>
>
>
> **Q**: *Although this is primarily a theoretical work, it would be beneficial to include empirical validations, such as numerical experiments, to demonstrate how the theory translates to practice.*
>
> **A**:  Thank you very much for your question. We have added several simulation studies in Section 6 and Appendix F in the revised version of our paper. The corresponding code can be found in the supplementary material. From simulation results, we can find that the estimation error is nearly linear in $\mathcal{D}$, and the estimation error decreases as the early stopping parameter $\tau$ and sample size $n$ increase. We thank you for your suggestion on adding experiments in our paper, which is very helpful and valuable for improving the quality of our work.
>
> **Q**: *Although the uniformization algorithm provides exact simulation for CTMCs, it has a random running time. Is it possible, within this work's framework, to further analyze the error bound for algorithms with a deterministic running time (e.g., through truncation in the uniformization or by using other sampling algorithms)?*
>
> **A**:  Thank you very much for your suggestions. Note that the transition probability matrix from time $t_1$ to time $t_2>t_1$ is a matrix exponential $\exp(\int_{t_1}^{t_2} U_s \text{d} s)$, where $U_s=(u_s(z,x))_ {z,x\in\mathcal{S}^\mathcal{D}}$. If we knew the explicit form of this transition probability matrix, we could update the state $X(t_1)$ with the transition probability. One might consider using time-homogenuous rate matrix $U_{t_1}$ in the time interval $[t_1,t_2]$. However, since calculating $\exp((t_2-t_1) U_{t_1})$ is intractable for reasonably sized $|\mathcal{S}|$ and $\mathcal{D}$ as mentioned in [2] (Page 6), they also use uniformization for sampling (see Algorithm 1 in [2]), where the number of function calls in each time interval also depends on the value of Poisson variables. For tau-leaping algorithm (Algorithm 1 in [5]), at each sampling step, it only requires 1 function call and the number of jumps is not larger than $\mathcal{D}$, which has a deterministic running time; analyzing tau-leaping algorithm in our framework is nontrivial since it not only uses a piecewise time-homogenuous transition rate in the sampling stage but also restricts the number of jumps in each time interval.
>
>
> **Q**: *Typos in line 033, 077, 277, and page 23.*
>
> **A**:  Thank you very much for pointing out these typos. We have fixed them in the revised version.
>
> [1]  Le-Tuyet-Nhi Pham, et al. Discrete markov probabilistic models: An improved discrete score-based framework with sharp convergence bounds under minimal assumptions. ICML, 2025
>
> [2] Zhang et al. Convergence of score-based discrete diffusion models: A discrete-time analysis. ICLR, 2025
>
> [3] Sitan Chen, et al. Sampling is as easy as learning the score: theory for diffusion models with minimal data assumptions. ICLR, 2023.
>
> [4] David Applebaum. Levy processes and stochastic calculus. Cambridge University Press, 2009.
>
> [5] Campbell et al. A continuous time framework for discrete denoising models. NIPS, 2022.

---

> > ### Comment · Reviewer_X6xy · 2025-11-25
> >
> > Thank you for your thoughtful response. I have reviewed the updated sections, including the additional explanation for Theorem 1 and the new empirical studies. These revisions address my previous concerns.
> >
> > Overall, I will maintain my original score, as I believe this paper presents valuable contributions.

---

### Official Review · Reviewer_8JzY · 2025-10-21

**Soundness:** 3
**Presentation:** 2
**Contribution:** 3
**Rating:** 6
**Confidence:** 4

**Summary:**

The paper provides with error bounds for a class of discrete flow-matching models. Starting from conditional transition rates $ u_t(y,x_t|x_1) $, and an empirical data distribution $ (Z_i)_{i \leq N} $ the models considered here work in two phases as usual. In the first phase, a family of Markovian mimicking rates $ \hat{u}_t(z,x) $ is learned through empirical risk minimization. If the Bregman divergence used in this phase, the empirical risk is an approximation of the KL on path space between the Markov chain with rates $ u_t(z,x) $ and the law of the non-Markov process with conditional transition rates $ u_t(y,x_t|x_1) $. In the second phase, the terminal distribution of Markov chain with rates $ \hat{u}_t(z,x) $ is sampled, starting from the initial distribution applying the so-called uniformization technique. Importantly, this sampling algorithm avoids time discretization.

The main result of this paper is a bound between the sampled distribution and the data distribution which scales almost linearly in the underlying dimension $\mathcal{D}$. Three sources of errors are taken into account here: an early stopping error, the approximation error, accounting for the error introduced by the the fact that $\hat{u}$ is chosen within a given class of functions, and the stochastic error which originates from the fact that averages are taken with respect the n-samples empirical distribution of the data distribution rather than the data distribution itself.

In fact most of the paper's efforts and emphasis is put on the estimation of the stochastic error. The main result of the paper, Theorem 3, is indeed a bound on the stochastic error that requires the following assumptions

- The conditional transition rates are upper bounded. The optimal Markov rates, called oracle rates,  are uniformly lower bounded.

- That the function class in which $ \hat{u} $ is chosen is made of functions that are controlled from above and below by the oracle rates up to multiplicative constants

**Strengths:**

The main strength of the paper are the stochastic error estimates that scale linearly in $ \mathcal{D} $. This is the first result of this kind I am aware of in the context of discrete flow models. Given that discrete flow-matching techniques are becoming increasingly popular in applications, these results are definitely intereting for the community of researchers working in the are of generative modeling. Moreover, they align with other theoretical results known for continuous (diffusion) flow matching models. It is likely that the finding of this paper will motivate further research efforts on error bounds for discrete flow models. The presentation of the main results is fairly clear.

**Weaknesses:**

There are three mean weaknesses

- The assumptions are not directly imposed on the source and target data but rather on the transition rates, which are not available in closed form. This makes it hard to check them in practice and  very difficult to understand how the regularity of the data distribution determines the size of the constants appearing in the assumptions. The authors show that the boundedness of the conditional rates can be ensured under reasonable assumptions in Appendix A. However, when bounding from below the optimal Markov rates, they appeal to the mixing constant of the data distribution $\alpha$ and seem to impose that the marginals of the data distribution have some kind of uniform log-Lipschitz property, quantified through the constant $ \beta $. Both $ \alpha $ and $ \beta $ could be very large, and it is not clear whether or not $ \alpha $ and $ \beta $ would depend on the dimension when computed on relevant examples.

- The authors claim that one of the main contributions is to develop stochastic calculus and Girsanov theory for continuous time Markov chains. In my opinion, this not a novelty of this paper. Girsanov theory for jump processes has a long history starting with works in the 1970s by Jacod, see also book by Applebaum cited in this work for a Girsanov theorem for Lévy processes, the book by Jacod and Shiryaev for abstract results and many others. Specifying and making these abstract results explicit for the family of Markov chains studied in the area generative modeling is interesting in its own right, but it appears that this is not the first paper doing this. For example, the authors can check Appendix F in *"Discrete Markov Probabilistic Models: An Improved Discrete Score-Based
Framework with sharp convergence bounds under minimal assumptions"* by Pham et. al, available on Open Review.

- The proof of Theorem 3 is involved and very difficult to follow. I wanted to read it line-by-line as the bounds of Theorem 3 constitute the main strength of the paper, and had to give up and content myself with just getting an idea about is going on. There are literally pages of mathematical formulas without any intermediate comment about the overarching strategy. In some parts, there are some clear formatting issues, see [ages 23 and 24. Please avoid situations like the one at page 25 where you write 'following the proof of Theorem 11.6 in...' without actually saying what the proof ideas are, and then giving a page-long chain of inequalities.

To make a stronger case for publication, I recommend that the authors

- Exhibit a reasonably wide class of distributions and a choice of conditional rates for which the constants appearing in Assumptions 1 and 2 can be explicitly computed. If possible, show that certain forms of regularity on the data distribution imply bounds on these constants.

- Rethink section 4 by acknowledging previous results on Girsanov theory for CTMCs and their use in the context of discrete GMs. It would be beneficial to briefly compare the use of Girsanov Theorem in this work and the use of Girsanov theorem in the paper by Pham et al. Moreover, I feel like it's fair to recognize that the usage of Girsanov theorem in the context of diffusion models in $\mathbb{R}^d$ has been pioneered by Chen et al. in the paper *"Sampling is as easy as learning the score: theory for diffusion models with minimal data assumptions"*.

- Make the proof of Theorem 3 more reader friendly, by introducing some notation that avoids repeating the same long expressions over and over and by explaining what is the idea behind each of the steps. Also, think about including a proof outline for Thm, 3 right after its statement.

**Questions:**

- Please explain what do you mean by "marginalization trick". As far as I can see, it boils down to using the property of conditional expectation but I may be missing something

- I understand that the uniformization technique has the advantage of avoiding time-discretization. However I wonder about its interest in application. Are there existing works on discrete diffusion models that make use of this technique? What are the experimental results that have been obtained with this technique so far?

- I find the choice of denoting $u_t(z,x)$ the rate at which we jump from $x$ to $z$ very unorthodox and in contrast with the classical conventions for Markov chains. Is there a reason behind it?

- In the display corresponding to equation (6) there are some typos; in particular there is an odd number of parentheses in some of the equations.

- In the display in the statement of Theorem 3: E should be $\mathbb{E}$

- The definition of $\mathcal{N}(1/2n,\mathcal{G}_n,L^{\infty}(\mathbb{P}_n))$ should be given at some point in the paper. Recalling to the reader what is the covering number would also help

- In the statement of Theorem 3 can you clarify the (qualitative) dependence of $K_1$ on the constants appearing in Assumption 1 and 2?

---

> ### Author Response · Authors · 2025-11-21
>
> We are grateful for the reviewer’s thorough evaluation and valuable suggestions. Several of the suggestions regarding the assumptions and theoretical positioning have led to substantial clarifications and revisions in the updated manuscript.
>
> **Q1**: *The assumptions are not directly imposed on the source and target data but rather on the transition rates, which are not available in closed form. This makes it hard to check them in practice and very difficult to understand how the regularity of the data distribution determines the size of the constants appearing in the assumptions. The authors show that the boundedness of the conditional rates can be ensured under reasonable assumptions in Appendix A. However, when bounding from below the optimal Markov rates, they appeal to the mixing constant of the data distribution $\alpha$ and seem to impose that the marginals of the data distribution have some kind of uniform log-Lipschitz property, quantified through the constant $\beta$. Both $\alpha$ and $\beta$ could be very large, and it is not clear whether or not $\alpha$ and $\beta$ would depend on the dimension when computed on relevant examples.*
>
> **A**:  Thank you very much for your question. Finding a suitable assumption on discrete data distribution is difficult, and two constants $\alpha$ and $\beta$ might be large especially when $|\mathcal{S}|$ is extremely large. Here, we give a suitable example of data distribution as follows. We consider the distribution of the $d$-th dimension $x_1^d$ which locally depends on tokens $x_1^{d-k}, x_1^{d-k+1}\dots x_1^{d-1}$, where $k$ is a constant. For the mixing constant $\alpha$ we defined, consider an AR($k$) autoregressive structure for the sequence; that is, $p_1(x_1^{d}|x_1^{<d})=p_1(x_1^d|x_1^{((l-k)\vee 1):(d-1)})$ for $d>2$. Then we have
>     $$\begin{aligned} \frac{p_1\left(x_1^{\backslash d} \mid x_1^d\right)}{p_1\left(x_1^{\backslash d}\right)} & =\frac{p_1\left(x_1^1, \ldots, x_1^{d-1}\right) \prod_{l=d}^{\mathcal{D}} p_1\left(x_1^l \mid x_1^{((l-k) \vee 1):(l-1)}\right)}{p_1\left(x_1^1, \ldots, x_1^{d-1}\right) p_1\left(x_1^d\right) \prod_{j=1}^k p\left(x_1^{d+j} \mid x_1^{(((d-k) \vee 1):(d+j-1)) \backslash d}\right) \prod_{l=d+k+1}^{\mathcal{D}} p_1\left(x_1^l \mid x_1^{((l-k) \vee 1):(l-1)}\right)} \\\\ & =\frac{p_1\left(x_1^d \mid x_1^{((d-k) \vee 1):(d-1)}\right) \prod_{j=1}^k p_1\left(x_1^{d+j} \mid x_1^{((d+j-k) \vee 1):(d+j-1)}\right)}{p_1\left(x_1^d\right) \prod_{j=1}^k p\left(x_1^{d+j} \mid x_1^{(((d-k) \vee 1):(d+j-1)) \backslash d}\right)}\end{aligned}$$
> If $\log p_1(x_1^d)$ and $\log p_1(x_1^d|x_1^{((d-2k)\vee 1):(d-1)})\in[-c,0]$ for each $d\in[\mathcal{D}],x_1\in\mathcal{S}^\mathcal{D}$, then the RHS of the above equation satisfying $\alpha<\frac{p_1(x_1^{\backslash d}|x_1^d)}{p_1(x_1^{\backslash d})}<\alpha^{-1}$ with $\alpha=\exp(-(k+1)c)$ and $\beta= \exp(-c)$, which are independent of $\mathcal{D}$.
>
> **Q2**: *The authors claim that one of the main contributions is to develop stochastic calculus and Girsanov theory for continuous time Markov chains. In my opinion, this not a novelty of this paper. Girsanov theory for jump processes has a long history starting with works in the 1970s by Jacod, see also book by Applebaum cited in this work for a Girsanov theorem for Lévy processes, the book by Jacod and Shiryaev for abstract results and many others. Specifying and making these abstract results explicit for the family of Markov chains studied in the area generative modeling is interesting in its own right, but it appears that this is not the first paper doing this. For example, the authors can check Appendix F in "Discrete Markov Probabilistic Models: An Improved Discrete Score-Based Framework with sharp convergence bounds under minimal assumptions" by Pham et. al, available on Open Review.*
>
> **A**:  Thank you very much for your question. We fully agree Girsanov's Theorem for jump processes has been developed for a long time. In this work, we derive an explicit form of Girsanov-type theorem tailored to CTMCs like Girsanov's Theorem used in [2] and Theorem 5.2.12 in Applebaum's book [1] for Brownian motion with drift. In the concurrent work by Pham et al [3], its formulation of Girsanov's Theorem is tailored to discrete diffusion models, the reference path measure is associated with the forward CTMC with rate $q(x,y)$, and the time-reversal has the rate with the form $q^\prime_t(y,x)=U(t,x,y)q(x,y)$. Our setting requires comparing two CTMCs whose transition rates could be arbitrary. To address this, we present an explicit expression for the RN derivative between two general CTMC path measures instead of a specific form $U(t,x,y)q(x,y)$ used in discrete diffusion models.

---

> > ### Comment · Reviewer_8JzY · 2025-11-25
> >
> > I thank the authors for addressing my questions. I have the following remarks
> >
> > - **Q1** I thank the authors for providing an explicit example of a class of distributions where $\alpha$ can be explicitly computed. The probability measures in this class are close to product measures, since $ x_1^d $ only depends on a neighboring class of tokens/coordinates. In comparison with known results for diffusion models in continuous space, for which explicit convergence guarantees are available under minimal assumptions, such as finite entropy, or finite second moment, this is not very satisfactory. It still appears to me that the theoretical results of this work can only be made fully explicit and with good dependence on the dimension on a small class of distributions.
> >
> > - **Q2** I essentially disagree with the given answer. In contrast to what the authors write in their answer, Theorem F.12 in Pham et al. applies to two general Markov chains whose rates are arbitrary. The fact that $q^\prime_t(y,x)=U(t,x,y)q(x,y)$ is a mere notational choice and not a restriction.
> >
> > I therefore maintain my score.

---

> > > ### Author Response · Authors · 2025-11-27
> > > **Official Comment by Authors**
> > >
> > > We thank the reviewer for the additional comments and for clarifying their concerns.
> > >
> > > **Q1**: I thank the authors for providing an explicit example of a class of distributions where $\alpha$ can be explicitly computed. The probability measures in this class are close to product measures, since
> > >  only depends on a neighboring class of tokens/coordinates. In comparison with known results for diffusion models in continuous space, for which explicit convergence guarantees are available under minimal assumptions, such as finite entropy, or finite second moment, this is not very satisfactory. It still appears to me that the theoretical results of this work can only be made fully explicit and with good dependence on the dimension on a small class of distributions.
> > >
> > > **A1**: Thank you very much for your question. Unlike the squared loss in continuous diffusion models, which is strongly convex, in our proof, we use the quantity $\underline{M}_c$ (or $\alpha$) to provide the strong convexity of the Bregman divergence (induced by the convex function $F(x)=x\log x$ rather than a quadratic function), which is different from the scenario in the continuous diffusion models. Furthermore, we acknowledge that there might be some cases where $\alpha$ depends on dimension, but we did not restrict the $\alpha$ to have a locally dependent structure (each token would depend on any other token). We aim to provide an explicit example to verify that $\alpha$ is not related to $\mathcal{D}$ under some suitable data structure, since in practice the assumption is hard to verify. Relaxation of our conditions requires further investigation.
> > >
> > > **Q2**: I essentially disagree with the given answer. In contrast to what the authors write in their answer, Theorem F.12 in Pham et al. applies to two general Markov chains whose rates are arbitrary. The fact that
> > > $q_t^\prime(x,y)=U(t,x,y)q(y,x)$ is a mere notational choice and not a restriction.
> > >
> > > **A2**: We thank the reviewer for pointing this out and apologize for our misunderstanding. Indeed, we can derive the R-N derivative between two path measures associated with two CTMCs whose rates are arbitrary by applying their Theorem F.12 twice like in their proof of Theorem 2.3 (taking the stationary forward path measure $R$ as the reference measure in their Theorem F.12). We will clarify in our revised version.

---

> ### Author Response · Authors · 2025-11-21
>
> **Q3**: *The proof of Theorem 3 is involved and very difficult to follow. I wanted to read it line-by-line as the bounds of Theorem 3 constitute the main strength of the paper, and had to give up and content myself with just getting an idea about is going on. There are literally pages of mathematical formulas without any intermediate comment about the overarching strategy. In some parts, there are some clear formatting issues, see [ages 23 and 24. Please avoid situations like the one at page 25 where you write 'following the proof of Theorem 11.6 in...' without actually saying what the proof ideas are, and then giving a page-long chain of inequalities.*
>
> **A**:  Thank you very much for your question. We apologize for our presentation of the proof of Theorem 3. We have added the sketch of the proof of Theorem 3 and the idea in each step in the revised version.
>
> **S1**: *Exhibit a reasonably wide class of distributions and a choice of conditional rates for which the constants appearing in Assumptions 1 and 2 can be explicitly computed. If possible, show that certain forms of regularity on the data distribution imply bounds on these constants.*
>
> **A**:  Thank you very much for your suggestion. We provide a class of distributions and a choice of conditional rates  in Appendix B.2 in the revised version. An AR(k) autoregressive structure imply bounds on these constants.
>
> **S2**: *Rethink section 4 by acknowledging previous results on Girsanov theory for CTMCs and their use in the context of discrete GMs. It would be beneficial to briefly compare the use of Girsanov Theorem in this work and the use of Girsanov theorem in the paper by Pham et al. Moreover, I feel like it's fair to recognize that the usage of Girsanov theorem in the context of diffusion models in $\mathbb{R}^d$ has been pioneered by Chen et al. in the paper "Sampling is as easy as learning the score: theory for diffusion models with minimal data assumptions".*
>
> **A**:  Thank you very much for your suggestion. We added a detailed comparison with these works following Theorem 1 in the revised version.
>
> **S3**: *Make the proof of Theorem 3 more reader friendly, by introducing some notation that avoids repeating the same long expressions over and over and by explaining what is the idea behind each of the steps. Also, think about including a proof outline for Thm, 3 right after its statement.*
>
> **A**:  Thank you very much for your suggestion. We included the definition of Uniform Covering Number in Definition 3 and improved the proof of theorem 3 in the revised version.
>
> **Q4**: *Please explain what do you mean by "marginalization trick". As far as I can see, it boils down to using the property of conditional expectation but I may be missing something*
>
> **A**:  Thank you very much for your question. Yes, we mean taking conditional expectation for conditional transition rate given $X(\mathbf{t})$ and $\mathbf{t}$, which is an argument from [7], i.e., $u_t(z,x)=\mathbb{E}[u_\mathbf{t}(z,x|X(1))|X(\mathbf{t})=x,\mathbf{t}=t]$.
>
> **Q5**: *I understand that the uniformization technique has the advantage of avoiding time-discretization. However I wonder about its interest in application. Are there existing works on discrete diffusion models that make use of this technique? What are the experimental results that have been obtained with this technique so far?*
>
> **A**:  Thank you very much for your question. There are some papers using this uniformization in theoretical analysis, e.g., [4,8]. However, these papers did not incorporate experiments using uniformization technique. In our revised version, we conduct simulation experiments using uniformization technique. In Table 1 in Appendix F, we demonstrate the uniformization algorithm typically performs better than tau-leaping for moderately small $\tau$ with 3x computational costs. The corresponding code can be found in the supplementary material.

---

> > ### Author Response · Authors · 2025-11-21
> >
> > **Q6**: *I find the choice of denoting $u_t(z,x)$ the rate at which we jump from $x$ to $z$ very unorthodox and in contrast with the classical conventions for Markov chains. Is there a reason behind it?*
> >
> > **A**:  Thank you very much for your question. We use the notation $u_t(z,x)$ from papers [5,6].
> >
> > **Q7**: *typos and readability.*
> >
> > **A**:  Thank you very much for your suggestion. We fixed all of the typos in the revised version.
> >
> > [1] David Applebaum. Levy processes and stochastic calculus. Cambridge University Press, 2009.
> >
> > [2] Sitan Chen, et al. Sampling is as easy as learning the score: theory for diffusion models with minimal data assumptions. ICLR, 2023.
> >
> > [3] Le-Tuyet-Nhi Pham, et al. Discrete markov probabilistic models: An improved discrete score-based framework with sharp convergence bounds under minimal assumptions. ICML, 2025.
> >
> > [4] Zhang et al. Convergence of score-based discrete diffusion models: A discrete-time analysis. ICLR, 2025.
> >
> > [5] Itai et al. Discrete flow matching. NIPS, 2024.
> >
> > [6] Neta Shaul et al. Flow matching with general discrete paths: a kinetic-optimal perspective. ICLR, 2025.
> >
> > [7] Lipman, Yaron, et al. Flow matching guide and code. arXiv preprint arXiv:2412.06264, 2024.
> >
> > [8] Hongrui Chen and Lexing Ying. Convergence analysis of discrete diffusion model: Exact implementation through uniformization. arXiv preprint arXiv:2402.08095, 2024.

---

### Official Review · Reviewer_9zVq · 2025-10-30

**Soundness:** 3
**Presentation:** 4
**Contribution:** 3
**Rating:** 8
**Confidence:** 3

**Summary:**

Using rigorous stochastic analysis, this paper establishes the first non-asymptotic error bound for distribution error of discrete flow models with uniformization technique.
The total error can be decomposed to estimation error and early stopping error.
To analyze the estimation error, they develop a Girsanov change of measure theorem for CTMCs, which can transform the KL between two path measures of CTMCs to some expectation of Bregman divergence between rate functions of CTMCs.
Based on it, the estimation error can be decomposed to two terms: stochastic error and approximation error, which can be dealt with using standard statistical learning theory.
And they analyze the early stopping error to get the final result.

**Strengths:**

The writing of this paper is great. Even though I am not a researcher in the field of generative models, I can still grasp the core content and contributions of the work through its clear presentation.
This paper employs rigorous stochastic analysis theory to establish a theoretical framework for analyzing the non-asymptotic errors of discrete flow matching algorithms. In particular, the authors clearly demonstrate a Girsanov-type change of measure theorem for CTMCs, which I believe will be highly valuable for subsequent related research.
Furthermore, the authors conduct a careful analysis of each type of error, complemented by intuitive explanations. They also provide an analysis of how to balance these errors.
In summary, I consider this to be a highly polished and well-executed work.

**Weaknesses:**

Given that the authors emphasize the Girsanov-type change of measure theorem for CTMCs as one of the contributions in Contribution 1, it is necessary for the paper to elaborate on the related work of Girsanov's theorem that is currently missing. Additionally, the paper should clarify the technical novelty or differences of the proposed theorem compared to previous works of this kind.

Minor Points:

In Proposition 2, there are typos regarding s.

In equation (3), the first p_t^d -> p_{t|1}^d

In line 248, Xi(ti) ∼ pt|1(·|Xi(1)) -> Xi(ti) ∼ pt_i|1(·|Xi(1))

**Questions:**

N/A

---

> ### Author Response · Authors · 2025-11-21
>
> We appreciate the reviewer’s helpful comments and the time invested in assessing our work. The feedback has been instrumental in refining both the presentation and the technical aspects of the paper.
>
> **Q1**: *Given that the authors emphasize the Girsanov-type change of measure theorem for CTMCs as one of the contributions in Contribution 1, it is necessary for the paper to elaborate on the related work of Girsanov's theorem that is currently missing. Additionally, the paper should clarify the technical novelty or differences of the proposed theorem compared to previous works of this kind.*
>
> **A**:  Thank you very much for your question. In contrast to the Girsanov's Theorem in existing work of discrete diffusion models [1][2], which requires the transition rate to be formulated as the product of a score function and a forward process rate, Theorem 1 of our paper presents the explicit form of the Radon–Nikodym derivative of two arbitrary CTMCs in terms of two transition rates associated with these CTMCs. This removes the structural restriction imposed in [1, 2] and therefore applies to a broader class of transition dynamics.
>
> Our formulation is fully consistent with the continuous-diffusion counterparts, e.g., Theorem 9 in [3] and Theorem 5.2.12 in [4], and can be viewed as their discrete-state analogue for general CTMCs. We have added a detailed comparison with these works following Theorem 1 in the revised version. We appreciate your suggestion, which has helped improve the clarity and completeness of our paper.
>
> **Q2**: *Typos in Proposition 2, equation (3), and line 248.*
>
> **A**:  Thank you very much for pointing out these typos. We have fixed them in the revised version.
>
>
> [1]  Le-Tuyet-Nhi Pham, et al. Discrete markov probabilistic models: An improved discrete score-based framework with sharp convergence bounds under minimal assumptions. ICML, 2025
>
> [2] Zhang et al. Convergence of score-based discrete diffusion models: A discrete-time analysis. ICLR, 2025
>
> [3] Sitan Chen, et al. Sampling is as easy as learning the score: theory for diffusion models with minimal data assumptions. ICLR, 2023.
>
> [4] David Applebaum. Levy processes and stochastic calculus. Cambridge University Press, 2009.

---

> > ### Comment · Reviewer_9zVq · 2025-11-27
> >
> > Thank you for your detailed response, i will keep my positive score unchanged

---

### Official Review · Reviewer_gwnn · 2025-10-30

**Soundness:** 3
**Presentation:** 3
**Contribution:** 3
**Rating:** 4
**Confidence:** 3

**Summary:**

This paper presents a rigorous theoretical analysis of discrete flow models, a class of generative models that operate on discrete state spaces using continuous-time Markov chains (CTMCs). While prior theoretical work has focused primarily on discrete diffusion models, this paper develops a complete error analysis framework for discrete flow matching.

The main technical inovation is a Girsanov-like theorem for CTMCs, which allows the authors to compute the KL divergence between two path measures of Markov chains with different transition rate matrices, which is a useful tool.The total error then decomposes  into:
1) Stochastic error (due to finite data and empirical estimation)
2) Approximation error (due to model capacity)
3) Early stopping error (due to singular behavior near t = 1).

Central is the use of uniformization, which represents a CTMC as a Poisson process with constant rate M. This makes simulation exact—eliminating the discretization error that plagues tau-leaping or Euler schemes—and simplifies the theoretical treatment of sampling and generalization.
The analysis provides explicit convergence rates for total variation distance between the learned and target distributions, matching those known for continuous flow matching in the limit. The work is purely theoretical but conceptually unifies diffusion, flow, and discrete generative modeling frameworks.

**Strengths:**

- Theoretical contribution: The paper develops the first formal stochastic analysis of discrete flow models, extending continuous-time flow matching theory to discrete state spaces.
- The derivation of a Girsanov-type theorem for CTMCs is technically elegant and significant; it bridges stochastic calculus and discrete dynamics.
- The Error decomposition (stochastic, approximation, early stopping) provides a clear framework for understanding training behavior and generalization.

**Weaknesses:**

Chief among the weaknesses is the lack of experiments: The paper is entirely theoretical. While the derivations are elegant, ICLR remains a machine learning venue, and some empirical validation, however small, would substantiate the claims.
What useful information can machine learners in practice derive from these bounds. This is not clear to me.
For example, a toy CTMC experiment comparing uniformization vs tau-leaping would illustrate the zero-discretization error advantage.
Demonstrating the predicted error scaling (with n, D, and tau) on a synthetic discrete dataset would strengthen the empirical relevance.
These weaknesses link, for me personally, to a limited connection to real-world use: Although discrete flows could apply to language, graph, or combinatorial data, the assumptions (bounded, smooth transition rates; fixed generator structures) are far from practical neural architectures.
This indicates for me that the manuscript might be better suited for a more technical publication outfit like TMLR. While the theoretical insights are definitely nontrivial, it’s not obvious how they translate into actionable design or training principles for practitioners.

**Questions:**

1) Is the uniformization technique essential for your analysis, or could similar results hold for non-uniformized CTMCs (where jump rates are not globally bounded by M)?
2) In practice, generator matching is optimized via stochastic gradient descent with minibatches. Do your stochastic error bounds extend to this setting, or do they assume exact gradient estimates? An experiment would be welcome here demonstrating the applicability of your approach to real world setups.
3) How does the error bound scale with the discrete system’s dimensionality D? Are your near-linear dependencies empirically realistic, or do they rely on sparsity in the transition structure?
4) In prior discrete diffusion work, tau-leaping introduced discretization error. Could you quantify, even empirically, how much uniformization improves accuracy for comparable computational cost?
5)How could the theoretical insights here guide model design or training? For example, does the early stopping trade-off suggest a concrete way to choose tau in practice?
6) How do you expect these bounds to hold for modern architectures used in discrete diffusion?

---

> ### Author Response · Authors · 2025-11-21
>
> We sincerely thank the reviewer for the detailed and constructive feedback. We appreciate the careful reading of our manuscript and the insightful comments that have helped us significantly improve the clarity and scope of the revision.
>
> **Q1**: *Chief among the weaknesses is the lack of experiments: The paper is entirely theoretical. While the derivations are elegant, ICLR remains a machine learning venue, and some empirical validation, however small, would substantiate the claims. What useful information can machine learners in practice derive from these bounds. This is not clear to me. For example, a toy CTMC experiment comparing uniformization vs tau-leaping would illustrate the zero-discretization error advantage. Demonstrating the predicted error scaling (with n, D, and tau) on a synthetic discrete dataset would strengthen the empirical relevance. These weaknesses link, for me personally, to a limited connection to real-world use: Although discrete flows could apply to language, graph, or combinatorial data, the assumptions (bounded, smooth transition rates; fixed generator structures) are far from practical neural architectures. This indicates for me that the manuscript might be better suited for a more technical publication outfit like TMLR. While the theoretical insights are definitely nontrivial, it’s not obvious how they translate into actionable design or training principles for practitioners.*
>
> **A**:  Thank you very much for your suggestions. We have added several simulation studies in Section 6 and Appendix F in the revised version of our paper. The corresponding code can be found in the supplementary material. From simulation results, we can find that the estimation error is nearly linear in $\mathcal{D}$, and the estimation error decreases as the early stopping parameter $\tau$ and sample size $n$ increase. For comparison between tau-leaping and uniformization algorithms, please refer to our response to question 5. For assumptions, our assumptions relax the strong assumptions on the estimation error in the prior works, which is reasonable in practice: the upper bound depends on the conditional transition rate we defined in practice, which has to be bounded from above; the lower bound of oracle rate depends on the time schedule and the structure of data distribution (in Appendix B2 we have further discussion and example for this assumption, e.g. an autoregressive data structure). This work aims to provide the convergence guarantees for the empirical risk minimization estimator in discrete flow-based models. Providing precise guarantees for estimation of the rate function is challenging, as it requires an understanding of the non-convex training dynamics of network optimization, which is out of our scope. In summary, we thank you for your suggestion on adding experiments to our paper, which is very helpful and valuable for improving the quality of our work.
>
> **Q2**: *Is the uniformization technique essential for your analysis, or could similar results hold for non-uniformized CTMCs (where jump rates are not globally bounded by M)?*
>
> **A**:  Thank you very much for your question. Indeed, uniformization plays a central role in our analysis, since it can exact simulate CTMCs in time intervals. For the second part of the question, in practice, the transition rate in the sampling stage is continuous in $t$, which means that the transition rate is bounded by a constant $M$ uniformly in the time interval $[0,1-\tau]$. That is why we have to avoid terminal time singularity by using the early stopping technique empirically.
>
> **Q3**: *In practice, generator matching is optimized via stochastic gradient descent with minibatches. Do your stochastic error bounds extend to this setting, or do they assume exact gradient estimates?*
>
> **A**:  Thank you very much for your question. In this work, our primary focus is on the estimator $\hat{u}$ based on empirical risk minimization and its statistical properties. While the optimization error is beyond the present scope, we believe studying the training dynamics of learning transition rate via neural networks would shed light on discrete flow models; we view this as a promising direction for future work.
>
> **Q4**: *How does the error bound scale with the discrete system’s dimensionality D? Are your near-linear dependencies empirically realistic, or do they rely on sparsity in the transition structure?*
>
> **A**:  Thank you very much for your question. We assess the estimation error with different $\mathcal{D}$ in our simulation study, which is nearly linear in $\mathcal{D}$ in our simulation results. Sparsity in the transition rate structure is crucial for both our derivation of the convergence rate and practical implementation, since a non-sparse transition rate is intractable in both the training and sampling stages.

---

> > ### Author Response · Authors · 2025-11-21
> >
> > **Q5**: *In prior discrete diffusion work, tau-leaping introduced discretization error. Could you quantify, even empirically, how much uniformization improves accuracy for comparable computational cost?*
> >
> > **A**:  Thank you very much for your suggestion. We conduct the experiments comparing uniformization and tau-leaping algorithms (see section 6 and Appendix F), where we find that the uniformization algorithm typically performs better than tau-leaping for moderately small $\tau$.
> >
> > **Q6**: *How could the theoretical insights here guide model design or training? For example, does the early stopping trade-off suggest a concrete way to choose tau in practice? How do you expect these bounds to hold for modern architectures used in discrete diffusion?*
> >
> > **A**:  Thank you very much for your question. We did not address the question of how to choose model architectures, when the network functions can be trained well and how to select an adaptive early stopping parameter in a concrete way, which may involve a complex mathematical formulation and requires further investigation; in practice, using a moderately small $\tau$ (e.g., 0.03 or 0.05) in the training stage and using the time-homogeneous rate $\hat{u}_{1-\tau}$ in the time interval $[1-\tau,1]$ in the sampling stage may help reduce overall error.

---

### Author Response · Authors · 2025-12-03

We would like to start by expressing our sincere appreciation for our Area Chair, especially in light of the unexpected incident that has increased the workload. We also thank the reviewers for their careful reading and insightful comments. To make our responses easy to follow, we summarize the main points of our rebuttal below:

Across the reviews, the strengths of our work are acknowledged.



**First theoretical analysis of discrete flow models with uniformization technique (Reviewer gwnn, Reviewer 9zVq,  Reviewer X6xy).**

Building on a Girsanov-type theorem for CTMCs with arbitrary rates, we established the first error analysis of discrete flow-based models under a generator matching framework. Using a uniforomization technique to exactly simulate CTMC, we take into account three sources of errors, including approximation error, stochastic error, and early stopping error.

**Clear presentation and intuitive explanation (Reviewer gwnn, Reviewer 9zVq, Reviewer X6xy).**

We conduct a careful analysis of each type of error, complemented by intuitive explanations. The error decomposition presented in Section 5 is clearly written and highly intuitive.

**Notable contribution of the main results (Reviewer 8JzY, Reviewer X6xy, Reviewer 9zVq).**

We develop a non-asymptotic stochastic error bound, which aligns with the result for continuous flow matching. We provide results for both general function classes and specific ReLU networks, which successfully demonstrates the generality of our theory. Additionally, we provide an analysis of how to balance these errors.

The main concerns are:

**The lack of experiments (Reviewer gwnn, Reviewer X6xy).**

Response: We have added several simulation studies in Section 6 and Appendix F in the revised version of our paper. The corresponding code can be found in the supplementary material. From simulation results, we can find that the estimation error is nearly linear in
$\mathcal{D}$, and the estimation error decreases as the early stopping parameter $\tau$ and sample size $n$ increase. The total variation
decreases first and then increases as $\tau$ increases, with the minimum achieved around $\tau$ = 0.03 or
0.05. We conduct the experiments comparing uniformization and tau-leaping algorithms, where we find that the uniformization algorithm typically performs better than tau-leaping for moderately small $\tau$.

**About our assumptions (Reviewer gwnn, Reviewer 8JzY).**

Response: By taking estimation error into consideration, we impose the boundedness condition on the oracle transition rate, which we further discuss in the Appendix. The upper bound $\overline{M}_c$ of conditional transition rates is naturally related to the early stopping parameter (with rate $O(1/\tau)$), since conditional transition rates would explode due to the terminal time singularity as $t\to 1^-$. For the lower bound $\underline{M}_c$, which is used to provide strong convexity of Bregman divergence induced by $F(x)=x\log x$ and is hard to compute in practice, we introduce a mixing constant $\alpha$ and a quantity $\beta$, and the $\underline{M}_c$ would be bounded from below if $\alpha$ and $\beta$ are bounded. In our revised version, we further verify the condition using a specific example of an autoregressive data structure.

**About comparison between our Theorem 1 and existing works (Reviewer 9zVq, Reviewer 8JzY, Reviewer X6xy).**

Response: As pointed out by reviewer 8JzY, our Theorem 1 is similar to Theorem F.12 in [1]. Indeed, we can also derive Theorem 1 by applying Theorem F.12 twice, similar to the proof of Theorem 2.3. Compared to the Girsanov’s theorem used in [2], our result can be applied to two CTMCs with arbitrary transition rates. In our result, we provide an explicit expression of the R-N derivative in terms of two transition rates. This form is also closely related to Theorem 9 in [3] and Theorem 5.2.12 in [4] for Brownian motion with drift. We have added the discussion below Theorem 1 in the revised version.

We believe these revisions address the concerns raised by the reviewers and have improved the overall clarity and strength of our paper.

[1]  Le-Tuyet-Nhi Pham, et al. Discrete markov probabilistic models: An improved discrete score-based framework with sharp convergence bounds under minimal assumptions. ICML, 2025

[2] Zhang et al. Convergence of score-based discrete diffusion models: A discrete-time analysis. ICLR, 2025

[3] Sitan Chen, et al. Sampling is as easy as learning the score: theory for diffusion models with minimal data assumptions. ICLR, 2023.

[4] David Applebaum. Levy processes and stochastic calculus. Cambridge University Press, 2009.

---

### Meta-Review · Area_Chair_GrpJ · 2026-01-06

**Summary:**

## Summary of the Paper

This paper develops the first convergence bounds for discrete flow models based on continuous-time Markov chains (CTMCs).

The central technical tool used throughout the paper is a Girsanov-type change-of-measure theorem for CTMCs. However, it appears that such results have already been established in the existing literature. This theorem is then used as a starting point to derive a decomposition of the overall distributional error between the learned model and the target data distribution, in a manner similar to studies developed for continuous diffusion or flow models.

Since the authors consider sampling via uniformization, the total error decomposes into three terms: a statistical error, an approximation error, and an early stopping error. Under appropriate assumptions, each of these terms can be bounded using tools borrowed from the literature. Nevertheless, the paper does not provide an explicit analysis of the sampling complexity, which remains unaddressed.

Finally, small numerical experiments are presented to illustrate the theoretical bounds derived in the paper.


## Summary of the Strengths

The reviewers generally agree that the paper makes an interesting theoretical contribution, in particular because:

- It provides the first convergence bounds for discrete flow models, addressing an important gap in the theoretical literature.

- It includes a statistical and approximation error analysis, which is missing in many existing works on discrete generative models.



## Summary of the Weaknesses

Most reviewers agree on the following points for improvement. Several weaknesses are identified, primarily concerning novelty, assumptions, clarity of exposition, and practical relevance.

- The results on the general KL divergence bounds (Theorems 1 and 2) appear very close to existing analyses for discrete diffusion models.

- Relatedly, the paper’s claim of novelty regarding the development of stochastic calculus and Girsanov theory for continuous-time Markov chains is inaccurate, as similar explicit derivations already exist.

- Another important concern lies in the assumptions underlying the main results. These assumptions are imposed on the transition rates rather than directly on the data distribution, yet these rates are not available in closed form in practice. As a result, some reviewers were unconvinced that these assumptions would be satisfied in most settings.

- One reviewer found the presentation and readability of the main technical result (Theorem 3) to be a significant challenge. Although this theorem constitutes one of the strongest contributions of the paper, its proof is long, dense, and difficult to follow.

- Finally, the numerical evaluation is limited.

## Assessment of the paper

One of the results that most reviewers initially identified as a key contribution of the paper is the derivation of a general Girsanov theorem for CTMCs. However, as pointed out by one of the reviewers, this result had in fact already been established in prior work. This was subsequently acknowledged by the authors in their revision and rebuttal, as they were apparently unaware of this earlier result at the time of the original submission. Nevertheless, I was surprised that results from [1] on discrete diffusion were not cited in the relevant section of the revised version of the paper. As a consequence, a substantial portion of the paper is not novel.

In light of this, the paper should, in my opinion, focus almost exclusively on Section 5, which, by its title, contains the main results. However, this section spans barely two and a half pages and is, in its current form, difficult to read, hard to parse, and incomplete. Overall, the presentation and organization of the paper require substantial revision, for the reasons detailed below.

First, the claimed main contributions of the paper should be updated to reflect the fact that the Girsanov theorem is not original. At present, nearly three pages are devoted to stochastic calculus preliminaries and general results for CTMCs that are already well known. Meanwhile, the real novel part of the paper, Section 5, is extremely dense, insufficiently explained, and does not adequately credit prior work on which the results build, as already noted by one of the reviewers. This lack of clarity is further illustrated by the fact that most reviewers did not clearly identify the main contributions of Section 5. Consequently, this section should be thoroughly revised to better motivate the results, clarify their statements and proofs, and explicitly discuss their novelty. In particular, important discussions currently deferred to the supplementary material should be integrated into the main paper.

More specifically, the analysis of the approximation error should be included in the main text, together with a clearer and more detailed discussion of the assumptions. On this subject, two reviewers found these assumptions to be very restrictive, while the authors appear to take the opposite view. After a careful reading, I am unable to form a clear opinion on this point, largely because the relevant discussion and derivations in App. B.3 are difficult to follow. This appendix should be substantially improved, with precise statements replacing vague or overly technical comments.

Furthermore, the paper should include a clearer discussion of the proofs of the main results, explicitly highlighting where the novelty lies. Based on my reading, the proofs consist of adaptations of existing results, which should be clearly cited and discussed in the main text.

Finally, Section 5 should include a comprehensive complexity analysis, including sampling complexity, which is currently entirely absent. This omission is particularly problematic given that sampling via uniformization (also known as thinning), while avoiding discretization error, is well known in the CTMC literature to be computationally expensive in general. On this point, the numerical experiments added during the rebuttal are, in my view, inconclusive.

For all these reasons, I recommend rejection of the paper and strongly encourage the authors to take this feedback into account in a substantial revision of their work.


[1] Le-Tuyet-Nhi Pham, et al. Discrete markov probabilistic models: An improved discrete score-based framework with sharp convergence bounds under minimal assumptions. ICML, 2025

**Reviewer Concerns:**

As detailed in my previous discussion, I believe that most of the reviewers’ concerns were not adequately addressed. The only exception is the numerical experiment requested by reviewer gwnn, which demonstrates the predicted error scaling.

**Reviewer Scores:**

In my opinion, since one of the claimed contributions was in fact already known, I believe that some of the reviewers would have lowered their scores as a result.

---

### Decision · Program_Chairs · 2026-01-26

Reject